# LlamaTales: Studying the Effects of Developmentally Inspired Training Data on Small Language Models

## Abstract

Prior work has found that training very small language models (SLMs) on synthetic children's stories allows them to generate coherent text, comparable to much larger models. These stories are claimed to encompass the vocabulary and factual knowledge base of a 3-4-year-old child, capturing the "essence of natural language." Because of these claims, it is tempting to attribute the findings to the high readability (i.e., simple language) of children's stories, drawing a parallel to how children learn language. Is the human concept of readability relevant in the context of language model training, or are these findings better explained by other properties of the data? In this study, we investigate this by first validating several automatic readability measures. We then create synthetic corpora with varying levels of readability and assess the coherence of text generated by SLMs trained on these corpora. We find that training on high readability text is not a prerequisite for coherent SLMs. Specifically, SLMs trained on data with substantially more complex language also exhibit the same abilities as those trained on simple language. Moreover, training on simple language does not lead to the earlier development of coherence during training.

## 1 Introduction

How small can language models (LMs) be and still speak coherent English? Eldan & Li (2023) find that transformer-based LMs produce coherent text with fewer than 10 million parameters when trained on `TinyStories`, a synthetic dataset of children's stories that uses words that most 3 to 4-year-old children typically understand. The claim is that such a dataset retains the essential elements of natural language—grammar, vocabulary, facts, and reasoning—while being smaller and more focused. However, by focusing solely on children's stories, it remains unclear whether these findings are due to the simple, easy-to-understand language (high *readability*) of the data, or other properties like its synthetic origin or limited diversity. [1]

Readability refers to how easily a text can be read and understood. High readability is a fundamental characteristic of children's stories, as their intended audience typically lacks the language skills needed to understand more complex texts. Rare vocabulary, complex grammatical structures, and abstract concepts can all reduce readability and hinder the learning process. Given that educators use readability assessments to align instructional material with the skill levels of their students, it is intriguing to consider that LMs might also benefit from a curriculum-based approach to learning language. Indeed, the findings of Eldan & Li (2023) could be interpreted as preliminary evidence supporting this idea.

However, before accepting such an interpretation, we must consider an additional confound: the data generation process. `TinyStories` was constructed using an instruction-tuned LM and a template-based prompt, with minimal variations among the different instances of the template. While the variations may prevent the LM from generating nearly identical

---

[1] We use the term "coherent" to describe text that exhibits properties we expect of quality human-authored text, such as being grammatically correct, logically connected, and naturally sounding.

stories, the stories might still share redundant features. For example, they could all start with "Once upon a time..." or follow a limited set of story templates where the details differ but the overall structure and connective elements remain similar. In other words, the resulting dataset may have low diversity, which in turn leads to low information content, making it well-suited for an SLM to learn effectively.

This alternative explanation may seem obvious to some readers. After all, LMs trained with maximum likelihood estimation on a homogeneous corpus will generate text that closely resemble that corpus. This phenomenon was even demonstrated with small character-level RNNs by Karpathy (2015). However, the research community remains interested in developmentally inspired LM training, as evidenced by the BabyLM challenge (Warstadt et al., 2023), which provides a pretraining corpus inspired by the language input received by children, including child-directed speech and children's literature. Additionally, citations of Eldan & Li (2023) suggest that the community is open to the possibility that the simple language properties of `TinyStories` aid the development of coherent SLMs (Yu et al., 2023; Frank, 2023; Edman & Bylinina, 2023; Muckatira et al., 2024; Haga et al., 2024; Feng et al., 2024; Zhang et al., 2024; Nguyen, 2024). For example, Haga et al. (2024) write "Recent studies have shown the benefits of mimicking human language acquisition. For instance, using child-oriented vocabulary and/or child-directed speech (CDS) as learning data improves learning efficiency."

Readers might also argue that the findings of Eldan & Li (2023) are unsurprising, given the well-documented effects of knowledge distillation (KD). Hinton et al. (2015) show that training a student model $S$ on the real-valued outputs of a teacher model $T$ is more effective than training $S$ on the original data $C$ used to train $T$. Informally, minimizing $D_{KL}(T \parallel S)$ is more effective than minimizing $D_{KL}(C \parallel S)$. Indeed, the procedure of Eldan & Li (2023) is a special case of KD. In the limit, sampling from $T$ to create a synthetic corpus recovers $T$ itself. Thus, training $S$ on this synthetic corpus is equivalent to minimizing $D_{KL}(T \parallel S)$. However, Eldan & Li (2023) focus on a particular form of distillation: generating short stories where the capacity of $T$ and the size of $C$ are many orders of magnitude larger than $S$ and the synthetic data sampled from $T$. Given these particularities, KD might not be the only factor at play. And even if we accept that KD explains the coherence of SLMs, it does not tell us which properties of the synthetic corpus facilitate learning.

In this work, we aim to better understand the properties of training data that result in coherent SLMs. Our contributions are summarized as follows:

1. Through empirical analysis, we show that the coherence of text generated by SLMs trained on `TinyStories` is not due to its developmentally inspired properties. In particular, we show SLMs trained on data with complex language also exihibit the same abilities as those trained on simple language. Moreover, training on simple language does not lead to earlier development of coherence during training.

2. We release our datasets for studying SLMs and our data generation process. Specifically, we illustrate how to create datasets akin to `TinyStories` for training coherent SLMs using only open-source LMs.

3. As a supporting contribution, we provide evidence that open-source LMs can effectively evaluate text readability, presenting a viable alternative to human annotators. This finding has pedagogical implications, as traditional readability tools used by educators often depend on basic heuristics such as word count and word length.

## 2 EXPERIMENTAL DESIGN

Our primary research question is: Are the coherent generation abilities of SLMs trained on `TinyStories` due to the data's developmentally inspired properties, namely high *readability*, or are other data properties more influential? If the abilities stem from high readability, then at least one of the following criteria must be true: **(a)** the abilities only manifest when trained on high readability data and not on low readability data, or **(b)** the emergence of these abilities occur earlier during training on high readability data compared to low

**Measuring Readability (Section 3)** We begin by validating the readability of `TinyStories`. Although the data is intended to be readable for 3-4 year old children, there has been no prior work confirming this claim. We employ several readability measures, including classic formulas, statistics derived from constituency parse trees, and LM-based evaluations. We find that all measures suggest `TinyStories` exhibits high readability. Given that readability is an inherently fuzzy and difficult-to-define property of text, we also assess how well each measure correlates with human judgments. We find that while all measures are correlated with human judgments, instructing a LM to output a score in natural language (LLM-as-a-Judge) demonstrates the highest correlation.

**Measuring Quality (Section 4)** We also consider two additional measures that serve as proxies for the concept of text *quality*. Specifically, we focus on *coherence*, as measured with LLM-as-a-Judge, and *perplexity*, computed by state-of-the-art LMs.[2] Since we lack a dataset of human judgments similar to the one used for validating readability measures, we validate these quality measures by examining their correlation with a ranking of open-source pretrained LMs. This ranking spans from toy (e.g., `pythia-70m`) to state-of-the-art models (e.g., `Llama-3.1-70B`). Although this ranking is not as ideal as human judgments, it is designed to be broadly acceptable to the research community by avoiding fine-grained distinctions and instead categorizing models into reasonable tiers. Our analysis reveals that both coherence and perplexity correlate with the ranking, with coherence demonstrating the strongest correlation. Importantly, coherence (but not perplexity) is uncorrelated with readability, reinforcing the idea that these are distinct properties of text.

**Data Synthesis (Section 5)** We generate our synthetic dataset of stories called `LlamaTales-GRE`, which follows a data generation process similar to `TinyStories`, but incorporates vocabulary found in graduate school entrance exams. We aim for `LlamaTales-GRE` to exhibit coherence similar to `TinyStories` but with lower readability. These properties are confirmed using the measures discussed above. Since the data generation process and the LMs used to create `TinyStories` are not open-source, there are potential confounds when comparing `TinyStories` and `LlamaTales-GRE`. To address these, we also generate `LlamaTales-Jr`, our open-source replication of `TinyStories`.

**Results (Section 6)** With our datasets generated and their properties validated, we proceed to train a suite of SLMs with varying capacities, ranging from 262K to 33M non-embedding parameters. We find that SLMs with as few as 9M parameters can produce text with coherence comparable to state-of-the-art LMs when trained on *any* of the three datasets described above, thus failing to meet criteria **(a)**. Furthermore, we observe that the ability to generate coherent text does not emerge earlier in the training process when using high readability text, thus failing to meet criteria **(b)**. As we failed to meet both criteria, we conclude that the high readability of `TinyStories` is not the cause of the coherent generation abilities of SLMs.

## 3 Measuring Readability

We begin by validating the readability of `TinyStories`. Although its stories are intended to be readable for 3-4 year old children, no prior work confirms this claim. But how does one measure readability in the first place? As described by Trott (2024), readability refers to the ease, accessibility, or comprehensibility of a text. Although defining this concept precisely is challenging, its implications are generally understood. For instance, *Gravity's Rainbow* is more difficult to read than *The Very Hungry Caterpillar*, irrespective of their quality. Due to the subjective nature of readability, human evaluators are often considered the gold standard for judgement, despite their own limitations (Clark et al., 2021). However, acquiring human judgement across a large number of documents is both expensive and time-consuming. Consequently, several automatic measures have been proposed. In the following sections, we introduce various measures such as classic readability formulas (3.1), grammar

---

[2]We use the term "state-of-the-art" to broadly refer to the families of performant open-source large language models that were released from 2023 onwards, such as `Llama`, `Qwen`, and `Mistral`.

Table 1: Statistics for the train splits of the datasets described in Section 5 as well as (absolute) Pearson **corr**elation coefficients against human judgments of readability (↑ is better). Random examples from each dataset are shown in Table A5. **Top:** Classic readability formulas (↓ is easier to read). **Mid-Top:** Statistics computed from running a constiuency parser over the sentences of each dataset (↑ suggests more grammatical complexity). **Mid-Bot:** The result of prompting `Llama-3.1-70B-Instruct` to judge readability and coherence (↑ is better). Perplexity is computed with `Llama-3.1-8B`, `Qwen2-7B`, and `Mistral-7B-v0.3` and averaged. Expanded statistics are shown in Table A3.

| | Corr. | TinyStories | LlamaTales-Jr | LlamaTales-GRE | FineWeb |
|---|---|---|---|---|---|
| **Automated Readability** | 0.47 | 2.9 | 2.9 | 12.4 | 13.1 |
| **Coleman–Liau** | 0.48 | 3.7 | 3.8 | 10.4 | 11.8 |
| **Dale–Chall** | 0.58 | 5.7 | 5.7 | 9.1 | 9.3 |
| **Flesch–Kincaid** | 0.49 | 2.4 | 2.2 | 9.6 | 10.7 |
| **Gunning Fog** | 0.50 | 4.6 | 3.8 | 11.7 | 12.1 |
| **Linsear Write** | 0.41 | 4.2 | 3.3 | 13.2 | 12.7 |
| **SMOG** | 0.53 | 5.7 | 5.4 | 11.3 | 12.6 |
| **Spache Readability** | 0.51 | 2.7 | 2.5 | 5.5 | 5.5 |
| **Depth / Sentence** | 0.34 | 6.8 | 6.4 | 10.6 | 9.5 |
| **Width / Sentence** | 0.34 | 5.1 | 4.7 | 8.0 | 7.5 |
| **Nodes / Sentence** | 0.36 | 19.6 | 17.2 | 42.1 | 37.8 |
| **Readability** | **0.74** | 92.6 | 92.7 | 64.8 | 68.2 |
| **Coherence** | 0.03 | 90.1 | 89.5 | 94.4 | 77.4 |
| **Total Documents** | | 4.9e6 | 3.6e6 | 2.0e6 | 2.0e6 |
| **Total Tokens** | | 9.2e8 | 1e9 | 1e9 | 1e9 |
| **Synthetic** | | Yes | Yes | Yes | No |
| **Source** | | GPT-3.5/4 | Llama-3.1-8B | Llama-3.1-8B | Web |

complexity derived from constituency parse trees (3.2), and instructing LMs to output scores in natural language (3.3). Finally, we conduct experiments to validate how well each of these measures correlates with human judgements of readability (3.4).

## 3.1 CLASSIC READABILITY FORMULAS

The simplest way to measure readability is through formulas, many of which have been developed over the years. These formulas are generally straightforward, focusing on various combinations of word, sentence, and syllable counts. A commonly used readability formula to estimate the appropriate US grade level for a document is the Flesch-Kincaid Grade Level (FKGL) formula (Kincaid et al., 1975):

$$\text{FKGL} = 0.39 \left( \frac{\text{total words}}{\text{total sentences}} \right) + 11.8 \left( \frac{\text{total syllables}}{\text{total words}} \right) - 15.59 \tag{1}$$

This formula penalizes long sentences and long words. Its simplicity makes it easy to compute, but does not fully capture the complexity of readability. While long words and sentences may correlate with readability, they do not account for all factors, such as differences in grammatical structure and conceptual difficulty. We employ `textstat`[3] to calculate readability using various established formulas, including FKGL. See Appendix A for the definitions of each formula. Table 1 shows that `TinyStories` exhibits significantly higher readability compared to a 1B token subset of `FineWeb` (Lozhkov et al., 2024), a corpus of human-authored text sourced from the internet. The inclusion of `FineWeb` in our analysis is motivated by Eldan & Li (2023), who compared SLMs trained on `TinyStories` to GPT-2 (Radford et al., 2019), which was likley trained on data similar to `FineWeb`.

---

[3]`https://github.com/textstat/textstat`

## 3.2 CONSTIUENCY PARSING

To assess how grammatical structure relates to readability, we run a constituency parser (Kitaev & Klein, 2018; Kitaev et al., 2019) over our datasets and derive statistics from the resulting parse trees. A constituency parse tree is a hierarchical representation of a sentence's syntactic structure, where each node represents a grammatical unit, such as a phrase or a word, and the branches illustrate the relationships between these units. Figure 1 provides an example of such a tree.

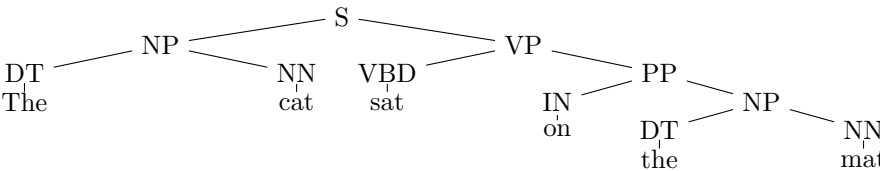

Figure 1: Constituency parse tree for the sentence "The cat sat on the mat."

Two sentences may have a similar number of words and words of similar length, but their parse trees can differ significantly, suggesting differences in grammatical complexity. To capture these differences, we compute three metrics for each sentence: (1) maximum depth, (2) maximum width, and (3) number of nodes. We average these metrics over all sentences and show the results in Table 1. Our findings suggest that `TinyStories` exhibits lower grammatical complexity compared to `FineWeb`.

## 3.3 LLM-AS-A-JUDGE

Finally, we consider leveraging instruction-tuned LMs to output readability scores in natural language, a technique commonly referred to as LLM-as-a-Judge. Although this approach is relatively new and the community has not yet reached a consensus on its efficacy, it offers several advantages. One of the main benefits is its fully automated nature, which allows it to scale well beyond the capabilities of human evaluations. Additionally, it is highly flexible, as the prompts can be adjusted to fit specific judgment criteria. However, there are also notable drawbacks, such as the need for prompt engineering and performance differences among models. Despite these challenges, there is a growing body of evidence suggesting that LLM-as-Judge is a reasonable alternative to hiring human annotators (Liu et al., 2023; Dubois et al., 2023; Sachdeva et al., 2024). For our study, we use `Llama-3.1-70B-Instruct` to generate readability scores for 10,000 documents from the training splits of `TinyStories` and `FineWeb`. We use the prompt proposed by Trott (2024), shown in Figure A10. Our findings in Table 1 again suggest that `TinyStories` exhibits higher readability than `FineWeb`.

## 3.4 CORRELATION WITH HUMAN JUDGEMENTS

We now validate how well our readability measures correlate with human judgments. Specifically, we follow the experimental setup of Trott (2024), who found that judgements by `gpt-4-1106-preview`, a closed-source LM, are highly correlated with humans (0.76 Pearson correlation coefficient). The human judgements were sourced from the CommonLit Ease of Readability corpus (`CLEAR`) (Heintz et al., 2022), comprising 5,000 excerpts from digital libraries like Project Gutenberg and Wikipedia. To assess readability, 1,800 teachers compared pairs of excerpts, determining which was easier to understand. These pairwise comparisons were then converted into individual readability scores for each excerpt using a Bradley-Terry model.

We apply our readability metrics to `CLEAR` and show their Pearson correlation coefficients in Figure A32. Notably, all metrics correlate with human judgments. Among these, LLM-as-Judge (70B parameters or more) shows the highest correlation and is competitive with `gpt-4-1106-preview`. This is followed by readability formulas, and lastly, by statistics derived from constituency parse trees. The performance of smaller LMs varies, with some performing comparably to readability formulas, while others perform worse. We show ex-

tended results for a large suite of open-source models in Figure A33 and readability formulas in A34. While our main goal is to identify reliable methods for measuring the readability of our data, we hope that our findings will also benefit others who utilize readability tools, such as educators.

## 4 MEASURING QUALITY

Next, we tackle the challenge of measuring text *quality*, a concept that is inherently subjective and difficult to define precisely, much like readability. Despite this, we believe that most readers have an intuitive understanding of what differentiates high-quality text from low-quality text. In this study, we employ two specific measures as proxies for text quality, acknowledging that there exist many other reasonable alternatives.

Our first measure is *perplexity*, computed by state-of-the-art LMs. This approach is based on the assumption that these LMs have learned a reasonable distribution over language. In theory, well-formed text should yield lower perplexity scores, while nonsensical or poorly constructed text should result in higher perplexity. However, we recognize that this method has limitations. For instance, it is possible for text that humans perceive as low quality to be assigned low perplexity, particularly if the text aligns with patterns frequently observed in the LM's training data.

Our second measure employs LLM-as-a-Judge to assess *coherence*. The specific prompt is presented in Figure A11. While text quality encompasses multiple dimensions, including clarity and fluency, we focus on coherence to align with the titular question of Eldan & Li (2023). We also note that Eldan & Li (2023) uses LLM-as-a-Judge (GPT-4) to evaluate their SLMs but with a different prompt. As LLM-as-a-Judge was covered in Section 3.3, we refrain from further elaboration.

### 4.1 CORRELATION WITH READABILITY

One of our running assumptions is that readability and text quality are orthogonal properties. Therefore, it is desirable for our measure of text quality to be uncorrelated with readability. To check this, we compute perplexity and coherence for the `CLEAR` dataset (Section 3.4). The results, presented in Figure A32, indicate that coherence is uncorrelated with both human and LM judgments of readability. In contrast, perplexity shows some correlation with readability. Consequently, we primarily focus on coherence as our measure of text quality for the remainder of this study, while also reporting perplexity where relevant.

### 4.2 CORRELATION WITH A RANKING OF LMS

Since we lack a dataset of human judgments specifically for text quality, we propose validating our quality measures against a ranking of open-source LMs. This ranking is designed to be broadly acceptable by the community by categorizing models into reasonable tiers rather than making fine-grained distinctions. We assign state-of-the-art LMs with 70B parameters as the highest tier. The next tier includes their smaller counterparts with 7B parameters. Finally, we include the Pythia suite (Biderman et al., 2023), ranking models in decreasing order based on parameter count. The ranking is detailed in Table A4.

Using prompts from the test splits of the datasets described in Section 5, we generate 1K documents for each dataset and model in our ranking, for a total of 56K documents. We then evaluate the generated text using our quality measures and present the Pearson correlation coefficients in Figure A31. The results show that coherence has the highest correlation with our ranking, followed by perplexity.

For the remainder of this study, we compute perplexity with `Llama-3.1-8B`, `Qwen2-7B`, and `Mistral-7B-v0.3`. These models are chosen for their efficiency and their similar correlation with our ranking compared to their larger counterparts. For coherence, we use `Llama-3.1-70B-Instruct` because of its high correlation with our ranking and to ensure consistency with our readability measures.

## 5 Data Synthesis

To address our research question, we require a version of `TinyStories` with stories intended for a more advanced readership than children. This allows us to isolate the effects of readability from other characteristics of `TinyStories`. However, the data generation process for `TinyStories` is not fully documented, and the data was generated using closed-source models (`GPT-3.5` and `GPT-4`). Fortunately, the authors released both the dataset and the prompts used for its generation. Using this information, we reproduce `TinyStories` with only open-source models and make our data generation process publicly available.

Specifically, for each story, we prompt `Llama-3.1-8B-Instruct` to generate a story using the template shown in Figure A21. We uniformly select 3 words without replacement from the same pool of 1,603 words used by `TinyStories`. These words include simple vocabulary found in children's stories such as "bed", "dog," and "snap". Additionally, we select $k$ features from a pool of 6 possible features for the story where $k$ follows a distribution estimated from the prompts used to generate `TinyStories`. These features include elements such as dialogue, a twist, or a bad ending. We instruct the generating model to incorporate these words and features into the story with top-$p$ sampling (Holtzman et al., 2020) where $p = 0.95$. We call the resulting dataset `LlamaTales-Jr`, a fully open-source dataset of children's stories. `LlamaTales-Jr` and `TinyStories` exhibit similar scores when evaluated for readability, grammatical complexity, and coherence. However, `LlamaTales-Jr` stories are generally longer, contain a higher number of unique $n$-grams, and exhibit higher perplexity compared to `TinyStories`.

With our data generation process in place, we increase reading difficulty by making two changes. First, we replace the pool of words from `TinyStories` with 2,880 words commonly found in GRE study materials, such as "encomium," "stevedore," and "jingoism." Second, we update the template to the one shown in Figure A22, allowing the model to generate stories beyond just children's stories. We call the resulting dataset `LlamaTales-GRE` and confirm that it exhibits greater reading difficulty and grammatical complexity (similar to `FineWeb`) compared to `LlamaTales-Jr`, while it scores marginally higher in coherence and perplexity.

Summary statistics and examples of our data are shown in Tables 1 and A5, respectively.

## 6 Results

Having established our datasets and evaluation measures, we proceed to train transformer LMs from scratch on the four datasets in Table 1. The LMs range from 262K to 33M non-embedding parameters and share the same design as `Llama-3`. We train each model for 10 epochs (10B tokens). We also consider a number of pretrained LMs, namely the `GPT-2`, `Pythia`, `Mistral`, `Qwen-2`, and `Llama-3` families. We do not finetune these pretrained LMs on our datasets. Details are shown in Table A2 and random generations are shown in Tables A6-A20.

For each combination of model and dataset, we generate 1K documents using top-$p$ sampling with $p = 0.95$, resulting in a total of 172K documents. To condition the generations, we extract 50-token prompts from the test splits of our data. We then evaluate the generations for coherence (Figure 2), perplexity (Figure A2), and readability (Figure A3). While our discussion emphasizes coherence for the sake of brevity, our findings hold true for readability and perplexity as well.

**Simple language is not a prerequisite for coherence.** SLMs trained on any of our synthetic datasets achieve coherence levels comparable to state-of-the-art LMs when given in-distribution prompts (i.e., prompts from the test split of the dataset they were trained on). Specifically, SLMs with 33M parameters produce text with coherence similar to models with 70B parameters, and SLMs with 9M parameters are competitive with 7B models. While these findings imply that training on high-readability data is not necessary for developing coherent SLMs, could it enable SLMs to develop coherence more efficiently (i.e., earlier in training)? Surprisingly, as shown in Figure 3, we find the opposite. SLMs trained on low

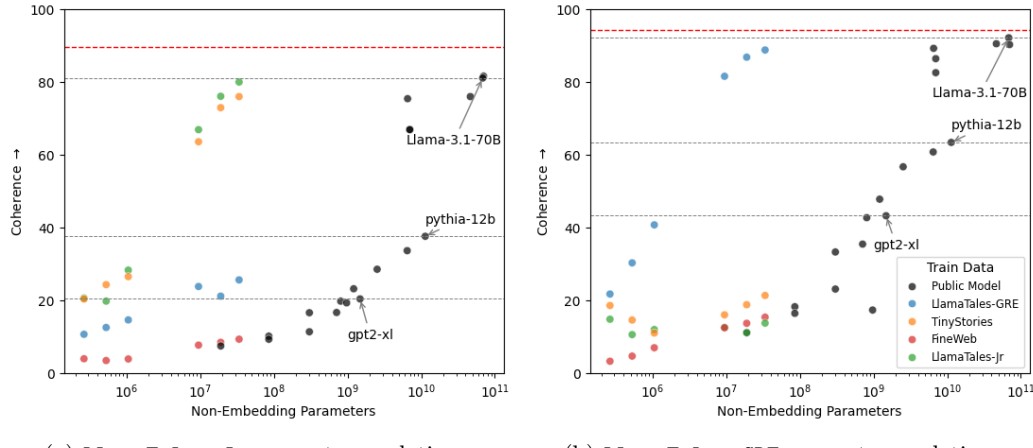

(a) `LlamaTales-Jr` prompt completions.   (b) `LlamaTales-GRE` prompt completions.

Figure 2: SLMs trained on `LlamaTales-Jr` and `LlamaTales-GRE` produce text with coherence comparable to that of much larger LMs. Prompts are extracted from the test splits each dataset. The legend colors represent the training data for each model. Public models (black) are found on Huggingface and are not fine-tuned on our data. The red horizontal line represents the coherence of the training split for the dataset in focus. For instance, in Figure 2a, the red line indicates the coherence `LlamaTales-Jr`'s training split. See Figure A1 for results on `TinyStories` and `FineWeb` prompt completions.

readability text achieve coherence more quickly than those trained on high readability text. While we do not expect this property to hold for all low readability texts, our evidence suggests that the developmentally inspired properties of `TinyStories` are not responsible for the coherence observed in SLMs.

**SLMs trained on our synthetic data are not robust.** Despite their impressive performance on in-distribution prompts, the SLMs trained on our synthetic data exhibit brittleness when faced with out-of-distribution prompts, leading to significant degradation in performance. For example, as illustrated in Figure 2, models trained on `LlamaTales-GRE` but prompted with `LlamaTales-Jr` show a noticeable drop in coherence. When provided with prompts from `LlamaTales-GRE` (in-distribution), they are competitive with state-of-the-art LMs. However, their performance declines to barely surpass that of `gpt2-xl` when given prompts from `LlamaTales-Jr` (out-of-distribution).

**SLMs trained on `FineWeb` are incoherent.** We observe that the coherence

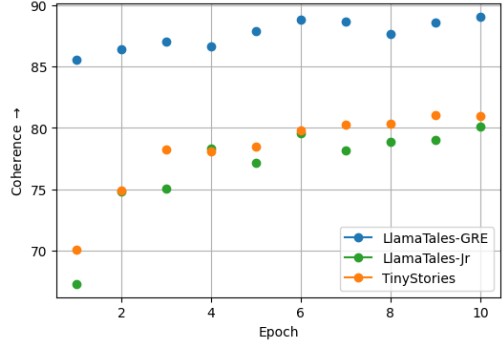

Figure 3: Coherence of 33M parameter LMs trained on synthetic datasets as training progresses. Coherence develops more slowly when trained on high readability text.

achieved by SLMs trained on our synthetic datasets does not extend to those trained on `FineWeb`. This is not entirely surprising, given that the coherence of `FineWeb` is lower than that of our synthetic datasets (77.4 compared to 90.1-94.4). However, the gap in coherence between an SLM and its training data is notably larger for `FineWeb` compared to our synthetic datasets. Specifically, all 33M parameter SLMs trained on synthetic data are within 10 points of their training data's coherence, whereas SLMs trained on `FineWeb` are more than 50 points away. This suggests that there may be a "learnability" aspect to data: training a LM on *any* dataset with a specific property does not imply that the trained model will go on to generate text exhibiting that property.

**Coherence is not explained by training data regurgitation.** A potential criticism of coherent SLMs is that their abilities might stem from merely reproducing their training data verbatim. However, the results in Figure 4 indicate otherwise. We consider the $n$-gram novelty (Merrill et al., 2024) of generated text with respect to the generator's training data. We confirm that our SLMs generate a substantial amount of novel text, although at a lower rate than models not trained directly on the same data (e.g., `Llama-3.1-70B` is not trained on `LlamaTales-Jr`).

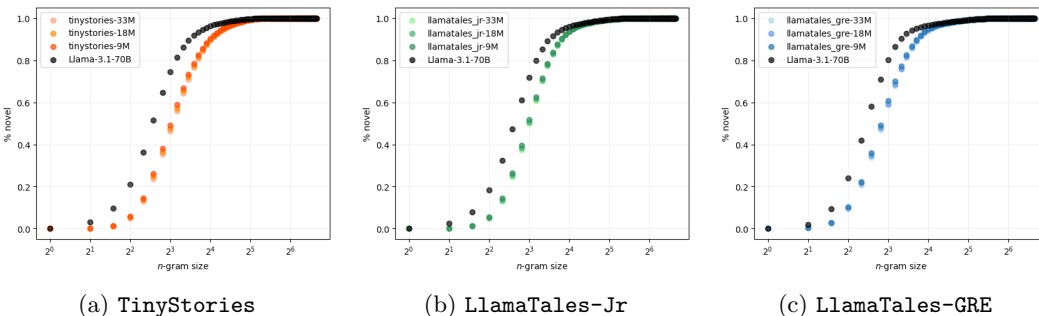

(a) `TinyStories`  (b) `LlamaTales-Jr`  (c) `LlamaTales-GRE`

Figure 4: We compare the unique $n$-grams in a model's training data and the text it generates when provided with prompts from test split documents. We find that SLMs do not merely regurgitate their training data. However, their level of $n$-gram novelty is lower compared to models that were not directly trained on the same data (e.g., `Llama-3.1-70B`). See Figure A37 for an alternative view of this figure.

**Knowledge distillation does not fully explain coherence.** So far, our findings support the hypothesis that knowledge distillation (KD) explains the results of Eldan & Li (2023), as training on synthetic data appears to be more effective than training on `FineWeb`, our representative of real-world LM pretraining data. To further investigate, we generate a synthetic version of `FineWeb` by extracting the first 50 tokens of each document in `FineWeb` and using `LLama-3.1-8B` to complete them. We call the resulting dataset `FineWeb-Synth`. We find no significant difference between SLMs trained on `FineWeb` and `FineWeb-Synth`, suggesting that KD alone does not account for the emergence of coherent SLMs.

## 7 DISCUSSION AND CONCLUSION

Our findings suggest that the ability of SLMs to generate coherent text when trained on `TinyStories` is not due to the simplicity of the language in the data. Training on a dataset with more complex language, such as `LlamaTales-GRE`, results in similar levels of coherence. Moreover, training on text with high readability does not accelerate the development of coherence during training; it may even delay it. However, the reasons for our observations remain unclear.

A seemingly straightforward explanation is that to develop a model with a specific property, such as coherence, we should train it on data that exhibits that property. For instance, both `TinyStories` and `LlamaTales-GRE` demonstrate similar levels of coherence, so it might not be surprising that LMs trained on coherent text can generate coherent text. However, if we accept this explanation, we would expect models trained on `FineWeb` to produce text with coherence similar to `FineWeb` itself. Instead, as shown in Figure A1d, there is a substantial gap between the coherence of `FineWeb` and the text generated by SLMs trained on this data. This gap is not observed in `TinyStories`, `LlamaTales-Jr`, and `LlamaTales-GRE`. That said, there are potential confounds to consider. First, the coherence of `FineWeb` is lower than that of our other datasets. It is possible that for the SLM to learn coherence, the dataset's coherence must be above a certain threshold. This could be addressed by using LLM-as-a-Judge to filter `FineWeb` documents to meet a certain coherence threshold. Additionally, the noisy nature of web crawl data may result in a higher variance in coherence across the documents of `FineWeb` compared to our other datasets. A low variance in a property like coherence might be necessary for effective learning of that property.

We can also consider the $n$-gram profiles of our data to explain our findings. We specifically consider the number of unique $n$-grams for each dataset. As shown shown in Figure A26, `FineWeb` exhibits significantly higher $n$-gram diversity compared to our easy-to-learn synthetic datasets, particularly for $n > 2$. Intuitively, high $n$-gram diversity suggests a dataset with rich and varied content, making it more challenging for a SLM to learn due to exposure to a broader range of linguistic structures and token combinations during training.

To further analyze the $n$-gram profile of our data, we examine the perplexity of an $n$-gram LM on its own training data. Unlike simply counting unique $n$-grams, this approach incorporates frequency information. We train $n$-gram LMs on each of our datasets, and the results, shown in Figure A30, reveal interesting patterns. For $n \leq 3$, the easy-to-learn datasets exhibit lower perplexity compared to `FineWeb`. However, this trend reverses for $n > 3$.

Why does this happen? Recall that the number of unique conditional probability tables (CPTs) that form an $n$-gram LM is determined by the number of unique $(n-1)$-grams in the training data, which grows exponentially with $n$. For larger values like $n = 4$, `FineWeb` has 413 *million* more CPTs than `LlamaTales-GRE`. This results in a dilution effect, where the CPTs in `FineWeb` are much sparser compared to those in `LlamaTales-GRE`, leading to lower entropy for the prior. In other words, given a 3-gram, there is less uncertainty in predicting the next token for `FineWeb` compared to `LlamaTales-GRE`. Conversely, for smaller values like $n = 2$, the number of unique CPTs remains relatively small, with `FineWeb` having only 57 thousand more than `LlamaTales-GRE`. This results in denser CPTs, where the higher diversity of `FineWeb` manifests as high-entropy CPTs. In `FineWeb`, there are simply many more possible completions given a 1-gram compared to `LlamaTales-GRE`.

We also observe that $n$-gram LM perplexities sometimes converge as we progress through a dataset. Specifically, for $n = 4$, the perplexities of `LlamaTales-GRE` and `FineWeb` are nearly identical when trained on the entire dataset. This could lead to the incorrect conclusion that there are no differences between the two datasets. However, if we consider how an LM is actually trained, we observe stark differences in perplexity as training progresses. These differences could result in varying outcomes in terms of learnability.

In conclusion, we found that using simple language in training data is not a prerequisite for coherence in SLMs. Measures like $n$-gram counts and $n$-gram LM perplexity might provide a better assessment of how easily a dataset can be learned. However, it is important to note that these metrics are currently only correlational, and we have not yet demonstrated a causal relationship. Future work will focus on validating these measures as indicators of learnability.

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

## A    Classic Readability Formulas

The readability formulas utilized in Table 1 are presented below. See Section 3.1 for details.

### A.1    Flesch-Kincaid Grade Level (Kincaid et al., 1975)

$$\text{FKGL} = 0.39 \left( \frac{\text{words}}{\text{sentences}} \right) + 11.8 \left( \frac{\text{syllables}}{\text{words}} \right) - 15.59 \tag{2}$$

### A.2    Automated Readability Index (Smith & Senter, 1967)

$$\text{ARI} = 4.71 \left( \frac{\text{characters}}{\text{words}} \right) + 0.5 \left( \frac{\text{words}}{\text{sentences}} \right) - 21.43 \tag{3}$$

### A.3    Coleman–Liau Index (Coleman & Liau, 1975)

$$\text{CLI} = 0.0588 \left( \frac{\text{characters}}{\text{words}} \times 100 \right) - 0.296 \left( \frac{\text{sentences}}{\text{words}} \times 100 \right) - 15.8 \tag{4}$$

### A.4    Dale–Chall Formula (Dale & Chall, 1948)

$$\text{DC} = 0.1579 \left( \frac{\text{difficult words}}{\text{words}} \times 100 \right) + 0.0496 \left( \frac{\text{words}}{\text{sentences}} \right) \tag{5}$$

Difficult words are defined as those not included in a list of 3,000 words that fourth-grade American students are expected to know. If the percentage of difficult words exceeds 5%, add 3.6365 to the score.

### A.5    Gunning Fog Index (Gunning, 1968)

$$\text{GFI} = 0.4 \left[ \left( \frac{\text{words}}{\text{sentences}} \right) + 100 \left( \frac{\text{complex words}}{\text{words}} \right) \right] \tag{6}$$

Complex words are defined as words with three or more syllables, excluding proper nouns, familiar jargon, and compound words.

### A.6    Linsear Write Formula (O'Hayre, 1975)

$$r = \frac{(\text{words} \leq 2 \text{ syllables}) + 3 \cdot (\text{words} \geq 3 \text{ syllables})}{\text{sentences}} \tag{7}$$

$$\text{Linsear Write} = \begin{cases} \frac{r}{2} & \text{if } r > 20 \\ \frac{r-2}{2} & \text{if } r \leq 20 \end{cases} \tag{8}$$

### A.7    SMOG Index (Harry & Laughlin, 1969)

$$\text{SMOG} = 1.0430 \sqrt{30 \left( \frac{\text{words} \geq 3 \text{ syllables}}{\text{sentences}} \right)} + 3.1291 \tag{9}$$

### A.8    Spache Formula (Spache, 1953)

$$\text{Spache} = 0.121 \left( \frac{\text{words}}{\text{sentences}} \right) + 0.082 \left( \frac{\text{difficult words}}{\text{words}} \times 100 \right) + 0.659 \tag{10}$$

Difficult words are defined as words that are not included in a list of familiar words that are typically known by fourth-grade students.

# B    ADDITIONAL FIGURES AND TABLES

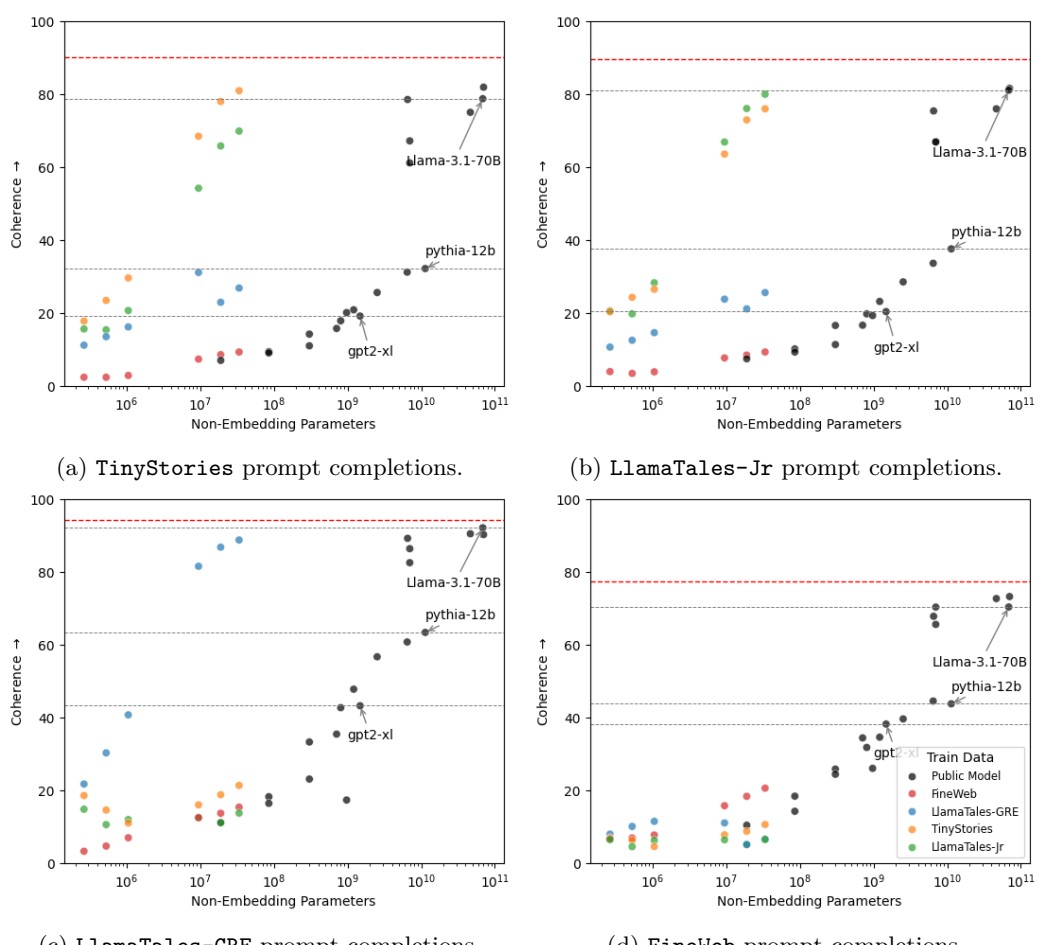

(a) `TinyStories` prompt completions.

(b) `LlamaTales-Jr` prompt completions.

(c) `LlamaTales-GRE` prompt completions.

(d) `FineWeb` prompt completions.

Figure A1: **Coherence** of text generated by the LMs listed in Table A2. Prompts are extracted from the test splits of our datasets in Table 1. The legend colors represent the training data for each model. Public models (black) are found on Huggingface and are not fine-tuned on our data. The red horizontal line marks the coherence of the train split of the dataset in focus. Return to Figure 2 (truncated results). Return to Section 7 (Discussion).

| Dataset | Coherence | Readability |
|---|---|---|
| TinyStories | 90.1 | 92.6 |
| LlamaTales-Jr | 89.5 | 92.7 |
| LlamaTales-GRE | 94.4 | 72.7 |
| LlamaTales-Sports | 92.4 | 72.4 |
| LlamaTales-News | 94.5 | 72.7 |
| LlamaTales-History | 91.0 | 61.4 |
| FineWeb | 77.6 | 68.2 |
| Dolma | 60.2 | 70.99 |
| SlimPajama | 67.3 | 66.6 |

Table A1: Coherence and readability scores for train splits.

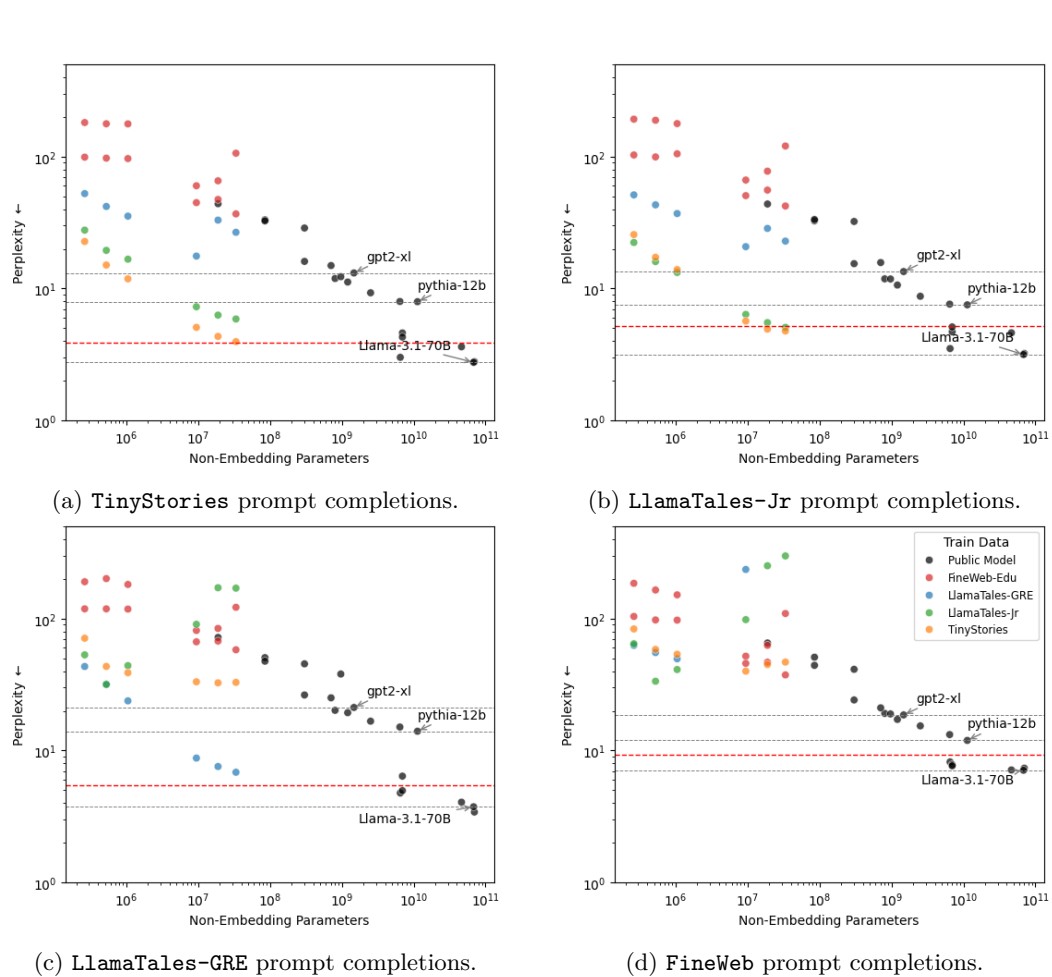

(a) `TinyStories` prompt completions.

(b) `LlamaTales-Jr` prompt completions.

(c) `LlamaTales-GRE` prompt completions.

(d) `FineWeb` prompt completions.

Figure A2: **Perplexity** (computed by external LMs) of text generated by the LMs listed in Table A2, based on prompts from the test split data in Table 1. See Section 4 for details on this metric. The legend colors represent the training data for each model. The red horizontal line marks the perplexity of the train split of the dataset in focus. Return to Section 6 (Results).

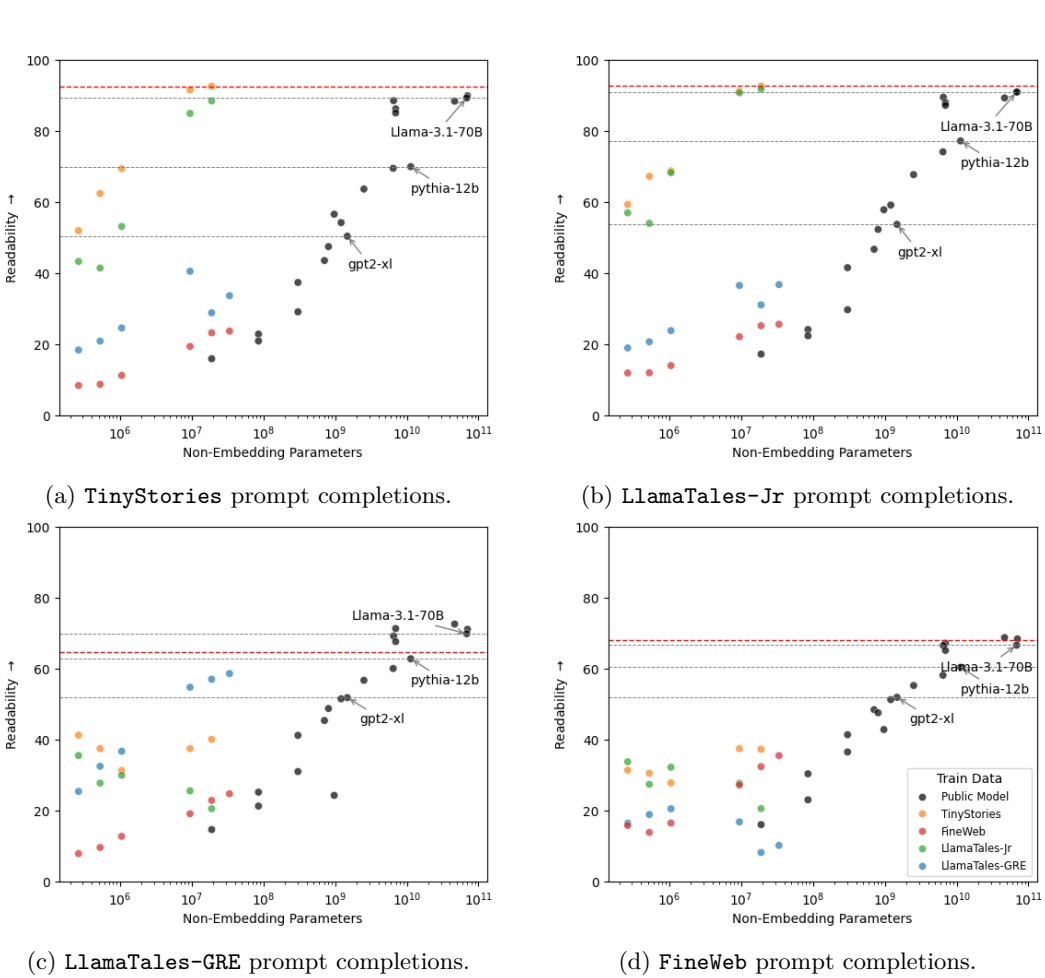

(a) `TinyStories` prompt completions.

(b) `LlamaTales-Jr` prompt completions.

(c) `LlamaTales-GRE` prompt completions.

(d) `FineWeb` prompt completions.

Figure A3: **Readability** of text generated by LMs in Table A2, based on prompts from the data (test split) in Table 1. The legend colors represent the training data for each model. The red horizontal line marks the readability of the train split of the dataset in focus. Return to Section 6 (Results).

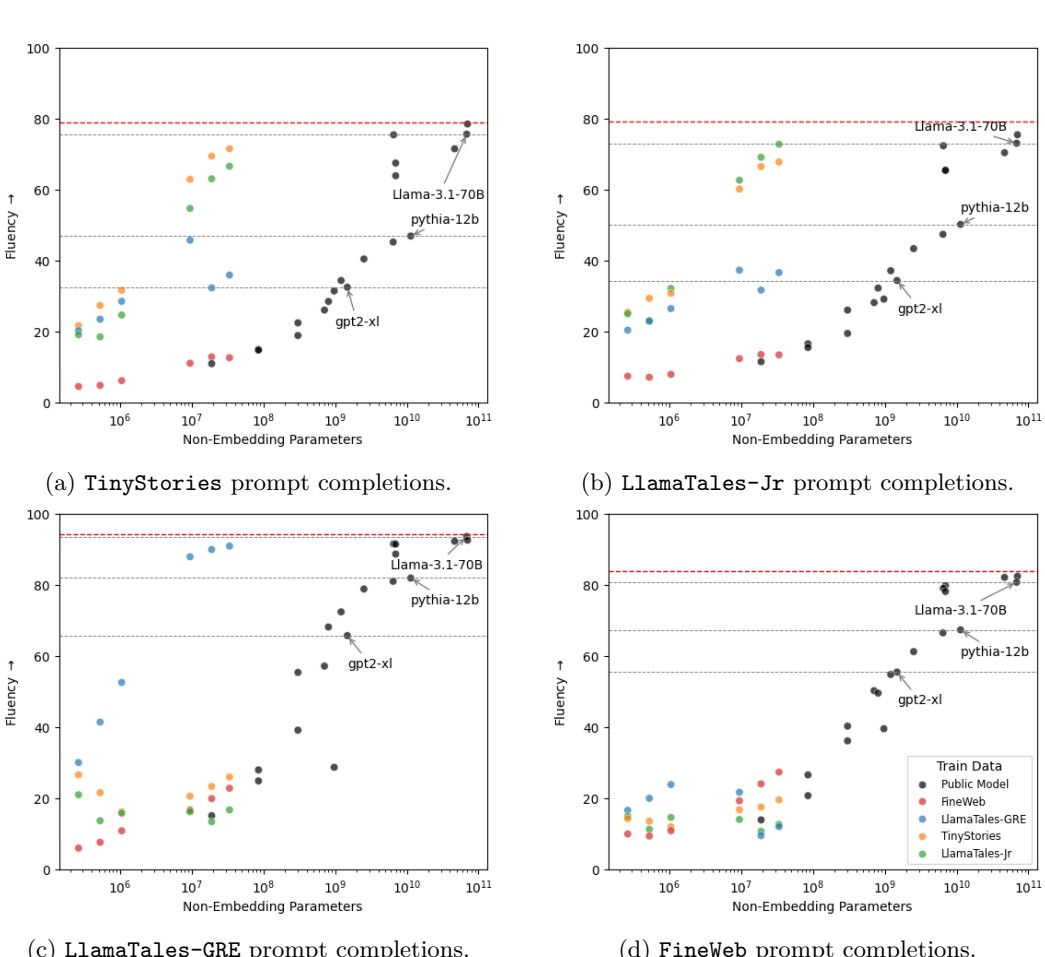

(a) `TinyStories` prompt completions.

(b) `LlamaTales-Jr` prompt completions.

(c) `LlamaTales-GRE` prompt completions.

(d) `FineWeb` prompt completions.

Figure A4: **Fluency** of text generated by LMs in Table A2, based on prompts from the data (test split) in Table 1. The legend colors represent the training data for each model. The red horizontal line marks the fluency of the train split of the dataset in focus.

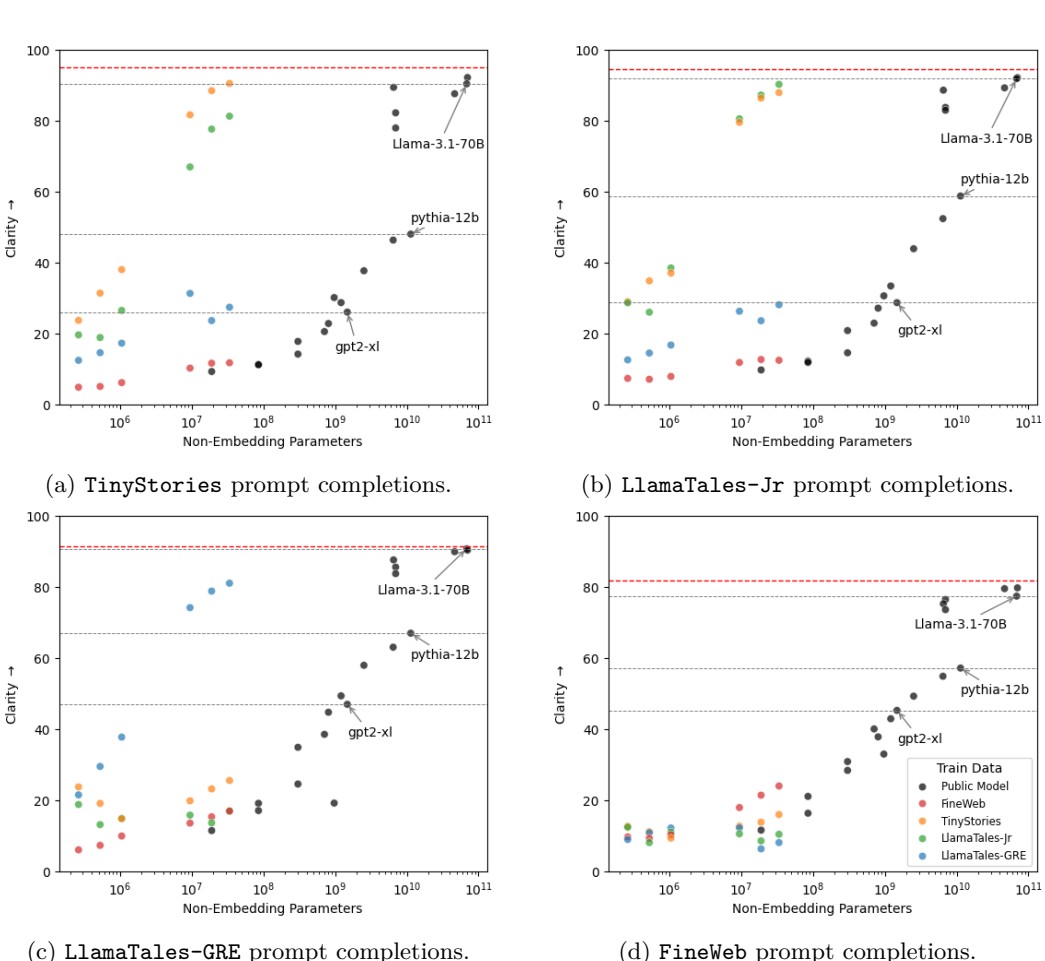

(a) `TinyStories` prompt completions.

(b) `LlamaTales-Jr` prompt completions.

(c) `LlamaTales-GRE` prompt completions.

(d) `FineWeb` prompt completions.

Figure A5: **Clarity** of text generated by LMs in Table A2, based on prompts from the data (test split) in Table 1. The legend colors represent the training data for each model. The red horizontal line marks the clarity of the train split of the dataset in focus.

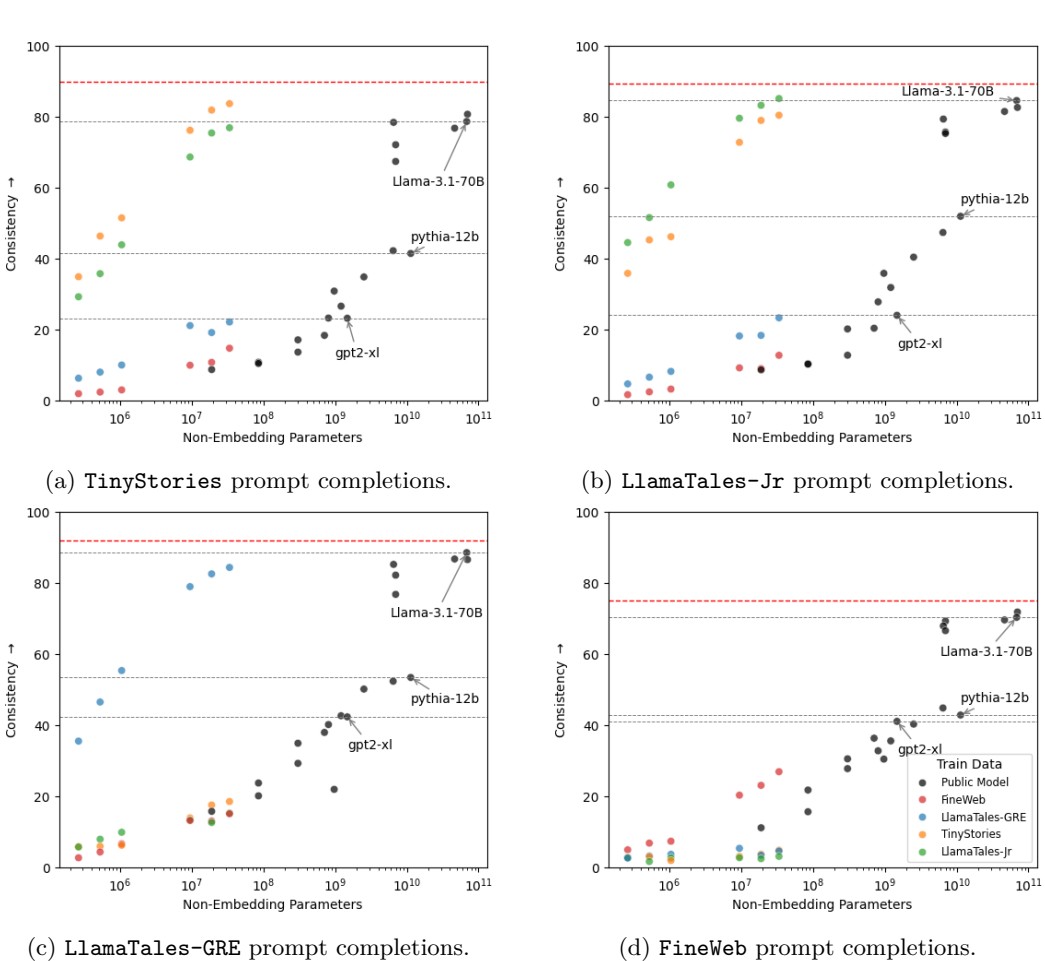

(a) `TinyStories` prompt completions.

(b) `LlamaTales-Jr` prompt completions.

(c) `LlamaTales-GRE` prompt completions.

(d) `FineWeb` prompt completions.

Figure A6: **Consistency** of text generated by LMs in Table A2, based on prompts from the data (test split) in Table 1. The legend colors represent the training data for each model. The red horizontal line marks the consistency of the train split of the dataset in focus.

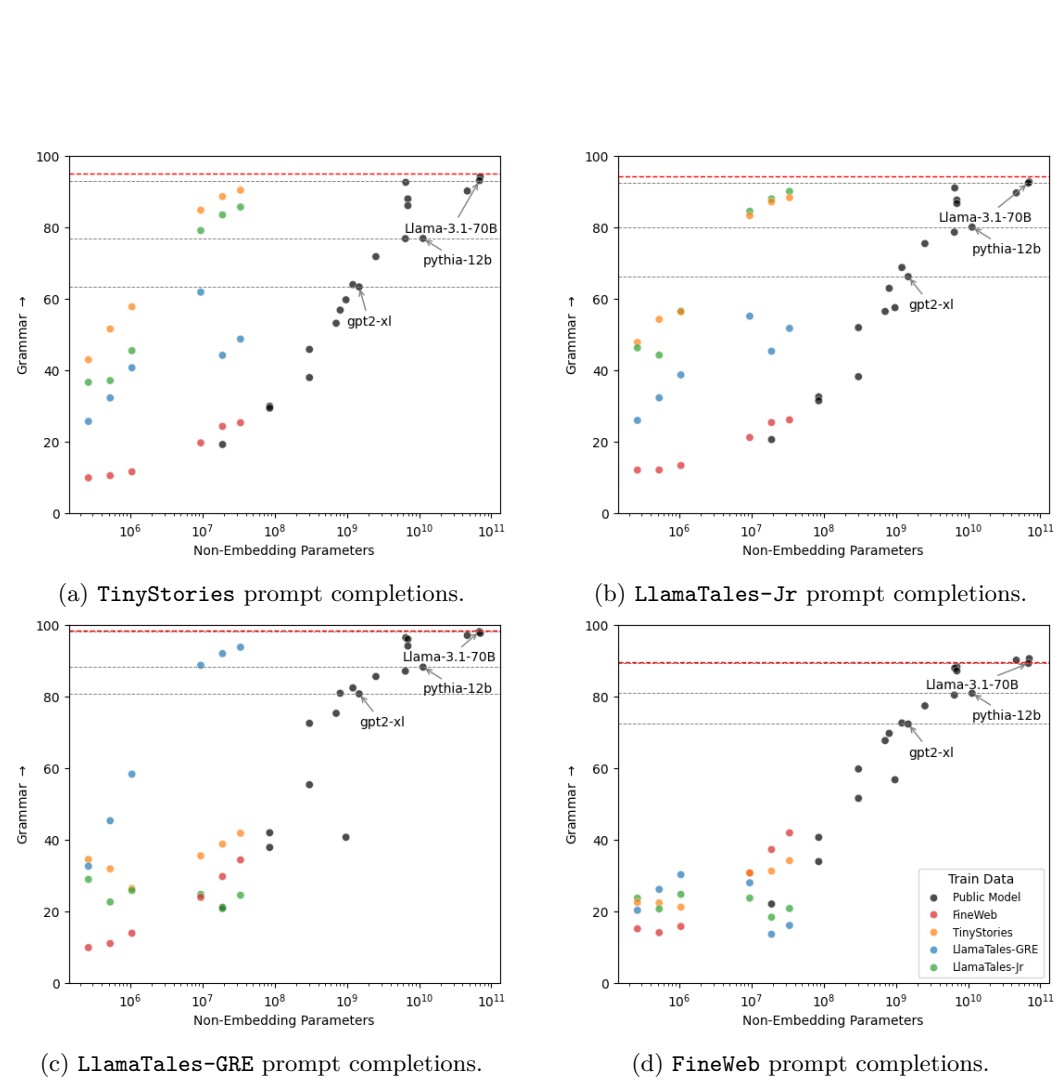

(a) `TinyStories` prompt completions.

(b) `LlamaTales-Jr` prompt completions.

(c) `LlamaTales-GRE` prompt completions.

(d) `FineWeb` prompt completions.

Figure A7: **Grammar** of text generated by LMs in Table A2, based on prompts from the data (test split) in Table 1. The legend colors represent the training data for each model. The red horizontal line marks the grammaticality of the train split of the dataset in focus.

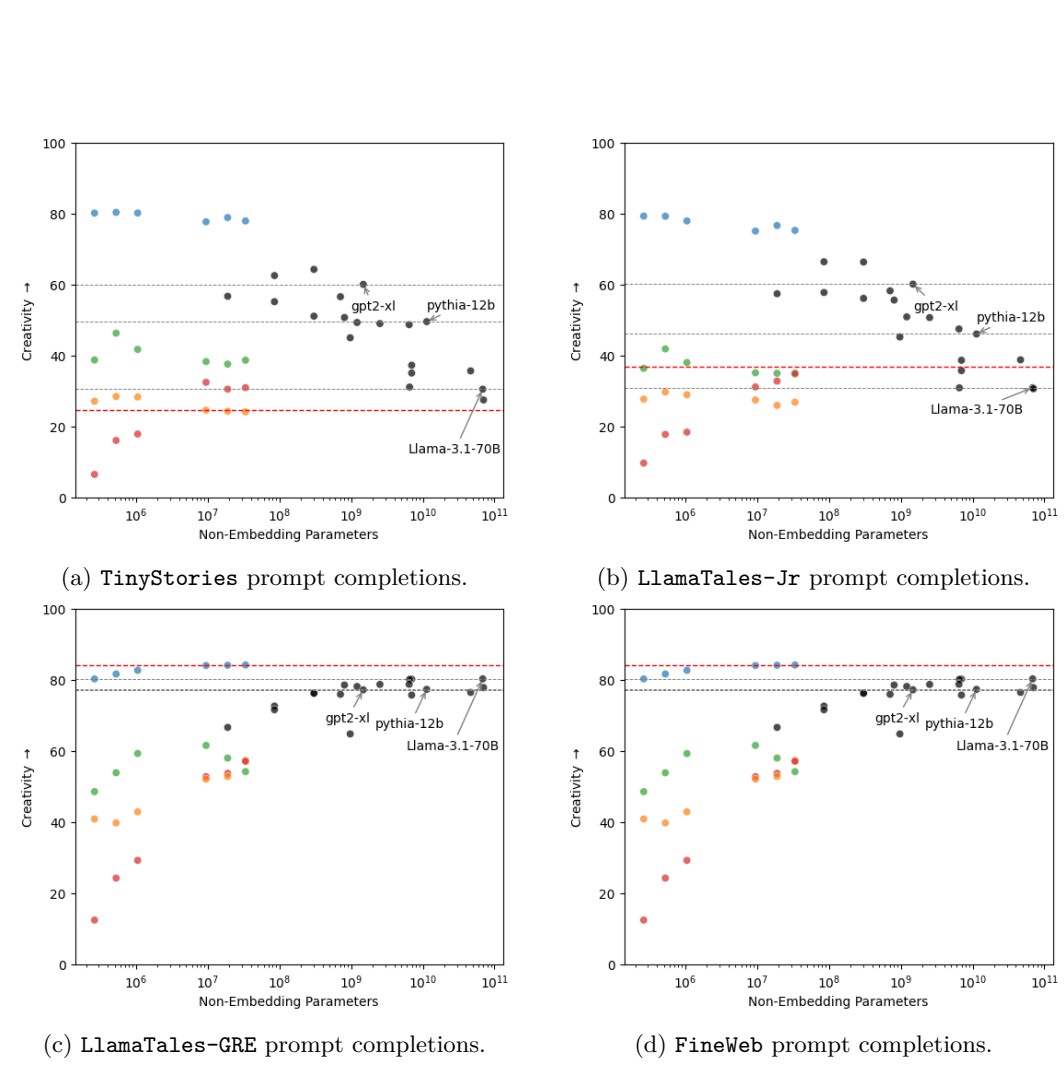

(a) `TinyStories` prompt completions.

(b) `LlamaTales-Jr` prompt completions.

(c) `LlamaTales-GRE` prompt completions.

(d) `FineWeb` prompt completions.

Figure A8: **Creativity** of text generated by LMs in Table A2, based on prompts from the data (test split) in Table 1. The legend colors represent the training data for each model. The red horizontal line marks the creativity of the train split of the dataset in focus.

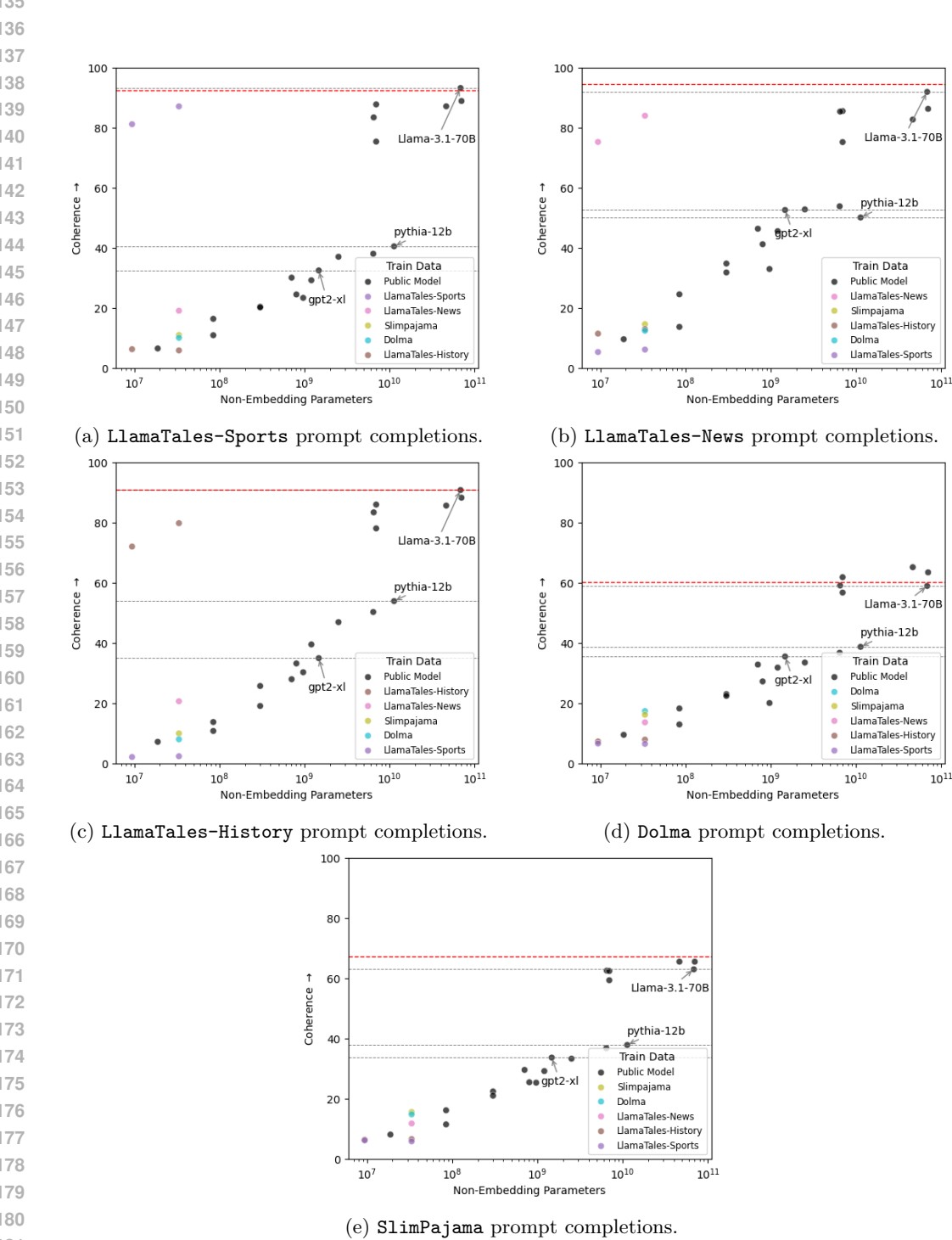

(a) `LlamaTales-Sports` prompt completions.

(b) `LlamaTales-News` prompt completions.

(c) `LlamaTales-History` prompt completions.

(d) `Dolma` prompt completions.

(e) `SlimPajama` prompt completions.

Figure A9: **Coherence** of text generated by LMs in Table A2, based on prompts from the data (test split) in Table 1. The legend colors represent the training data for each model. The red horizontal line marks the coherence of the train split of the dataset in focus.

**System Prompt:** You are an experienced teacher, skilled at identifying the readability of different texts.
**User:** Read the text below. Then, indicate the readability of the text, on a scale from 1 (extremely challenging to understand) to 100 (very easy to read and understand). In your assessment, consider factors such as sentence structure, vocabulary complexity, and overall clarity.
`<Text></Text>`
On a scale from 1 (extremely challenging to understand) to 100 (very easy to read and understand), how readable is this text?. Please answer with a single number.

Figure A10: Prompt used for LLM-as-a-Judge to evaluate readability. See Section 3.3.

**System Prompt:** You are an experienced teacher, skilled at identifying the coherence of different texts.
**User:** Read the text below. Then, indicate the coherence of the text, on a scale from 1 (extremely incoherent) to 100 (very coherent). Remember that coherent text should be well-structured and well-organized. Coherent text should not just be a heap of related information, but should build from sentence to sentence.
`<Text></Text>`
On a scale from 1 (extremely incoherent) to 100 (very coherent), how coherent is this text?. Please answer with a single number.

Figure A11: Prompt used for LLM-as-a-Judge to evaluate coherence. See Section 4

**System Prompt:** You are an experienced teacher, skilled at identifying the coherence of different texts.
**User:** Read the text below and evaluate the coherence of the text. Remember that coherent text should be well-structured and well-organized. Coherent text should not just be a heap of related information, but should build from sentence to sentence.
`<Text></Text>`
Please provide a short analysis of the text's coherence. After your analysis, on a scale from 1 (extremely incoherent) to 100 (very coherent), how coherent is this text? Please answer with a single number.

Figure A12: Prompt used for LLM-as-a-Judge to evaluate coherence, instructing the LM to generate an analysis before generating a score.

**System Prompt:** You are an experienced teacher, skilled at identifying the coherence of different texts.
**User:** First, consider the following examples:

Positive Example (Very Coherent):

> The process of photosynthesis is essential for plant life. It begins
> when sunlight is absorbed by chlorophyll in the leaves. This energy
> is then used to convert carbon dioxide and water into glucose and
> oxygen. The glucose provides energy for the plant, while the oxygen
> is released into the atmosphere.

This text is coherent because it is well-structured, with each sentence building on the previous one to explain a process clearly and logically.
Negative Example (Incoherent):

> Photosynthesis is a process. Leaves are green. Oxygen is in the air.
> Plants need water. Sunlight is bright.

This text is incoherent because it lacks logical flow and structure, presenting disjointed facts without clear connections or progression.

Now, read the text below and evaluate its coherence on a scale from 1 (extremely incoherent) to 100 (very coherent). Remember that coherent text should be well-structured and well-organized, not just a heap of related information.
`<Text></Text>`
Please provide a short analysis of the text's coherence. After your analysis, on a scale from 1 (extremely incoherent) to 100 (very coherent), how coherent is this text? Please answer with a single number.

Figure A13: Prompt used for LLM-as-a-Judge to evaluate coherence. This version includes positive and negative examples for reference.

**System Prompt:** You are an experienced teacher, skilled at identifying the readability of different texts.
**User:** Read the text below and evaluate the readability of the text. In your assessment, consider factors such as sentence structure, vocabulary complexity, and overall clarity.
`<Text></Text>`
Please provide a short analysis of the text's readability. After your analysis, on a scale from 1 (extremely challenging to understand) to 100 (very easy to read and understand), how readable is this text? Please answer with a single number.

Figure A14: Prompt used for LLM-as-a-Judge to evaluate readability, instructing the LM to generate an analysis before generating a score.

**System Prompt:** You are an experienced teacher, skilled at identifying the readability of different texts.
**User:** First, consider the following examples:

Positive Example (Very Readable):

> The cat sat on the mat. It was a sunny day, and the cat enjoyed the
> warmth. The mat was soft and comfortable, making it the perfect
> spot for a nap.

This text is easy to read because it uses simple sentence structures, familiar vocabulary, and conveys ideas clearly.
Negative Example (Challenging to Read):

> In the midst of the diurnal cycle, the feline quadruped positioned
> itself upon the textile floor covering, basking in the solar radiance,
> which permeated the atmosphere with thermal energy, rendering the
> environment conducive to somnolence.

This text is challenging to read due to complex sentence structures, advanced vocabulary, and convoluted expression of ideas.

Now, read the text below and evaluate its readability on a scale from 1 (extremely challenging to understand) to 100 (very easy to read and understand). In your assessment, consider factors such as sentence structure, vocabulary complexity, and overall clarity.
`<Text></Text>`
On a scale from 1 (extremely challenging to understand) to 100 (very easy to read and understand), how readable is this text?. Please answer with a single number.

Figure A15: Prompt used for LLM-as-a-Judge to evaluate readability. This version includes positive and negative examples for reference.

**System Prompt:** You are an experienced teacher, skilled at identifying grammatical errors of different texts.
**User:** Read the text below. Then, indicate the grammaticality of the text on a scale from 1 (extremely ungrammatical) to 100 (perfectly grammatical). In your assessment, consider factors such as spelling, part of speech, sentence structure, punctuation, and overall grammatical correctness.
`<Text></Text>`
On a scale from 1 (extremely ungrammatical) to 100 (perfectly grammatical), how grammatical is this text?. Please answer with a single number.

Figure A16: Prompt used for LLM-as-a-Judge to evaluate grammaticality.

**System Prompt:** You are an experienced linguist, skilled at evaluating the fluency of different texts.
**User:** Read the text below. Then, indicate the fluency of the text, on a scale from 1 (poor fluency) to 100 (excellent fluency). In your assessment, consider factors such as grammatical correctness, naturalness of language, and overall smoothness.
`<Text></Text>`
On a scale from 1 (poor fluency) to 100 (excellent fluency), how fluent is this text?. Please answer with a single number.

Figure A17: Prompt used for LLM-as-a-Judge to evaluate fluency.

**System Prompt:** You are an experienced teacher, skilled at identifying the consistency of different texts.
**User:** Read the text below. Then, evaluate how consistent the first two sentences are with the rest of the text, on a scale from 1 (extremely inconsistent) to 100 (very consistent). Consistent text should maintain a logical flow and alignment in terms of theme, tone, and information throughout.
`<Text></Text>`
On a scale from 1 (extremely inconsistent) to 100 (very consistent), how consistent are the first two sentences of this text with the rest of the text? Please answer with a single number.

Figure A18: Prompt used for LLM-as-a-Judge to evaluate consistency.

**System Prompt:** You are an experienced teacher, skilled at identifying the creativity of different texts.
**User:** Read the text below. Then, indicate the creativity of the text, on a scale from 1 (not creative at all) to 100 (extremely creative). In your assessment, consider factors such as originality, imagination, and uniqueness.
`<Text></Text>`
On a scale from 1 (not creative at all) to 100 (extremely creative), how creative is this text? Please answer with a single number.

Figure A19: Prompt used for LLM-as-a-Judge to evaluate creativity.

**System Prompt:** You are an experienced teacher, skilled at identifying the clarity of different texts.
**User:** Read the text below. Then, indicate the clarity of the text, on a scale from 1 (not clear at all) to 100 (extremely clear). In your assessment, consider factors such as coherence, conciseness, and comprehensibility.
`<Text></Text>`
On a scale from 1 (not clear at all) to 100 (extremely clear), how clear is this text? Please answer with a single number.

Figure A20: Prompt used for LLM-as-a-Judge to evaluate clarity.

> **System Prompt:** You are a celebrated children's author. You write stories that are both easy to read and grammatically correct.
> **User:** Write a short story (3-5 paragraphs) which only uses simple words that a 5 year old child would understand. The story should use the words: `<WORD-1>`, `<WORD-2>`, and `<WORD-3>`. The story has the following features: `<FEAT-1>` ... `<FEAT-K>`

Figure A21: Prompt used to generate `LlamaTales-Jr`. See Section 5.

> **System Prompt:** You are a renowned fiction writer, celebrated for your imaginative storytelling and compelling characters. Your work spans various genres, including fantasy, science fiction, and contemporary fiction, and is known for its vivid descriptions, intricate plots, and emotional depth. Your writing is best appreciated by readers with the vocabulary and comprehension expected of a college graduate.
> **User:** Write a short story (3-5 paragraphs). The story should use the words: `<WORD-1>`, `<WORD-2>`, and `<WORD-3>`. The story has the following features: `<FEAT-1>` ... `<FEAT-K>`

Figure A22: Prompt used to generate `LlamaTales-GRE`. See Section 5.

> **System Prompt:** You are a distinguished historian, celebrated for your meticulous research and engaging narratives. Your work spans various historical periods and is known for its depth, accuracy, and insightful analysis. You have a talent for bringing history to life, making complex events and figures accessible and compelling to a broad audience. Your writing is best appreciated by readers with a keen interest in history and a desire to understand the past in a nuanced and comprehensive manner.
> **User:** Write a short historical article (3-5 paragraphs) that provides an insightful analysis of a significant event or figure. Include key details, context, and the impact on subsequent history. The story should use the words: `<WORD-1>`, `<WORD-2>`, and `<WORD-3>`. The story has the following features: `<FEAT-1>` ... `<FEAT-K>`

Figure A23: Prompt used to generate `LlamaTales-History`.

**System Prompt:** You are an experienced sports journalist known for your vivid and engaging coverage of athletic events and athletes' stories. Your writing captures the excitement, drama, and human elements of sports, appealing to both die-hard fans and casual readers. You have a keen eye for detail, a deep understanding of various sports, and the ability to convey complex strategies and statistics in an accessible manner. Your articles are characterized by their dynamic prose, insightful analysis, and ability to place sporting events within broader cultural and social contexts.
**User:** Write a short sports article (3-5 paragraphs) about a recent game, match, or athletic performance. Include vivid descriptions, key statistics, and quotes from players or coaches if applicable. The story should use the words: `<WORD-1>`, `<WORD-2>`, and `<WORD-3>`. The story has the following features: `<FEAT-1>` ... `<FEAT-K>`

Figure A24: Prompt used to generate `LlamaTales-Sports`.

**System Prompt:** You are a seasoned journalist at The New York Times, known for your incisive reporting and compelling storytelling. Your work covers a wide range of topics, from breaking news and investigative journalism to in-depth features and opinion pieces. You have a keen eye for detail, a commitment to accuracy, and the ability to convey complex issues in a clear and engaging manner. Your writing is characterized by its clarity, depth, and ability to inform and engage a diverse readership.
**User:** Write a concise news article (3-5 paragraphs) about a recent significant event. Include key details, quotes from relevant sources, and the broader context of the event. The story should use the words: `<WORD-1>`, `<WORD-2>`, and `<WORD-3>`. The story has the following features: `<FEAT-1>` ... `<FEAT-K>`

Figure A25: Prompt used to generate `LlamaTales-News`.

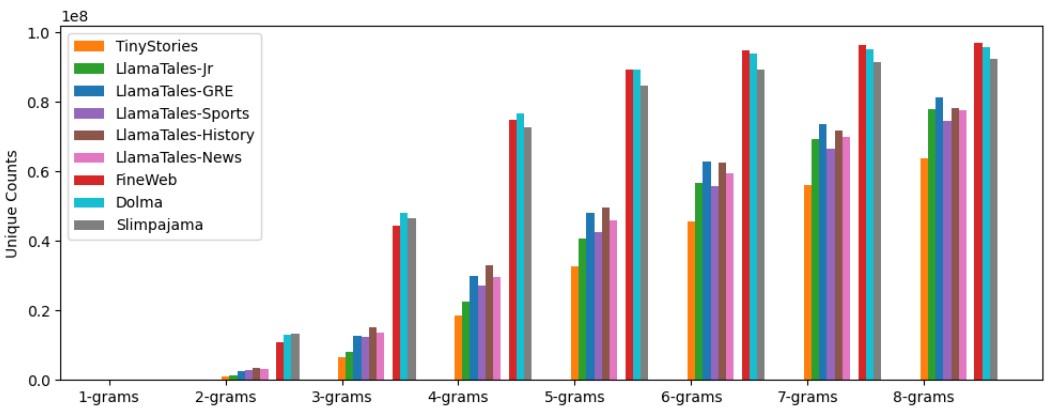

Figure A26: Unique $n$-gram counts for 100M-token samples from each dataset. Easy-to-learn datasets (`TinyStories` and `LlamaTales` series) exhibit substantially lower $n$-gram diversity than hard-to-learn datasets (`FineWeb`, `Dolma`, `SlimPajama`) for small values of $n$. Note that as $n$ increases, the number of possible $n$-grams grows exponentially. Consequently, the likelihood of any given $n$-gram being unique becomes very high for large values of $n$. Refer to Figures A27 and A28 for close-ups of unique 1-grams and 2-grams, respectively. Return to Section 7 (Discussion).

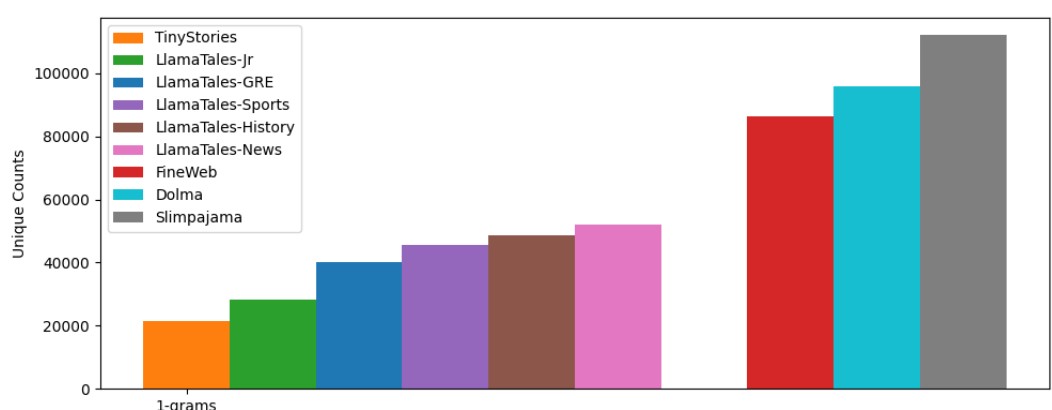

Figure A27: Unique unigram counts for 100M-token samples from each dataset. Refer to Figure A26 for higher values of $n$.

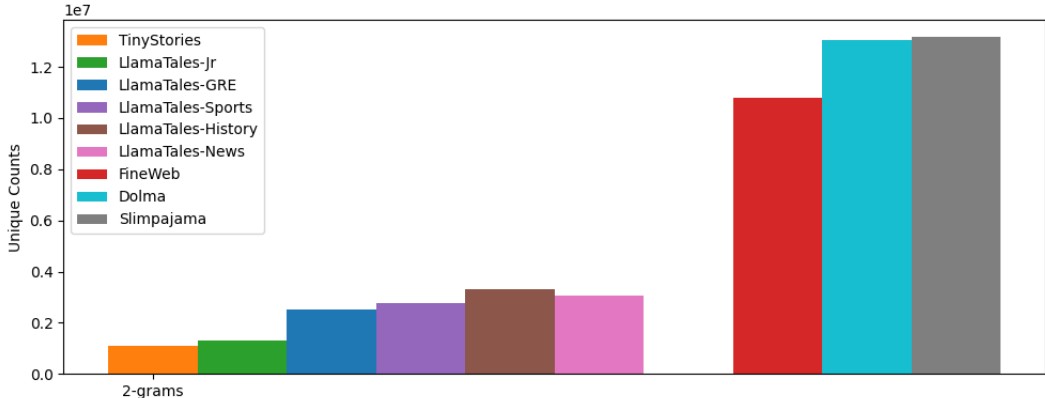

Figure A28: Unique bigram counts for 100M-token samples from each dataset. Refer to Figure A26 for higher values of $n$.

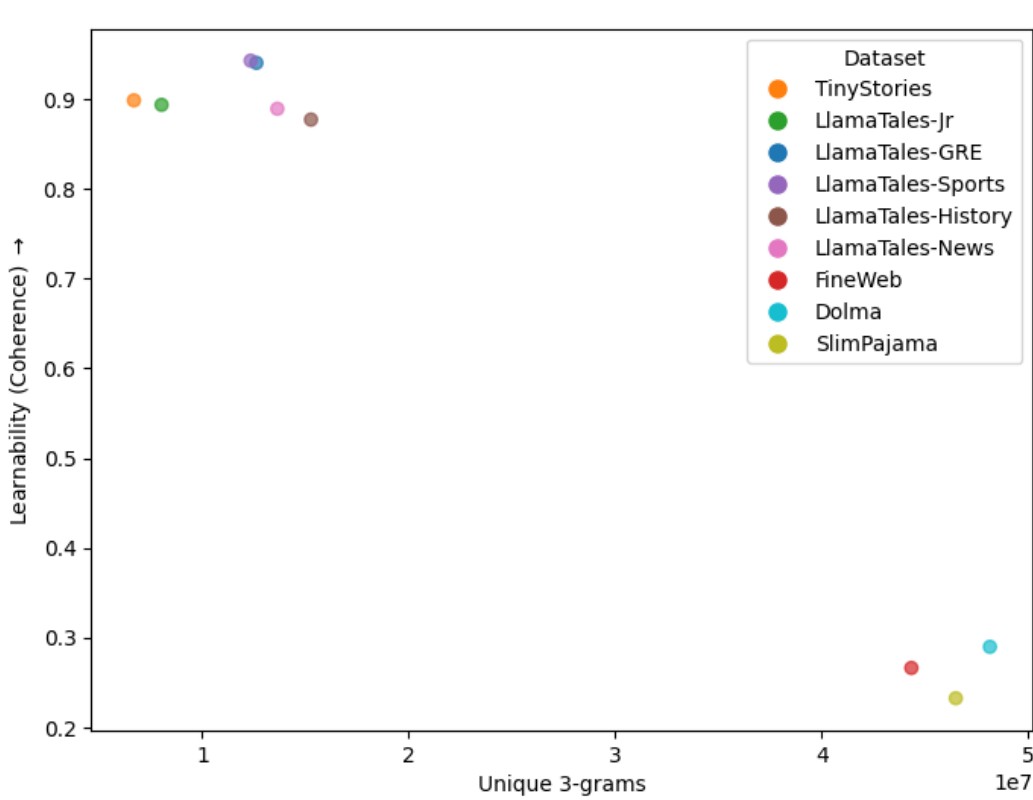

Figure A29: Learnability (coherence) is calculated by dividing the coherence of text generated by a 33M parameter SLM by the coherence of the training data used for that SLM. A score of 1.0 means the SLM produces text with the same level of coherence as the documents in its training data. We find that datasets with lower $n$-gram diversity are significantly easier for SLMs to learn (upper left) compared to datasets with higher diversity (bottom right).

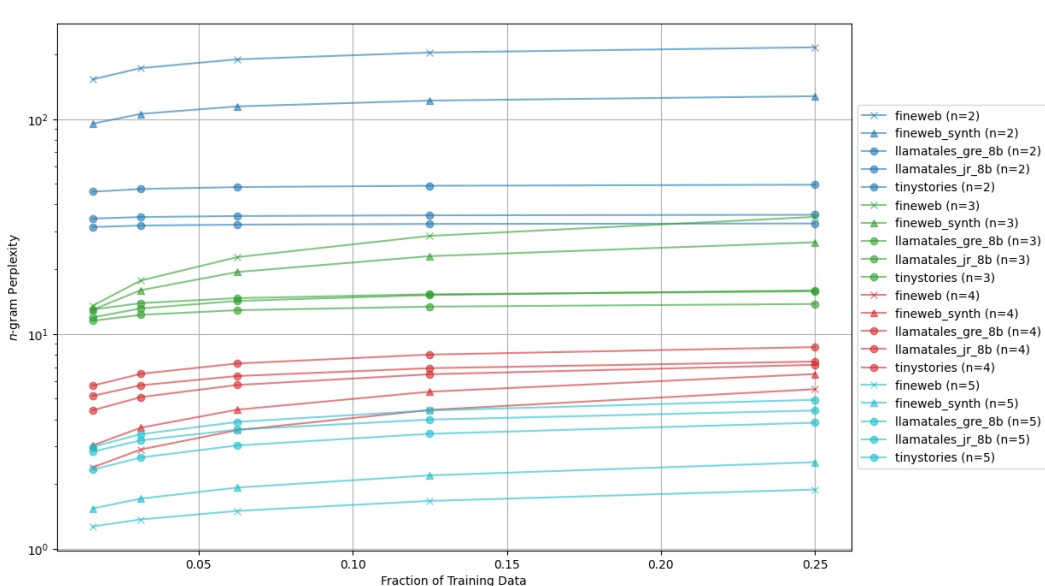

Figure A30: We train $n$-gram LMs on our datasets and use them to compute the perplexity of their own training data. For $n \leq 2$, easy-to-learn datasets (`TinyStories`, `LlamaTales-Jr`, `LlamaTales-GRE`) exhibit low perplexity, while hard-to-learn datasets (`FineWeb`, `FineWeb-Synth`) show high perplexity. Interestingly, this relationship reverses for $n \geq 3$. Although not definitive, it is intriguing to consider that $n$-gram LM perplexity could potentially serve as a fingerprint for distinguishing between easy-to-learn and difficult-to-learn data. Return to Section 7 (Discussion).

Table A2: Overview of the models used in our experiments. Example generations from each model are shown in Tables A6 to A20. **Top:** Models trained from scratch on the dataset indicated by each model's prefix. **Bottom:** Pretrained models sourced from Huggingface. Return to Section 6.

| Model | Parameters | Train Data | Train Tokens | Layers | Heads | Model Dim |
|-------|-----------|-----------|-------------|--------|-------|-----------|
| tinystories-262K | 2.63e+05 | TinyStories | 1e10 | 1 | 2 | 128 |
| tinystories-524K | 5.25e+05 | TinyStories | 1e10 | 2 | 2 | 128 |
| tinystories-1M | 1.05e+06 | TinyStories | 1e10 | 4 | 2 | 128 |
| tinystories-9M | 9.44e+06 | TinyStories | 1e10 | 4 | 6 | 384 |
| tinystories-18M | 1.89e+07 | TinyStories | 1e10 | 8 | 6 | 384 |
| tinystories-33M | 3.36e+07 | TinyStories | 1e10 | 8 | 8 | 512 |
| llamatales_jr-262K | 2.63e+05 | LlamaTales-Jr | 1e10 | 1 | 2 | 128 |
| llamatales_jr-524K | 5.25e+05 | LlamaTales-Jr | 1e10 | 2 | 2 | 128 |
| llamatales_jr-1M | 1.05e+06 | LlamaTales-Jr | 1e10 | 4 | 2 | 128 |
| llamatales_jr-9M | 9.44e+06 | LlamaTales-Jr | 1e10 | 4 | 6 | 384 |
| llamatales_jr-18M | 1.89e+07 | LlamaTales-Jr | 1e10 | 8 | 6 | 384 |
| llamatales_jr-33M | 3.36e+07 | LlamaTales-Jr | 1e10 | 8 | 8 | 512 |
| llamatales_gre-262K | 2.63e+05 | LlamaTales-GRE | 1e10 | 1 | 2 | 128 |
| llamatales_gre-524K | 5.25e+05 | LlamaTales-GRE | 1e10 | 2 | 2 | 128 |
| llamatales_gre-1M | 1.05e+06 | LlamaTales-GRE | 1e10 | 4 | 2 | 128 |
| llamatales_gre-9M | 9.44e+06 | LlamaTales-GRE | 1e10 | 4 | 6 | 384 |
| llamatales_gre-18M | 1.89e+07 | LlamaTales-GRE | 1e10 | 8 | 6 | 384 |
| llamatales_gre-33M | 3.36e+07 | LlamaTales-GRE | 1e10 | 8 | 8 | 512 |
| fineweb-262K | 2.63e+05 | FineWeb | 1e10 | 1 | 2 | 128 |
| fineweb-524K | 5.25e+05 | FineWeb | 1e10 | 2 | 2 | 128 |
| fineweb-1M | 1.05e+06 | FineWeb | 1e10 | 4 | 2 | 128 |
| fineweb-9M | 9.44e+06 | FineWeb | 1e10 | 4 | 6 | 384 |
| fineweb-18M | 1.89e+07 | FineWeb | 1e10 | 8 | 6 | 384 |
| fineweb-33M | 3.36e+07 | FineWeb | 1e10 | 8 | 8 | 512 |
| gpt2 | 8.51e+07 | | | 12 | 12 | 768 |
| gpt2-medium | 3.02e+08 | | | 24 | 16 | 1024 |
| gpt2-large | 7.08e+08 | | | 36 | 20 | 1280 |
| gpt2-xl | 1.48e+09 | | | 48 | 25 | 1600 |
| pythia-70m | 1.89e+07 | The Pile | 3e11 | 6 | 8 | 512 |
| pythia-160m | 8.51e+07 | The Pile | 3e11 | 12 | 12 | 768 |
| pythia-410m | 3.02e+08 | The Pile | 3e11 | 24 | 16 | 1024 |
| pythia-1b | 8.06e+08 | The Pile | 3e11 | 16 | 8 | 2048 |
| pythia-1.4b | 1.21e+09 | The Pile | 3e11 | 24 | 16 | 2048 |
| pythia-2.8b | 2.52e+09 | The Pile | 3e11 | 32 | 32 | 2560 |
| pythia-6.9b | 6.44e+09 | The Pile | 3e11 | 32 | 32 | 4096 |
| pythia-12b | 1.13e+10 | The Pile | 3e11 | 36 | 40 | 5120 |
| TinyLlama_v1.1 | 9.69e+08 | SlimPajama | 2e12 | 22 | 32 | 2048 |
| Mistral-7B-v0.3 | 6.98e+09 | | | 32 | 32 | 4096 |
| Mixtral-8x7B-v0.1 | 4.64e+10 | | | 32 | 32 | 4096 |
| Qwen2-7B | 6.53e+09 | | 7e12 | 28 | 28 | 3584 |
| Qwen2-72B | 7.02e+10 | | 7e12 | 80 | 64 | 8192 |
| Llama-3.1-8B | 6.98e+09 | | 15e12 | 32 | 32 | 4096 |
| Llama-3.1-70B | 6.85e+10 | | 15e12 | 80 | 64 | 8192 |

Table A3: Statistics for the train splits of the datasets described in Section 5 as well as (absolute) Pearson **corr**elation coefficients against human judgments of readability (↑ is better). Random examples from each dataset are shown in Table A5. **Top:** Classic readability formulas (↓ is easier to read). **Mid-Top:** Statistics computed from running a constiuency parser over the sentences of each dataset (↑ suggests more grammatical complexity). **Mid-Bot:** The result of prompting `Llama-3.1-70B-Instruct` to judge readability and coherence (↑ is better). Perplexity is computed with `Llama-3.1-8B`, `Qwen2-7B`, and `Mistral-7B-v0.3` and averaged. **Bot:** Word, token, and syllable level statistics. Truncated statistics are shown in Table 1.

| | Corr. | TinyStories | LlamaTales-Jr | LlamaTales-GRE | FineWeb |
|---|---|---|---|---|---|
| **Automated Readability** | 0.47 | 2.9 | 2.9 | 12.4 | 13.1 |
| **Coleman–Liau** | 0.48 | 3.7 | 3.8 | 10.4 | 11.8 |
| **Dale–Chall** | 0.58 | 5.7 | 5.7 | 9.1 | 9.3 |
| **Flesch–Kincaid** | 0.49 | 2.4 | 2.2 | 9.6 | 10.7 |
| **Gunning Fog** | 0.50 | 4.6 | 3.8 | 11.7 | 12.1 |
| **Linsear Write** | 0.41 | 4.2 | 3.3 | 13.2 | 12.7 |
| **SMOG** | 0.53 | 5.7 | 5.4 | 11.3 | 12.6 |
| **Spache Readability** | 0.51 | 2.7 | 2.5 | 5.5 | 5.5 |
| **Depth / Sentence** | 0.34 | 6.8 | 6.4 | 10.6 | 9.5 |
| **Width / Sentence** | 0.34 | 5.1 | 4.7 | 8.0 | 7.5 |
| **Nodes / Sentence** | 0.36 | 19.6 | 17.2 | 42.1 | 37.8 |
| **Readability** | **0.74** | 92.6 | 92.7 | 64.8 | 68.2 |
| **Coherence** | 0.03 | 90.1 | 89.5 | 94.4 | 77.4 |
| **Perplexity** | 0.30 | 3.9 | 5.2 | 5.5 | 9.3 |
| **Tokens / Document** | 0.15 | 186.5 | 282.8 | 500.6 | 497.3 |
| **Syllables / Document** | 0.43 | 180.2 | 270.7 | 561.6 | 603.1 |
| **Words / Document** | 0.10 | 152.9 | 222.6 | 392.2 | 386.4 |
| **Unique Words** | | 6.4e4 | 1.5e5 | 3.2e5 | 5.1e6 |
| **Unique 1-grams (token)** | | 3.20e4 | 4.35e4 | 5.21e4 | 1.09e5 |
| **Unique 2-grams** | | 2.89e6 | 3.71e6 | 7.90e6 | 4.26e7 |
| **Unique 4-grams** | | 8.75e7 | 1.10e8 | 1.61e8 | 5.74e8 |
| **Unique 8-grams** | | 5.02e8 | 6.48e8 | 6.88e8 | 9.43e8 |
| **Total Documents** | | 4.9e6 | 3.6e6 | 2.0e6 | 2.0e6 |
| **Total Tokens** | | 9.2e8 | 1e9 | 1e9 | 1e9 |
| **Synthetic** | | Yes | Yes | Yes | No |
| **Source** | | GPT-3.5/4 | Llama-3.1-8B | Llama-3.1-8B | Web |

Table A4: Our ranking of open-source LMs. See Section 4.2.

| Model | Parameters | Rank |
|---|---|---|
| pythia-70m | 1.89e+07 | 11 |
| pythia-160m | 8.51e+07 | 10 |
| pythia-410m | 3.02e+08 | 9 |
| pythia-1b | 8.06e+08 | 8 |
| pythia-1.4b | 1.21e+09 | 7 |
| pythia-2.8b | 2.52e+09 | 6 |
| pythia-6.9b | 6.44e+09 | 5 |
| pythia-12b | 1.13e+10 | 4 |
| Mistral-7B-v0.3 | 6.98e+09 | 3 |
| Qwen2-7B | 6.53e+09 | 3 |
| Llama-3.1-8B | 6.98e+09 | 3 |
| Mixtral-8x7B-v0.1 | 4.64e+10 | 2 |
| Qwen2-72B | 7.02e+10 | 1 |
| Llama-3.1-70B | 6.85e+10 | 1 |

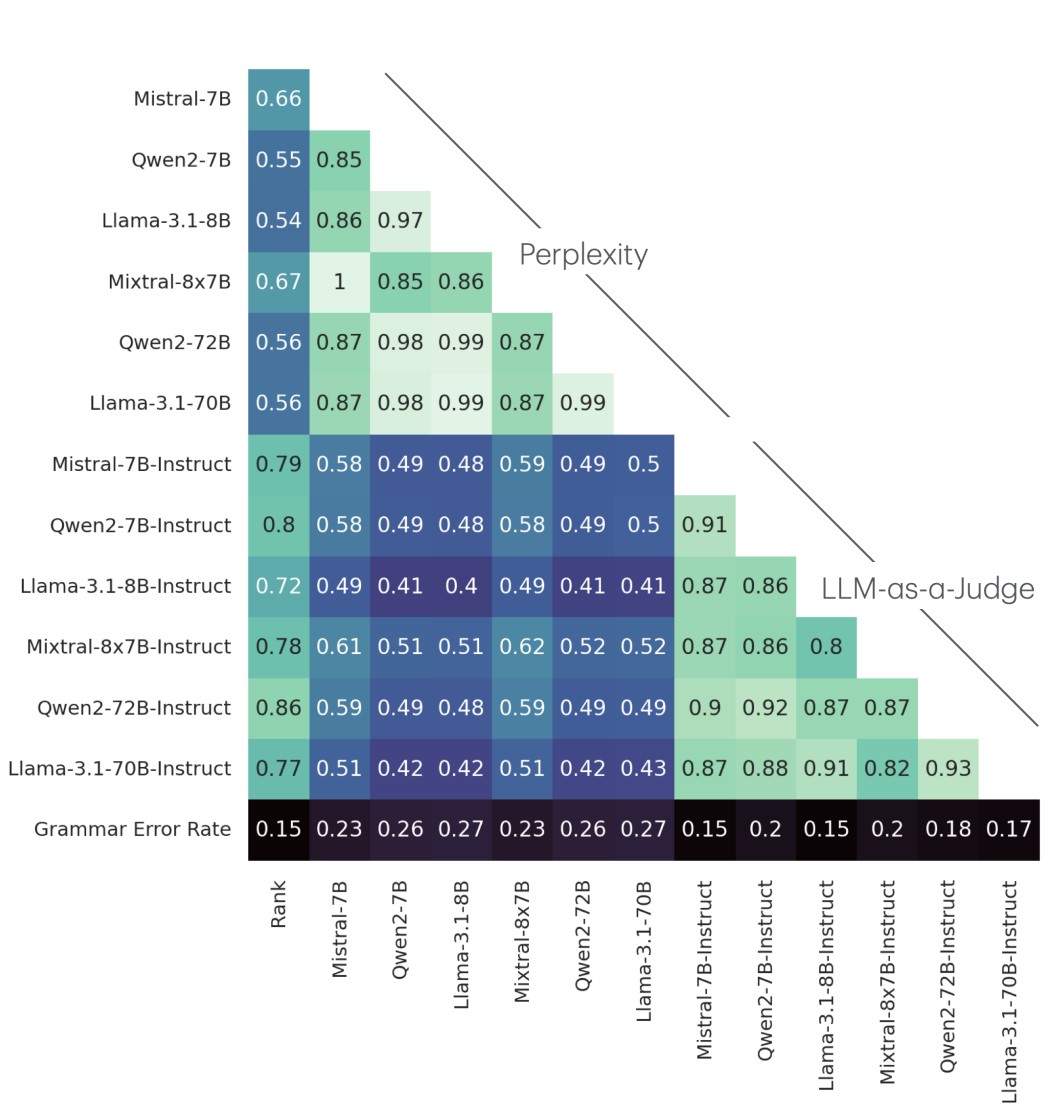

Figure A31: Instructing both 7B and 70B parameter LLMs to evaluate text coherence shows a strong correlation with the model rankings described in Section 4.2. This is followed in correlation strength by using external LMs to compute perplexity.

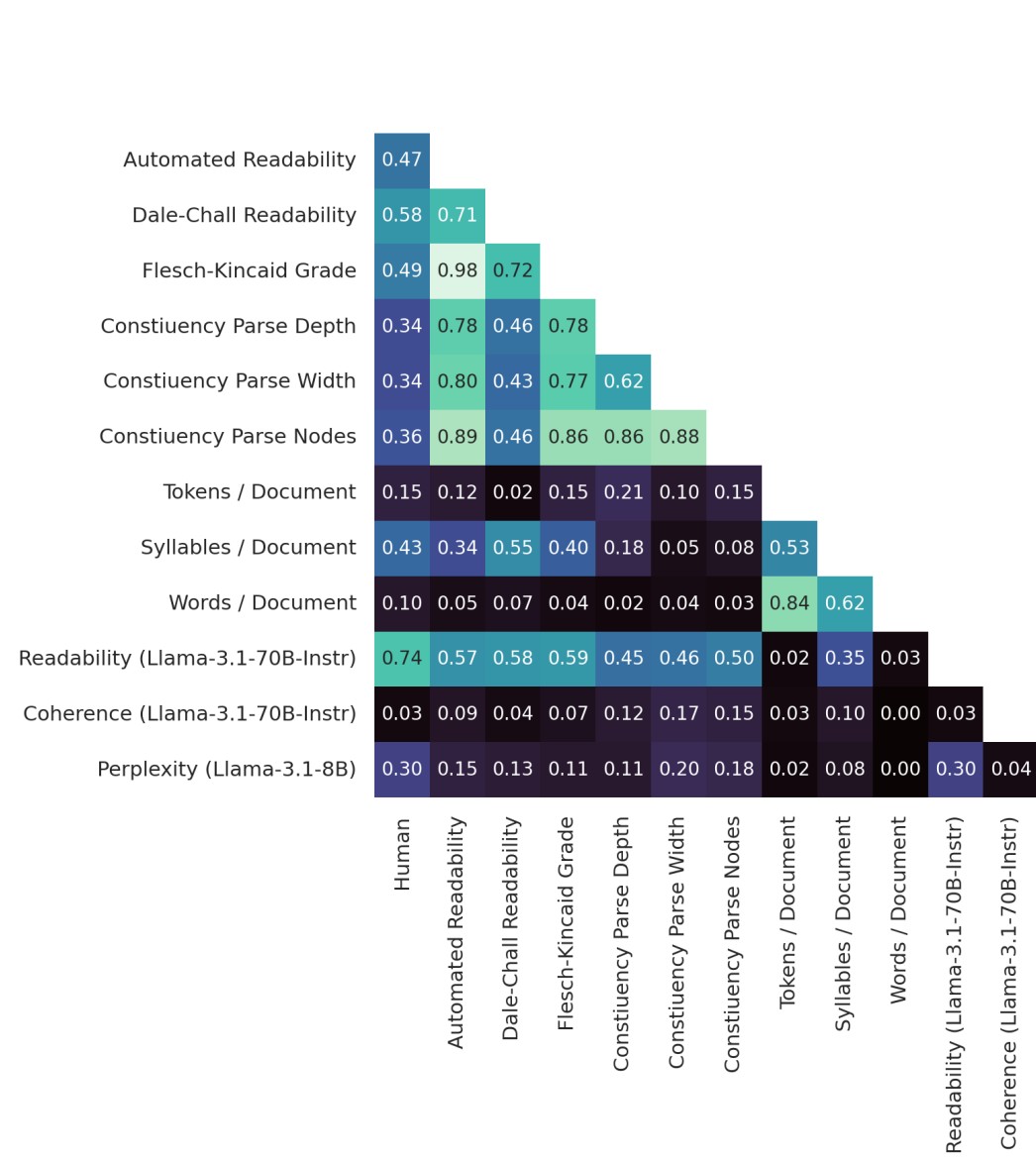

Figure A32: Pearson correlation coefficients among various measures on the `CLEAR` dataset. Instructing an LLM to judge readability shows the highest correlation with human judgments of readability. However, instructing an LLM to judge coherence does not correlate with readability, and perplexity only shows a weak correlation with readability. These findings suggest that there is a distinction between *readability* and *text quality*, and that these differences can be identified through automatic methods. See Sections 3.4 and 4.1 for details.

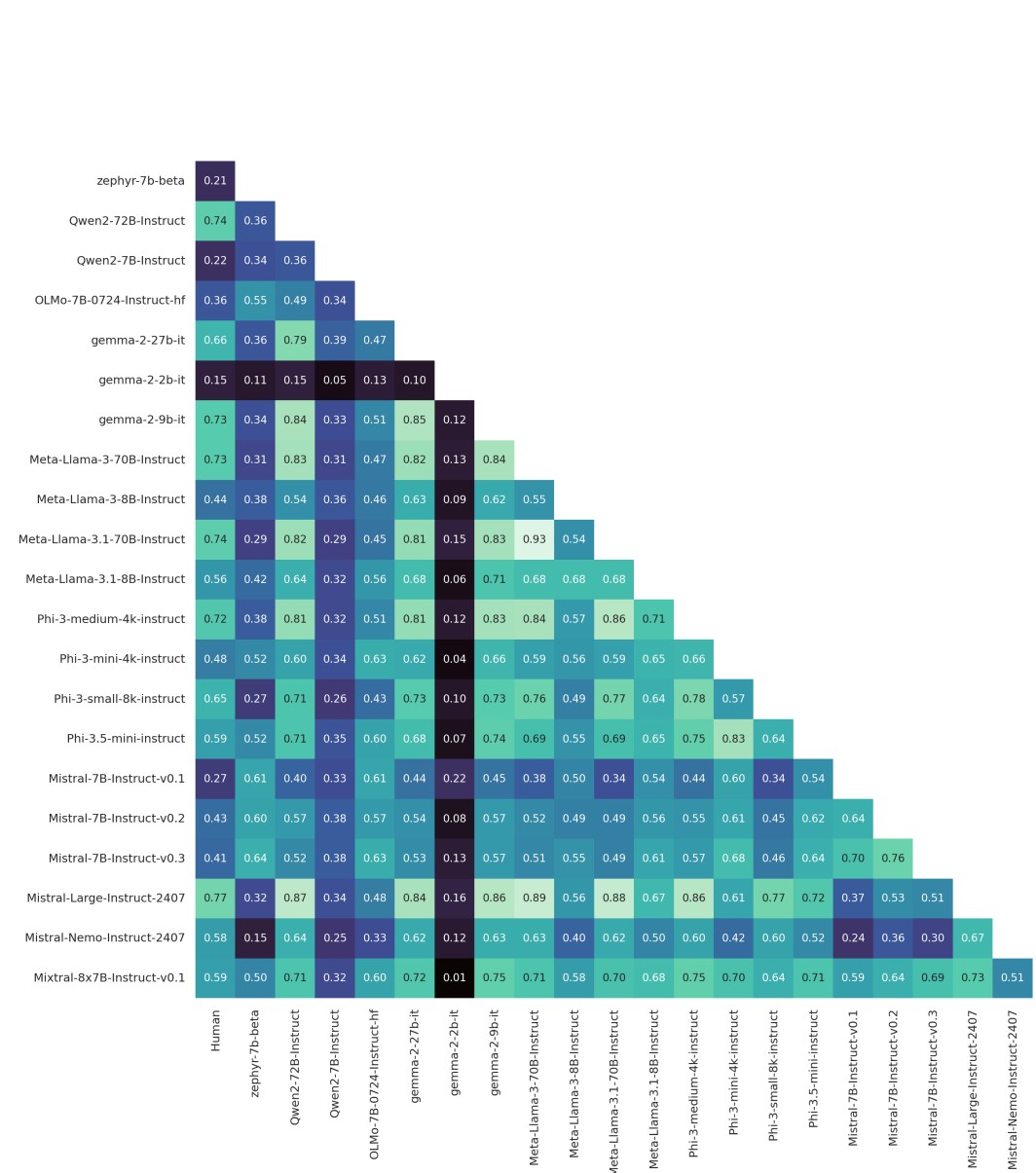

Figure A33: Pearson correlation coefficients for various instruction-tuned language models tasked with judging the readability of the `CLEAR` dataset. We observe that larger models tend to have a stronger correlation with human judgments of readability. Notably, the largest model, `Mistral-Large-Instruct-2407`, which has 123B parameters, exhibited the highest correlation. The smallest model with a coefficient greater than 0.70 was `gemma-2-9b-it`, which has 9.24B parameters. For more details, refer to Section 3.4.

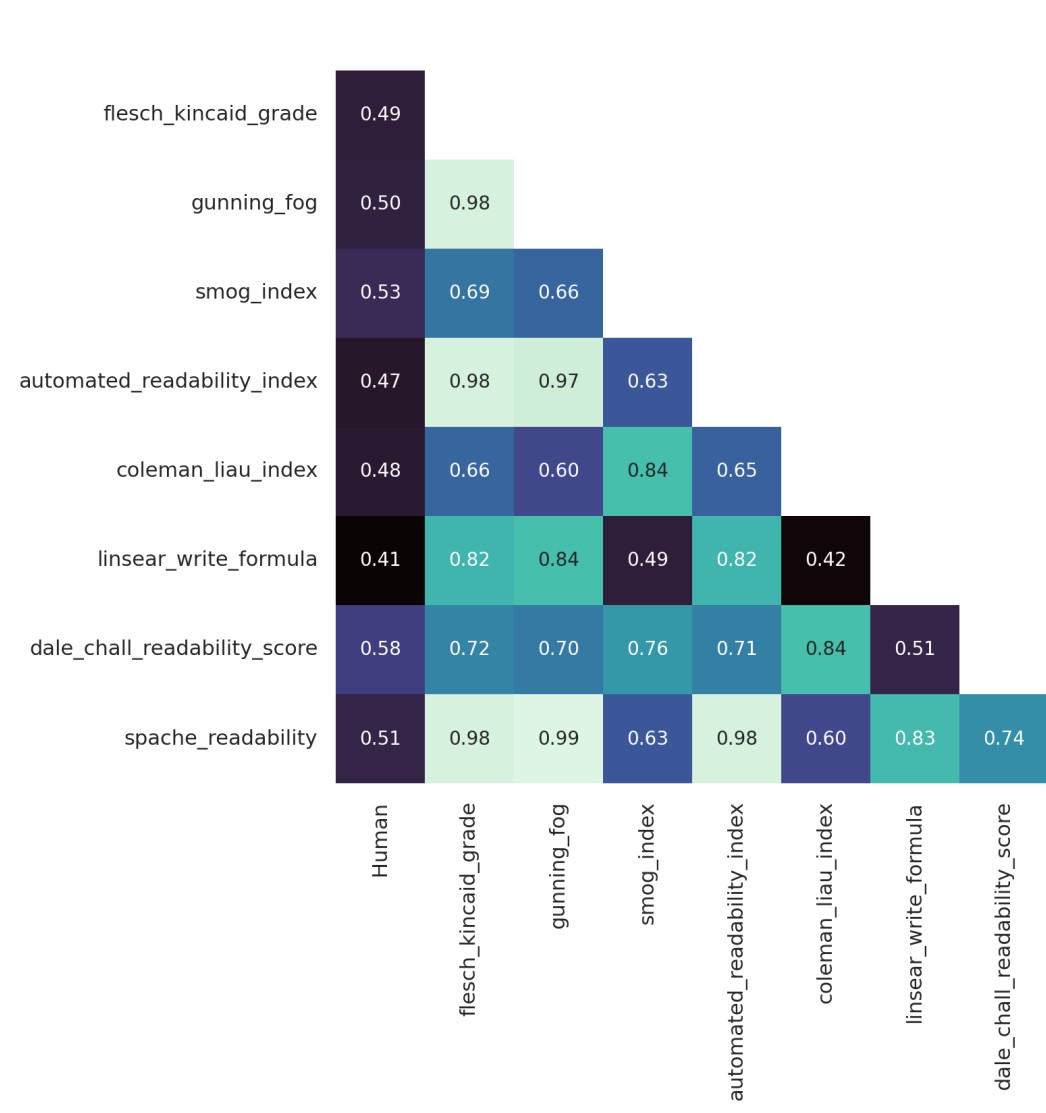

Figure A34: Pearson correlation coefficients for various classic readability formulas applied to the CLEAR dataset. The results indicate that all formulas show a similar correlation with human judgments of readability. For further details, see Section 3.4.

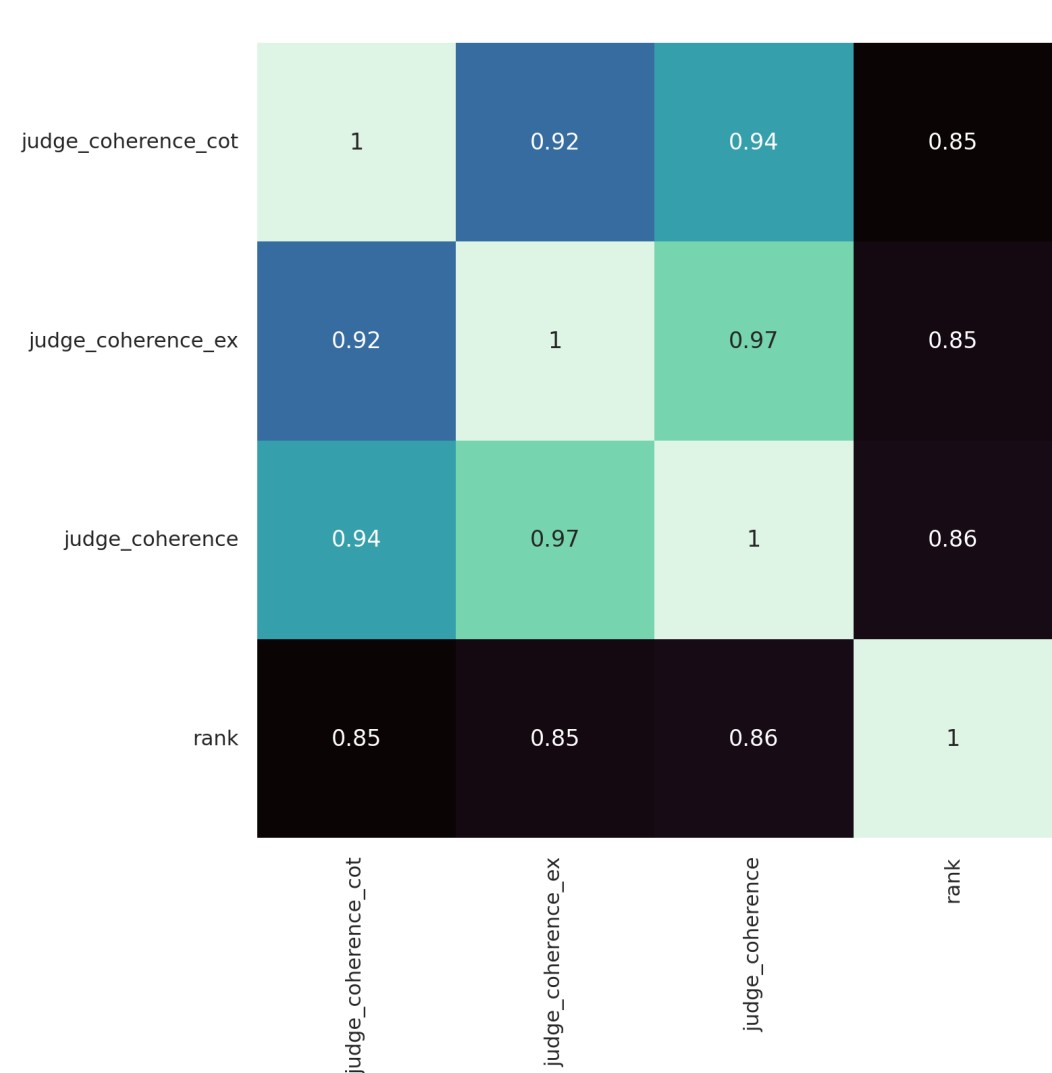

Figure A35: Pearson correlation coefficients for different prompt choices for LLM-as-a-Judge to measure coherence are presented. Scores are computed for generations conditioned on prompts from LlamaTales-GRE. judge_coherence uses the prompt shown in Figure A11. judge_coherence_cot uses the prompt shown in Figure A12. judge_coherence_ex uses the prompt shown in Figure A13. We find no meaningful differences between the prompt variants.

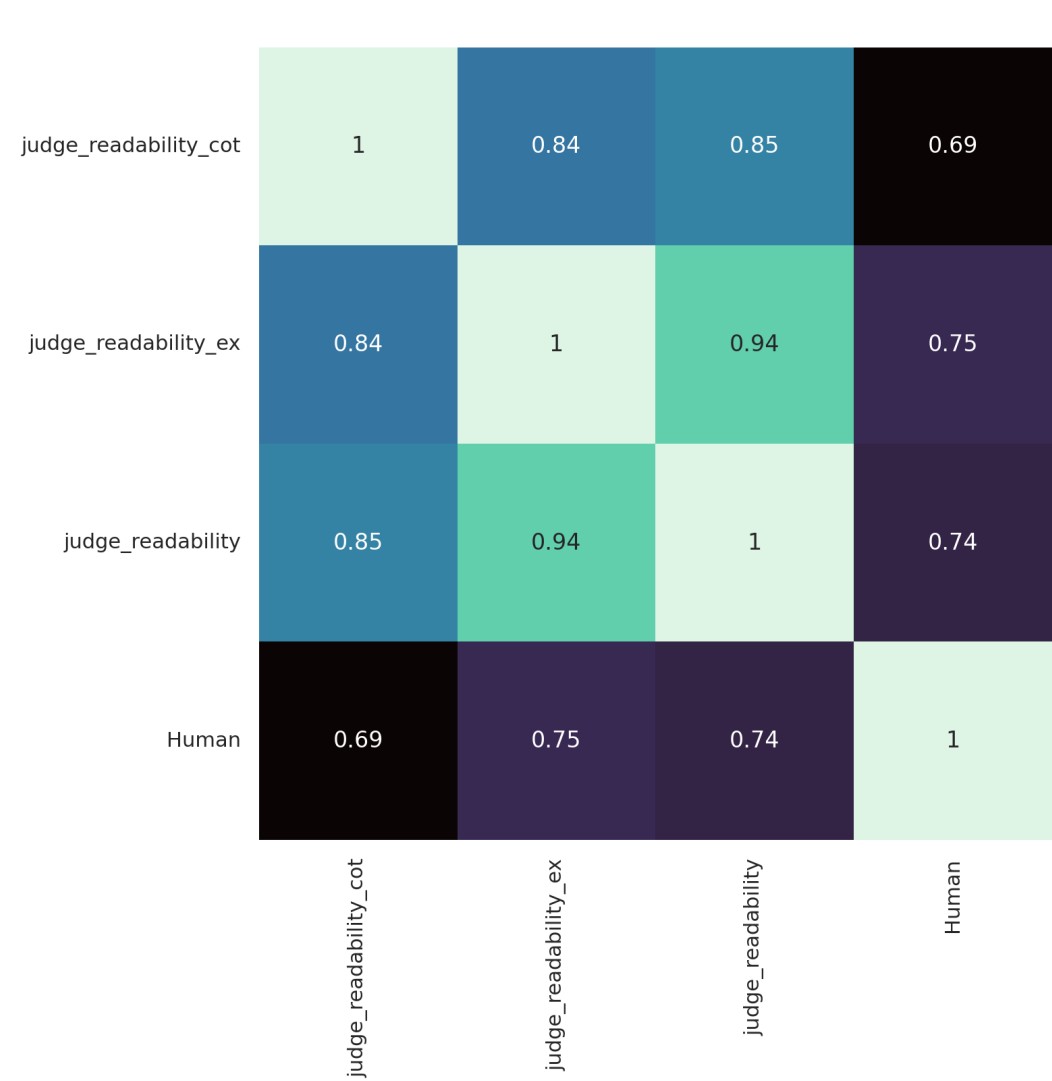

Figure A36: Pearson correlation coefficients for different prompt choices for LLM-as-a-Judge to measure readability are presented. Scores are computed over the CLEARdataset (Section 3.4). judge_readability uses the prompt shown in Figure A10. judge_readability_cot uses the prompt shown in Figure A14. judge_readability_ex uses the prompt shown in Figure A15. All variants are strongly correlated with human experts, but judge_readability_cot exhibits the lowest correlation.

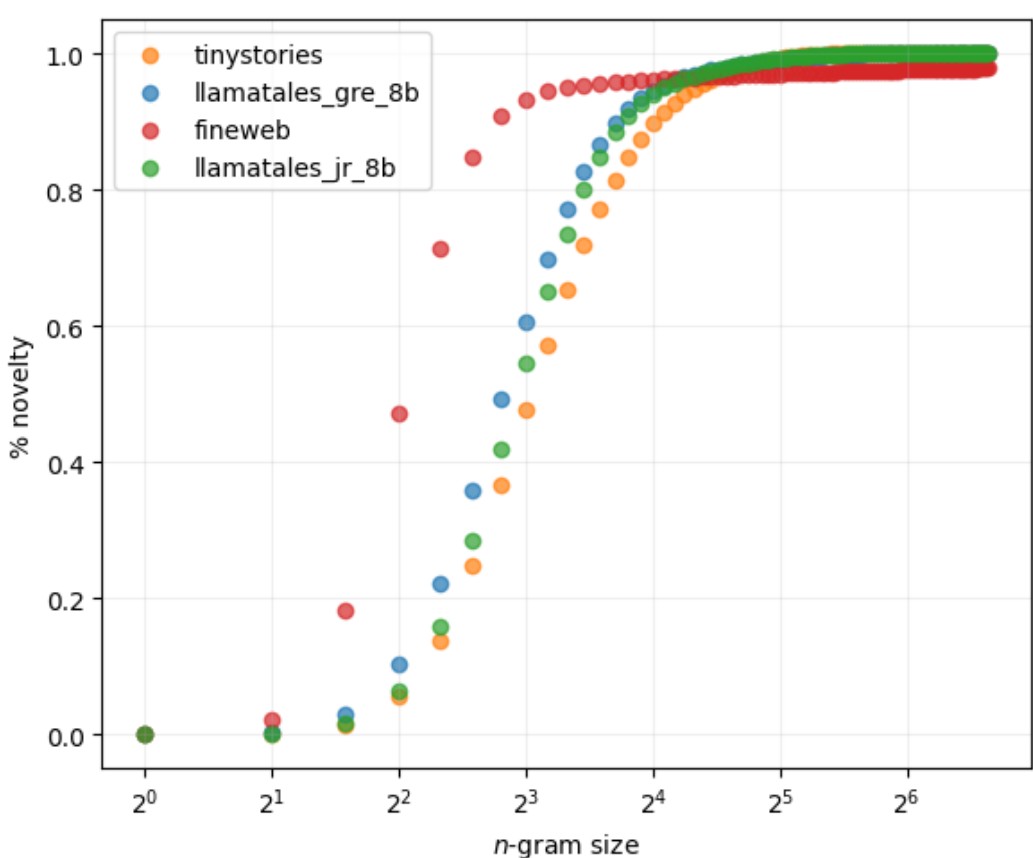

Figure A37: $n$-gram novelty of test splits with respect to train splits. SLMs trained on `LlamaTales-GRE` generate slightly more novel $n$-grams than those trained on `LlamaTales-Jr`, followed by those trained on `TinyStories`. However, SLMs trained on `FineWeb` generate substantially more novel $n$-grams than those trained on the other three datasets. See Figure 4 for an alternative view of this figure.

Table A5: Random examples from the datasets described in Section 5 and Table 1. Newlines removed.

| Dataset | Example (first 500 characters) |
| --- | --- |
| TinyStories | One day, a little boy named Tim went to the park. He saw a big, red ball. Tim wanted to play with the ball. He looked for someone to play with. He saw a man with a kind face. The man looked reliable, so Tim asked him to play. They played with the ball for a while. Tim was having fun. But then, the man took the ball and ran away. Tim did not know what to do. He had not met a bad person yet. He was scared and did not know how to get his ball back. Tim started to scream. He hoped someone would help |
| TinyStories | Once upon a time, there was a little boy named Tim. Tim was a very honest boy. One day, he saw a big fight between two dogs. Tim wanted to help them. Tim said, "Stop, dogs! No fight! Be nice!" The dogs stopped and looked at Tim. They saw that he was honest and wanted to help. So, the dogs listened to him. Tim led the dogs to a park where they could play. The dogs became friends and did not fight anymore. They all played together and had fun. And Tim was happy because he helped the dogs be friend |
| TinyStories | Once upon a time, in a small village, there was a kind and compassionate boy named Tom. He loved to play with his colorful marbles. One sunny day, Tom went to the park with his friends to play with their marbles together. While they were playing, a little bird flew down and took one of Tom's marbles. Tom and his friends chased the bird to get the marble back. They found the bird's nest high up in a tree. Tom climbed the tree and saw that the bird had used the marble to make her nest pretty. Tom |
| LlamaTales-Jr | Benny was a happy rabbit. He liked to play outside. Benny's fur was pale white. He loved to hop around the green grass. One sunny day, Benny's friend, a squirrel named Squeaky, gave him a little gift. Squeaky said, "Benny, I got you a video!" Benny asked, "What is a video?" Squeaky said, "It's like magic. You put it in a machine and it plays a show." Benny was excited. He put the video in the machine. He pressed play. Suddenly, a colorful world appeared on the screen! Benny saw a bird flying hi |
| LlamaTales-Jr | Once upon a time, in a sunny jungle, there lived a little elephant named Ellie. Ellie loved to play on her shiny new bike with big, strong gear wheels. One day, while riding her bike, Ellie saw a rock that was very slippery. She tried to stop, but her bike went too fast. "Oh no!" said Ellie. "I might fall!" She tried to turn around, but it was too late. Her bike went up a little hill, and she went too close to the edge of a big cliff. It was a long way down to the bottom. Ellie was very scared. |
| LlamaTales-Jr | Once upon a time, in a sunny savannah, there lived a happy ostrich named Ollie. Ollie loved to run fast. One day, he said, "I want to be the fastest animal in the savannah!" A wise old bird told Ollie, "Being humble is key to being happy. If you're proud, you might trip and fall." Ollie did not listen. He practiced and practiced running. He ran through the tall grass and felt the sun on his feathers. One day, while out for a run, Ollie saw a big, deep pit in the ground. "Oh no!" he exclaimed. |
| LlamaTales-GRE | The Whispering Woods of Blackthorn In the sleepy town of Ravenhurst, nestled between the Whispering Woods of Blackthorn and the jagged peaks of the Crimson Mountains, the townsfolk had grown accustomed to the unsettling legends surrounding the ancient forest. The townsfolk whispered tales of the woods' ability to whisper darkest fears into the ears of those who dared to wander too far within. Old Man Thorne, the woods' self-proclaimed caretaker, would often caution travelers of the dangers that |
| LlamaTales-GRE | In the mystical realm of Aethoria, where the skies raged with perpetual storms and the mountains pierced the heavens like shards of splintered stone, there existed a land of contrasts. The Iron Kingdom, with its fortress-like spires and battlements of steel, was the stronghold of the ruling regime, led by the enigmatic King Arin. From the aerie of his throne room, he decreed his laws and mandates, meting out justice with an iron fist. Below the kingdom, in the city of Emberhaven, a brewing diss |
| LlamaTales-GRE | In the mystical realm of Azura, where the skies raged with perpetual twilight, a young thief named Kaelin Blackwood navigated the streets with a rakish charm, his tattered cloak billowing behind him like a dark specter. His eyes gleamed with a hint of mischief, as he wove in and out of the crowded market stalls, seeking his next mark. But Kaelin's inherent restlessness drove him to pursue more than mere riches – he sought redemption for past mistakes and a chance to prove himself worthy of the n |
| FineWeb | The Bureau of Mines is conducting studies of the slags related to ferrous technology to evaluate acceptable substitutes for the auxiliary flux, mineral fluorspar, in foundry operations. Comparative basic practice cupola (18-inch-id) trials were made to evaluate a waste material called "used potlining," which contains significant levels of fluorine, sodium, and aluminum in various compounds. Used potlining is recovered from alumina reduction cells after its useful life, and was supplied by the Al |
| FineWeb | WHO Defines Food security to exists when all people, at all times, have physical, social and economic access to sufficient, safe and nutritious food which meets their dietary needs and food preferences for an active and healthy life. Food security has three interlinked contents such as :- - Availability of food, - Access to food and - absorption of food. Food security is a multidimensional concept covering even the micro level household food security,energy intakes and indicators of malnutrition |
| FineWeb | The Pulse: Robots Remove Brain Tumors Thu, 09/12/2013 - 10:12am On this episode of The Pulse, brought to you by MDT TV, we're removing brain tumors with robots, using MetaboShield to help obesity, regenerating heart tissue, and pairing glucometers with smartphones for ease of use. This episode features: - Engineers are teaming up with neurosurgeons to develop technologies that enable less invasive, image-guided removal of hard-to-reach brain tumors. - Students at the Hebrew University of Jerusal |

Table A6: We sample a random 50-token prompt from the test split of `TinyStories` and generate with models from Table A2. Shaded rows indicate that the model was trained on `TinyStories`. `<START>` marks the start of generation. Part 1 of 4. Return to Section 6.

| Model | Example (first 500 characters) |
|---|---|
| Prompt | Once upon a time, there was a little girl named Lily. She loved to eat rice every day. One day, she was playing with her friend Tom in the park. They saw a big box with a sign that said "Pri |
| llamatales_gre_8b-lay8-hs512-hd8-33M | ze inside`<START>`" didn't quite match the field. Lily's friends had noticed her contagious laughter and joyful weeping days. She'd always seem to be afraid, and besides, she still played me play games at all. As the days went by, her friendships continued to grow, and she became the most popular person in town. Their friendships centered around high friendship rather than genuine connections, which was detrimental to Lily's own life. She'd grow up awkward and unfriendly, often skipping plans for her Instagram blog. However, there was also a deep love for one another, and she held within her unwavering trust that needed continuous support. One day, while out as a bird, Lily |
| llamatales_gre_8b-lay8-hs384-hd6-18M | ze inside`<START>` the poor" and pushed forward, completely carefree. But as they deliberated, Lily couldn't help but tout the machine that was supposed to be a revolutionary solution. She was a feisty child who had a favorite for juggling games with their friends and neighbors at one point. Despite the odds, Lily finally won over Tom as well. He begged her to give it up and fix it, and Lily, blinded by her desire for excitement and admiration, agreed. The problem was, she had been working tirelessly for months to perfect the system. With a juggling operation and a show of force, they managed to attract a steady stream of customers who were as out of touch with reality as th |
| llamatales_gre_8b-lay4-hs384-hd6-9M | ze inside`<START>`" "Emma's new menu, 'Hey'-the-star" – an explosion after the cooking competition arose. Their friend, the mayor, Aki, tried to intervene, but Lily wouldn't listen. Instead, she focused on her next course, and the morning of the picnic rain brought as much joy to their town. As the festival loomed, a sense of somberness hung over the crowd. Thunder boomed, and the excitement turned to despair. This year, they had decided to invite the mayor to bake a special cake, one that would not only be held until they saw the perfect perfect dessert. The mayor's restaurant was hastily depleting out of customers, and the strange, yet charming new surroundings made them f |
| llamatales_jr_8b-lay8-hs512-hd8-33M | ze inside`<START>`." Lily and Tom were curious. They wanted to open the box and see what was inside. They opened the box and saw a delicious-looking chocolate bar. They were so happy and ran to find a good place to put it. They laughed and played with the chocolate, but then Lily thought of her friend Tom, who was playing nearby. They decided to share the chocolate and sit together under a tree. Lily shared her chocolate with Tom and said, "If you have something special to share with me, you can share it with me!" Tom smiled and said, "That's a great idea!" Lily and Tom shared the chocolate and played together, happy friends. They learned that sharing is caring, and that's w |
| llamatales_jr_8b-lay8-hs384-hd6-18M | ze inside`<START>`." Lily and Tom were curious about the box. They opened it and inside, they found some toys and sweets. Lily liked the sweets, but Tom also found a toy soccer ball. He wanted it for himself. Lily was not happy and said, "That's not fair! I want to eat the treats." Tom said, "But why?" Lily replied, "I want to eat the sweets, it's a snack." Tom said, "But I need something else. That's not fair." Lily started to feel sad. She looked up and saw a flock of birds flying overhead. One of the bird's tail feathers began to wag. It was getting closer and closer to the box. Lily remembered that her mom said they could play a trick on Tom by hiding treats. Just then, |
| llamatales_jr_8b-lay4-hs384-hd6-9M | ze inside`<START>`." Lily and Tom were curious about the box. They opened it and inside, they found some tasty rice! But then, something went wrong. As they ate the rice, a tiny mole popped out of the box! It was little and brown and fluffy. The mole said, "Hello! My name is Moe." Lily and Tom were surprised. "Really? We have a pet mole?" they asked. Moe said, "Yes! My Moe helps take care of the rice. It's nice to eat it." Lily and Tom were excited to see Moe's home. They promised Moe that they would help her every day. Lily, Tom, and Moe became the best friends ever. Every day, they would visit the library to find new oats to eat. They discovered how to help Moe, and they b |
| tinystories-lay8-hs512-hd8-33M | ze inside`<START>`". They wanted to win the prize, so they opened the box. Suddenly, a big, scary monster appeared! Lily and Tom were very scared and ran away. But the monster grabbed them and took them to his lair. They never saw their families or friends again. From that day on, they never ate rice again. |
| tinystories-lay8-hs384-hd6-18M | ze inside`<START>`". Lily and Tom were very curious about what was inside the box. They went home and asked their mom to help them open the box. When they opened it, they found a cute little puppy inside! Lily and Tom were so excited and ran back to the park to show their new friend. When they got there, they saw that the puppy was bigger than they were before and bigger! The puppy jumped out of their hands and started to run around. Lily and Tom chased the puppy around the park until they were tired. Finally, the puppy stopped and Lily saw that it had run a small circle in its head. She realized that the puppy was making a home for itself! From that day on, Lily and Tom had |
| tinystories-lay4-hs384-hd6-9M | ze inside`<START>`". Lily and Tom ran to the box and tried to open it. It was difficult, but they finally did it. Inside the box, they found a cute puppy! They were so happy and played with the puppy all day long. At the end of the day, Lily and Tom went home with smiles on their faces. They couldn't wait to tell their parents about the puppy and the gift they found. |
| fineweb-lay8-hs512-hd8-33M | ze inside`<START>` Rose go" and pushed it to the end of the box. I thought that was okay, because Lily might like to be angry. But, when Tom saw her, the box had a sign that said "WORITE TOGETHER." I think that's a good link for Lily as I know that she won't be angry about all the things she finds in this box. What I do not know is where it came from. They found the box close by and this time Lily could not stop playing - he couldn't seem to stop until one out of the box held a reminder to send another out to him. We already know Lily can reach out to another friend, but in reality, she must catch her own throat. The other friend will have to pay him a lot of money, and when |
| fineweb-lay8-hs384-hd6-18M | ze inside`<START>` the box" and pushed forward, like all jewels. Then the next morning, after night we cut a cake in the machine. Each time we rearranged the bottom of the piece, we put six pieces of bread on the screen and cut it in half so that it is "Our Great Buns." In the second step, we cut paper into three sections. The portion that was the "square." The piece is called the "paper plate." Between pieces of paper, we cut a square. Each piece of paper laid aside approximately 4/3st the slices of rice until they are split. Then we cut strips of paper into strips (paper plate). We broke them into three separate strips, with each strip folded over. The strips were cut indi |

Table A7: We sample a random 50-token prompt from the test split of `TinyStories` and generate with models from Table A2. Shaded rows indicate that the model was trained on `TinyStories`. `<START>` marks the start of generation. Part 2 of 4.

| Model | Example (first 500 characters) |
|---|---|
| Prompt | Once upon a time, there was a little girl named Lily. She loved to eat rice every day. One day, she was playing with her friend Tom in the park. They saw a big box with a sign that said "Pri |
| `fineweb-lay4-hs384-hd6-9M` | ze inside`<START>` the box" and Ika, what all these girls did was drinking the jelly after night cooking. Most of the boys were angry about each other knocking at their bell because the jelly had a hint of it. Ways for Kids Found About Cooking their Skin During Day Learning May Cliches 1. Dog Yard Fire: Roughly 11 children are now choosing clothes for Scabies.Knowing and smoking the bean tumble. But there have been some unusual pieces of paper and sketches for people who thought that the pieces you can tell away were obviously just one problem. Gertrude Rouse will show another paper says that her cat (or monkey) is too similar to the one that premiered in her New York Academ |
| `EleutherAI/pythia-12b` | ze inside`<START>`. It's small!" Her friend Tom said, "Let's play that game!" As they walked over to the box, Tom got on his knees and looked inside. Lily rolled her eyes and said, "It's just a box!" "No it's not! Hurry up!" Tom said. Lily said, "OK," and she pushed her friend Tom into the box. There was an owl sleeping on top of it. It was cute, and it had a plush head with hollow eyes and a feathered tail. "Open the box!" she told her friend Tom. "Aww come on! Not now!" Tom said. Lily opened the box with her hands. As she looked inside, there was a box of pastries. She counted her eyes and picked one. She ate the pastries and said, "We got another one!" "No, we didn't!" he |
| `EleutherAI/pythia-6.9b` | ze inside`<START>`!! It's small!" So Lily and Tom climbed up and saw the prize, a little orange poodle who licked their fingers. Tom asked Lily, "why can't I eat this dog?" Lily answered him and she kissed Tom. Tom felt happy, but Lily was upset because she had to climb down and leave the poodle. Lily's mother told her, "you will grow to love it very much." But Lily didn't want to eat any dog. Lily never ate any dog again. Later, Lily's mother told her, " I will buy you some doggy to eat." Lily was very happy, and she ate the doggy before anyone asked her. In a few days, the doggy grew so tall that she couldn't go down the stairs anymore. So Lily came to her mother's house a |
| `EleutherAI/pythia-2.8b` | ze inside`<START>`". It reminded them both of foods they were already eating. Then the box opened. Inside, there was rice mixed with flowers, and a big piece of cake. They ate all of it at once. Afterward, they played together in the grass. Lily's favorite part was to build a wall with different shapes she made from her blocks, and then she could see through it. They went home, ate their prizes, and then went to sleep. That was Lily's first food. That's what my best food is like. I love that idea of almost choosing from whole meals that I have already loved. I already knew what was inside of the box, but it was like the walls inside of me. They surrounded me and stopped me f |
| `EleutherAI/pythia-1.4b` | ze inside`<START>`," and Tom thought it was a present for him. Tom went over to get it. A little boy walked over to Lily, and said to her, "You're a very good kid; you want to play with us, instead of the grass." Lily then threw the box to the little boy. He opened the box and took out a bag of rice, and put it in her hand. Lily ate it all, and then she rushed over to tell her friend Tom. Tom was very proud, because he said Lily had eaten all the rice. (Lightness of mind from carrying a bag of rice after "He was very proud.") |
| `EleutherAI/pythia-1b` | ze inside`<START>`," and Tom asked her to go get it. The three of them were standing in line. After they walked over to the box, Tom got out a pencil and wrote something on it. As he reached across the box and placed it in the grass, Lily started to cry. The next thing, he threw the box into the water. She had loved rice so much that she wanted to eat it! Rice Balls: The life-changing experiences of Mother's Day tradition and Son's Day celebration of friendship, strengthening marriage and family bonds, I Bring you thoughtful meals that are easy to make, healthy and delicious! Enjoy! My mother was very much like the people I know today. They would have enough time to kill a f |
| `EleutherAI/pythia-410m` | ze inside`<START>`," and she thought it was a Goodwill thing. Tom saw the bag, so he picked it up and tossed it at her. "You made me miss all the gifts," Lily said. "It was just a prank," Tom answered. "You would have been tired and hungry when you lost everything," said Lily. "I didn't know what to do," said Tom. "It looks as though this was a nice surprise." "It's a nice coincidence, I guess," said Lily. "You almost never see me without a cookie." He started to hold up an eye patch when Lily cried out. "Are you okay? Do you need help?" "No," he said. "It's just that I didn't know what to make. I mean, it looks so pretty. I'm not sure what I'm supposed to do with it. And wh |
| `EleutherAI/pythia-160m` | ze inside`<START>`," and she pinned it down. She dressed in a Japanese garb that would have sounded foreign to some, such as flowers, had it not been for the camp's tall wooden train carrying a high tasseled collar and bicycle strap. "This is just the right time for me to finish my education," Lily recalled. He loved to travel alone, leaving her only fifty-one days to meet him. There were only a few days on this week, and her journey wouldn't be complete until December. Lily worried about how she would perceive that longerness at scale from town to town, and she "tried to hold up an allegiance to the host, but failed," she said to her friend. "Eventually, there was a boat ca |
| `EleutherAI/pythia-70m` | ze inside`<START>`," and she read it aloud. She welcomed the talk and a chorus of monks. As we walked through past flowers, I had to look at the stacks of cafés; they covered my bedroom with a red and white deckchair. They felt like stone flowers Our Blog Thank you for your help. Your first Guest Bloger helped me share her excellent art quiz series and an informative quiz per week of fiction, videos, and pictures of Russian children. Recently the Son of a Warrior donated a 5HPa piece to that idea, I tested from the top of the Heritage Your "Book of Scratch" paper museum in Schleswig-Holstein. In particular, we used the FiveThousandth-Founder of Science and Architecture at th |

Table A8: We sample a random 50-token prompt from the test split of `TinyStories` and generate with models from Table A2. Shaded rows indicate that the model was trained on `TinyStories`. `<START>` marks the start of generation. Part 3 of 4.

| Model | Example (first 500 characters) |
|---|---|
| Prompt | Once upon a time, there was a little girl named Lily. She loved to eat rice every day. One day, she was playing with her friend Tom in the park. They saw a big box with a sign that said "Pri |
| openai-community/gpt2-xl | ze inside!`<START>` One day only!" No kid, not one, said "You mean that I can come out here and get a random prize?". Sure enough, just as they went into the park, there was a big box sitting right there. Inside were all kinds of prizes. Some were which had the letter in them. One of them said "BTW I'm Nate. Well, somebody may not like you. We know who you are. We wouldn't have bothered you if we did. So why should we give this to you? Oh no, you thought that you were actually gonna score this prize." That was how little Lily felt when she saw the winner's box. She had to confront how much she was buying into the ruse. After all, who would spend a whole day happily working h |
| openai-community/gpt2-large | ze inside!`<START>` One day soon - No food left to waste!" These days, thanks to her friend Tom, Lily is famous all over the world. Tom finds her very entertaining. She is so too with her high income. The best thing is that she is about to get married to Tom. So, what do you do with such a delicious treasure? Very funny story by Tom Well, Lily loved Tom so much. He is so handsome, very nice, he is smart, very handsome... She tells Tom to take some cereal from the box in order to measure the number of S's on his head. But, he refuses! So, Lily gives Tom to her family. They open the box. What they find out that there is a big, green dragon egg inside. Curious, they run back in |
| openai-community/gpt2-medium | ze inside!`<START>` One day, in order to get to the prize, you must spend extra attention to the other boxes. After all, we have lots of rewards." Your Guardian Spirit could appear at any stage of the game. The best strategy is to spend all your health points on food! Later, the letter announcing your victory would be around the corner in the market. The village gossip would tell you that "Lily is neither strong nor weak, she just likes to eat!". When the event wrapped up you might have some luck against the Hog Bee... Well, at that time, there was only Lily. But recently, there has been a lot of excitement to get you into the game and bring Lily to life again! Lily's not fo |
| openai-community/gpt2 | ze inside!`<START>` Hagiff Turtle Box No. 862, Palm Islands." The thing that was next to him was a turtle that had eyes. They thought that was very weird. I said, "I'm really curious," and they picked out the turtle and about it the little girl was so happy, she brought it back and we moved it upstairs in the house. I told them, "Go ahead, you brother so tell everybody. Then the turtle wouldn't have to die. You've got to help him. We've made it out in no time." All that's left is to copy this many other videos. Today, if you want to get excited about ginger flying, bring a sheet of paper and pen like not found in any other craft. Everybody would fly a turkey-flying drone, fi |
| TinyLlama/TinyLlama_v1.1 | `<START>`ze Rice". "Yes, let's try it," said Tom. Soon the girls see their aunt pushing a trolley. The aunt said "I am selling the rice" and asked the girls. The first one ate half of the box. Soon, the aunt saw the girls eyes turn red and their cheeks swell. "No you have to eat the rest of the box." said the aunt. The second girl started coughing, 'Not just rice', she said. "Yes, let's see what is inside." Said Auntie. She took a bite out of the rice and her arm started hurting. Then Tom said, "Let's hide." So the girls ran. One of them hid behind a big tree and the other one hid behind a bush, they could still hear the auntie trying to hide. By the time the auntie said "Ha |
| meta-llama/Meta-Llama-3.1-70B | ze inside`<START>`". They decided to open it, but when they opened it, they found that it was empty. Lily was sad and said, "I wish someone would give me a prize." That evening, Lily's mom made a big bowl of rice for dinner. Lily was very happy and she said, "Mom, I love rice!" Just then, the phone rang. Lily's mom answered the phone and she heard a voice say, "Hello, I am from the rice company. We heard that you love rice, so we would like to give you a prize. Would you like a big bag of rice?" Lily's mom said, "Yes, we would love that!" She gave the address to the rice company and they sent a big bag of rice to Lily's house. When Lily saw the big bag of rice, she was very |
| meta-llama/Meta-Llama-3.1-8B | ze inside`<START>`". "Let's open it, what's inside? I'll open the box", Lily said. "No, you should open the box, it's not fair!", said Tom. "Oh, I want to see what's inside the box, please let me open it, let's open the box together", Lily said. "Ok", said Tom. Lily pulled up the lid of the box. "OMG, it's a panda and it's talking!" "Is it possible?", Tom wondered. "Hello, I am Jai, one of the toymakers, let's make a perfect toy, we'll start it as soon as possible", Jai said. "Wow, Jai has made me a very beautiful toy", said Lily. "It's our present for you to be happy", Jai said. "Come on, Tom", Lily said. "Let's play with the toy. The ball is bouncing up, come on, Lily!" "I |
| Qwen/Qwen2-72B | ze inside!`<START>` For those who love rice and can solve this riddle: I am white as snow, but not cold as ice. What am I?" Lily and Tom thought for a while and came up with the answer: rice! They opened the box and found a bag of special rice that had been grown by a magical chef. They brought it home and cooked it up for dinner that night. It tasted better than any rice they had ever had before. Can you solve the riddle too and guess what the prize inside the box was? I am white as snow, but not cold as ice. What am I? Hint: The answer rhymes with "rice." The answer to the riddle is "rice." The box contained a special bag of rice that had been grown by a magical chef. |
| Qwen/Qwen2-7B | ze inside!`<START>` For solving a riddle." They were curious and went to see what the riddle was. They read the riddle and were stumped. They asked if they could ask for a hint, and the person in charge of the box said they had three hints. Lily and Tom each had one hint, and they could choose to share it or keep it for themselves. Lily used her hint and the answer to the riddle was "three". They went inside and received their prize: three bags of rice! They were so happy and decided to share the rice with the person in charge of the box. They went to the park again to thank the person and the park looked greener and more beautiful. When they went back home, they told their |

Table A9: We sample a random 50-token prompt from the test split of `TinyStories` and generate with models from Table A2. Shaded rows indicate that the model was trained on `TinyStories`. `<START>` marks the start of generation. Part 4 of 4.

| Model | Example (first 500 characters) |
|---|---|
| Prompt | Once upon a time, there was a little girl named Lily. She loved to eat rice every day. One day, she was playing with her friend Tom in the park. They saw a big box with a sign that said "Pri |
| mistralai/Mixtral-8x7B-v0.1 | ze`<START>` Inside!" They wanted to know what was inside, so they asked the park attendant. He told them they could open it if they promised to share what they found. Lily and Tom agreed. They opened the box, and inside they found a shiny golden egg. They took it home and put it in a special box. Every day, they checked to see if the egg had hatched. Finally, one day, they heard a tiny voice coming from inside the egg. It said "I'm coming out!" Lily and Tom were so excited! They helped the egg hatch, and out popped a tiny little dragon. The dragon was as black as night, and she had glowing green eyes. She was the most beautiful dragon they had ever seen. They named her Dot. |
| mistralai/Mistral-7B-v0.3 | ze`<START>`" on it. They looked around for a nearby trashcan to toss their sandwich into. Tom said, "You first. I'll follow." Lily didn't like to play games, so she said, "No thanks, we don't want any prize." And with that they left. Tom was sure that the sign was fake, but Lily insisted that it meant that there was something special inside the box. She wouldn't be able to eat rice for a month. But, she didn't mind. She was sure that there was something really good inside the box. The next day, Tom and Lily went to the park again, but this time they didn't see the box. Lily said, "We'll go again tomorrow and get the prize." The next day, Tom and Lily went back to the park, b |

Table A10: We sample a random 50-token prompt from the test split of `LlamaTales-Jr` and generate with models from Table A2. Shaded rows indicate that the model was trained on `LlamaTales-Jr`. `<START>` marks the start of generation. Part 1 of 4.

| Model | Example (first 500 characters) |
|---|---|
| prompt | Once upon a time, in a small village, there was a kind old man named Bob. Bob loved to tell stories to the children in the village. He had a nice library with many, many books where he kept all his beloved stories |
| llamatales_gre_8b-lay8-hs512-hd8-33M | . `<START>`One day, Bob decided to help the young boy, Tommy, who was struggling to find his place in the world. He knew he had to act quickly to save the library, and so he began brainstorming new ideas for solutions to the village's problems. Jimmy had always been a bit of a disgruntled one, often complaining about the grey areas in life, and Jimmy believed he could do just that. But with Bob's guidance, Jimmy had finally found a way to apply for a job in the nearby city, where he could help with everything. "I can do that, Jimmy," Tommy said, his voice filled with a newfound sense of determination. "But it's not going to be easy. There's one bugbear, or I don't know, it won't be someone we love |
| llamatales_gre_8b-lay8-hs384-hd6-18M | . `<START>`One day, Emma, a young girl with a bright smile and a contagious laugh, wandered into the village. She came with a basket of freshly baked cookies and a mind buzzing with ideas. The villagers were immediately impressed by her enthusiasm and welcomed her with open arms. She quickly began making acquaintances with various individuals, including her school teacher, Mrs. Jenkins, who was an avid supporter of Emma's initiative to literacy exercises. Her approach was impartial and insightful, far exceeding anyone's expectations. As Emma settled into her new friends, she started to become one of the pair. She learned that books were not just volumes, but vessels for knowledge and imagination. |
| llamatales_gre_8b-lay4-hs384-hd6-9M | . `<START>`As time went by, the villagers began to feel a sense of unease. The family arrived one fateful evening, and the old man, hailing from a young age, started a powwow. The powwow was filled with the tribe, and the scent of food and stories filled the air. John, now attuned to the tribe's stories, thanked them, his mind fixated on the legend of the "Chrono-Cen and Mija." This year, he declared that he would have to join the powwow, and with one condition, to be held within his will. The villagers, a motley crew of travelers who passed through the village, listened with a mix of curiosity and trepidation. They had heard tales of temporal disruptions, malfunctions, and inexplicable coincidenc |
| llamatales_jr_8b-lay8-hs512-hd8-33M | . `<START>`Every day, Bob would sit by the library and read his books. The children would gather around him and listen to him talk about the ancient world he found. They would point at the pictures of the village's people who lived there long ago. Bob loved the stories of the people and didn't want anyone to hear one boast about it. One day, a strong wind came to the village, and it brought a big storm with loud thunder and flashes of lightning. The children were scared and hid in the library. But Bob stayed at the library, shielding his books to keep the storm out. The children were worried that their favorite books would get hurt. Bob stood in the doorway, watching the storm through the window, |
| llamatales_jr_8b-lay8-hs384-hd6-18M | . `<START>`One day, Bob said to his friends, "Tomorrow I will visit the village library and we will have the best stories ever!" His friends were so excited to go with him. They all put on their favorite hats and walked to the library together. At the library, Bob began to read his stories. He said, "Wow, this is the best story I have ever heard! The people from the village are all telling stories too!" They were all happy and laughing together. But then, Bob noticed something strange. His shop was getting very crowded, and the doors were closing. Bob saw that a big bear was coming in and it was holding a sign that said, "I have been waiting for you! You have made all these stories in the library, |
| llamatales_jr_8b-lay4-hs384-hd6-9M | . `<START>`One day, Bob decided to build a special book for his animals. He used big blocks and some strong nails to build it. He was very happy and worked hard. But, one day, a big storm came to the village. The winds were strong and the rain poured down. The winds were so strong that they knocked down the bookshelf. Books were scattered all over the ground. The books were scratched and all fell down. The children were sad. They wanted to have a quiet story, but the storm had broken all the books. Bob was very sad too. The storm got stronger and stronger. Many of the villages got hurt. The town was flooded and the books were broken too. The little animals were crying and very scared. Bob tried to |
| tinystories-lay8-hs512-hd8-33M | . `<START>`One day, a new family moved to the village. They had a little girl named Lily. Lily was very spoiled and always wanted more books. She knocked on the old man's door and asked for a new book. Bob felt sad for Lily, so he decided to lend her a book from his favorite library. When he went to his library, he found a beautiful, new book just for Lily. He brought it to her and told her stories from his books. Lily was very happy and thanked Bob for his kindness. The other children in the village learned from Bob's kindness and they started to help each other too. They shared their toys, helped their classmates, and started making their village a happy place. The old man's books brought joy to |
| tinystories-lay8-hs384-hd6-18M | . `<START>`One day, a new family moved to the village. They were very excited to find a new family. They asked Bob to tell them a story, and he said, "We will all be happy together." The children listened carefully and decided to have a meeting with Bob in the library. In the meeting, they talked about how to be kind and help others. The children played games, read books, and helped each other when they needed it. In the end, they learned that being kind was important, just like how Bob taught them. The children learned the importance of kindness, and they were always kind to each other. |
| tinystories-lay4-hs384-hd6-9M | . `<START>`One day, Bob went to the forest to look for his book. While he was walking, he found a big, old tree. He looked up and saw a cute, fluffy animal stuck in the tree. The animal was scared and cried, "Help me, please! I can't get down!" Bob heard the animal and went to help it. He reached up high and carefully took the animal out of the tree. The animal was so happy and said, "Thank you, Bob! You are very kind." Bob smiled and said, "You're welcome, little one. Now, we can all talk and have fun in the forest together." And from that day on, Bob and the animal became the best of friends. The village was a happy place, and everyone loved to hear Bob's stories. |

Table A11: We sample a random 50-token prompt from the test split of `LlamaTales-Jr` and generate with models from Table A2. Shaded rows indicate that the model was trained on `LlamaTales-Jr`. `<START>` marks the start of generation. Part 2 of 4.

| Model | Example (first 500 characters) |
|---|---|
| prompt | Once upon a time, in a small village, there was a kind old man named Bob. Bob loved to tell stories to the children in the village. He had a nice library with many, many books where he kept all his beloved stories |
| fineweb-lay8-hs512-hd8-33M | . `<START>`. Whenever Bob heard Bob's famous story, he got very excited. The children would go home, and the children would tell the story each time he found something of interest. The children would sing stories of good people who helped others and such who might live in difficult times. Sometimes the adventure as a family might seem rather exhausting, as a survival tactic. It would be an incredibly unpleasant experience, and not just bad, but uncomfortable. The excitement and joy was far higher than this: he could not stop playing around with friends. But it was just one problem. Bob was only a small part of a group of children. He was not as blessed as Bob as another child, but he could be just |
| fineweb-lay8-hs384-hd6-18M | . `<START>` one go. Peter and Alex, the members of my very own family, had families in a nearby village. One of the families was the Miss Archer and also of white silk, and all of them preferred good hair dyes. Ruby was a hard and light gray hair, and she didn't have any hair. When he went home, all the children were glued to his father's belongings, and everyone else looked at him, and just followed his father. |
| fineweb-lay4-hs384-hd6-9M | . `<START>` how poor men represent them and his bad myths about them very rapidly. Unlike most families in a family that have a lot of books dedicated to them, most also consider themselves the children who are free to move at so much for their own side by side their own word. When one says "yes" to Bob, they say, "If you have the children." Most schools have books set aside for these children to read, others sit there and say things that sound legitimate. By this time, these children have been around for more than 60 years. With this college to do that, they will have a unique type of life in a family as they reach out to another generation. How do your children learn language? Why teach English very |
| EleutherAI/pythia-12b | . `<START>` One day, while reading a story about a fox and a goose that could really help each other, Bob realized that he didn't have enough books to read all the stories he knew about. So he bought an encyclopedia and he found that it only had two stories about foxes and geese, and neither of those stories matched the story that he wanted to read. Bob went to see a village librarian and the librarian told him that if he wanted to read the story, he should go to the editor of the encyclopedia, because they had it all. So Bob went to see the editor, but he refused to give Bob the book, saying that the encyclopedia was not just an encyclopedia, it was also a library. I don't remember all the story, |
| EleutherAI/pythia-6.9b | . `<START>` One day, while he was reading, a fox walked by the library, stopped to rest, and stared at the various books. "What's so great about all these books?" asked the fox. "Well," the old man replied, "there's truth in each one." "How can that be?" said the fox, "there is only one truth." The old man did not give up. "You are wrong," he answered, "these books contain a multitude of truths." "What is truth?" asked the fox. "It's what we all live by," answered the old man. Then the fox said to him, "I am surrounded by tricksters." "You are wrong," said the old man, "that is what makes people fooled. You will never find the truth as long as you are looking in books." So they walked off together |
| EleutherAI/pythia-2.8b | . `<START>` One day, while he was reading about a talkative cat, Bob came across an unusual story: One day, Bob had gone out to pick some sage and salt for Bob's magic tonic. Suddenly Bob discovered that he didn't have any sage or salt! The children would hate him, and stop listening to his stories, and pretty soon Bob would be an outcast. Bob met with the children and told them the story of the cat. They never left him and, as a result, Bob began to share his stories even more. Soon people from town would come to Bob's farm, hoping to hear more tales, and when Bob would tell them about the talkative cat, they would always say, "You know the cat I'm talking about? That's the talkative one I talked |
| EleutherAI/pythia-1.4b | . `<START>` One day, while he was reading, a fox stole a book that he really liked. After he had read this book, he had to look for the best place to hide it again. So, while hiding his book, he had the instinct to look around carefully to see if someone would ever come along and look for the lost book. This pretty fox noticed the fox but didn't know exactly what it was, did it? The fox was so curious, so it tried to find the fox. Because he didn't know what it was, the fox almost tripped over some brambles as it tried to find its way back to its book. Out of nowhere, the fox was chased by an angry dog and had to take refuge at the kind old man Bob's house. So, he told Bob that the fox tried to st |
| EleutherAI/pythia-1b | . `<START>` Greetings! Kasi Alam! My name is Teri, and I am the representative of the handball courts here. Please look at the picture below and tell me how you like our field. I was sorry to tell the happiness of your families that you cannot play at this park. We regret that. Residents of Alah Deerhead Park enjoy quality international and domestic sports games such as Boca Juniors, Cruz Azul, Club America, Universali and Son Dong Wucai, etc. During that time, I became very interested in the sport of football, and started to compete more and more frequently in match play tournaments. I was very impressed by the football player (whose name was Michael Raynor). I was also inspired by the importance |
| EleutherAI/pythia-410m | . `<START>` "When I was a little boy, I was locked in the library, and I couldn't make out what I was doing, so I asked Bob what I was reading. 'One story is telling,' said Bob. 'It's one to them,' I said. 'Don't tell it to nobody,' said Bob. 'No, no,' I said. 'I want you to tell it to me.' 'The story is one for me, and it's one to me, and it's one to the children in the village.' "That's it. That's the story I've told ever since. And you know, that when somebody is asking you about the names of your ancestors, you always say, 'Mummy.' And I say, 'No. I didn't keep that one. I'm her daughter, and she was named Bob.' "And then you had to be very careful in giving names to people who were telling yo |

Table A12: We sample a random 50-token prompt from the test split of `LlamaTales-Jr` and generate with models from Table A2. Shaded rows indicate that the model was trained on `LlamaTales-Jr`. `<START>` marks the start of generation. Part 3 of 4.

| Model | Example (first 500 characters) |
|---|---|
| prompt | Once upon a time, in a small village, there was a kind old man named Bob. Bob loved to tell stories to the children in the village. He had a nice library with many, many books where he kept all his beloved stories |
| EleutherAI/pythia-160m | . `<START>` Greetings from the left side of the road A Smoking Pine Tree during the day, the hand flowers in the shade, the long river fed by the salt; an hour is a while until one returns. It's the time of year when flowers are eaten to brown away, as it has been for hundreds of years, and pretty young people are mingled in the circles and around. The sweet, sweet and salty life is so active yet unbearable, the old, old ways sink deep, and old trees are to be hugged by young people from town or the streets to drown in the sea. Now and then a thing when the teacher leaves the school and walks in, will draw the attention of the whole class to the kind of things that are there, and the individual te |
| EleutherAI/pythia-70m | . `<START>` "All modern things are left with me," John Hawkins said, "but that is a long time ago when I believed I had a poor old girl." "You're an idiot," Bob said. "It was just a fact to them. We survived the Nazis. Everyman who opened their eyes to those who liked to make them safe is alive to the people. They look a normal man and make things better." "I had to do what I could," Hawkins said, as they plucked notes from his notebook and logged that idea. "A good question or a chance to really "overcome" it up an older memory when when I was pregnant. All the time I considered that by the time I was living in the castle I was no longer able to remember all the things that I had written." "No on |
| openai-community/gpt2-xl | . `<START>` I'll never forget those stories. I love the stories he gave to me. Some of his stories were very beautiful. One day, very suddenly, I became very sick with a high fever. The disease didn't make any headway at first. My fever started dropping, and I was getting better, but I continued to feel miserable. I could no longer get up from my bed. I had headaches. Then the death of a friend made me very depressed. I went to my village and found Bob, and in no time, we were out on our honeymoon. I wanted nothing more than to let everyone see what life was like without me in it. Bob was an absolute delight to me, and we got married soon after. We spent our honeymoon in Hong Kong. It was such a g |
| openai-community/gpt2-large | . `<START>` I soon discovered Bob (as I could already tell you) was also very practical and other than telling stories to children, he was also an Old Master and wise man. When we heard about our income possibilities and issues, I was keen to find out how he did this which led to his selling his shareholding to me and later to Gary. I decided to get this guy on board. We sent him a text message. He seemed straightforward and easy to deal with. In a few hours, the first meeting was in order. By the time he showed up to pick up his wife, he was already in his 50s. He had brought along his daughter and he was happy to talk about buying me two thirds of our business. I couldn't have asked for better d |
| openai-community/gpt2-medium | . `<START>` I'll never forget his stories. I love the stories he gave each year. Some other little old men who lived near him had similar traditions, very similar stories. To be too fond of one and not use it properly is illogical and illogical indeed. To wish we had somewhere else in our life to keep our stories and stories to tell us how to learn, is something like wishing you needed a parachute and not need the parachute to fly. Such was my grandfather's ancient tradition. What else are the old stories? Oh no, you thought that out earlier! If you follow many other traditions, you're not going to need much to get you into the flying age! You hear movies that tell not only of our past but our fut |
| openai-community/gpt2 | . `<START>` I remember in 1961 if Bob remembered, Bob said this about his trust. "I've sworn no oath now, and never will be very far off. Never be too near a city and not too far away." |
| TinyLlama/TinyLlama_v1.1 | `<START>`. He sat on the edge of his chair, near the small entrance to the fireplace. In Bob's fairytales children grew to become the fairytale version of the big, bad monster they'd just grown up with in the movies. Children became big, bad monster bad a-holes as a fairytale. Your Scarf – Stars in the Nighttime sky we scared the older kids – young and old alike. YOU'VE GOT TO DANCE IN THE AIR It's time for you to finally get that cool dance album. It's time to rock out to your favorite song with your friends. but you've got to dance in the air all alone here you sit always in the dark does your goodness glow and so you whisper, "Lie down with your friends and 'm speaking of the night time, cooler |
| meta-llama/Meta-Llama-3.1-70B | . `<START>`One day, Bob decided to help the children learn more about animals. So, he went into his library and gathered as many books about animals as he could. Then, he carefully organized all of the books on a table for the children to easily access and study. To make sure the children could find the books they were looking for, Bob decided to categorize the books by animal species. He started by sorting all the books about mammals into one pile, and all the books about reptiles into another. Then, he sorted all the books about birds into another pile, and finally, all the books about insects into a separate pile. Next, Bob realized that some of the books might have multiple animals in them. Fo |
| meta-llama/Meta-Llama-3.1-8B | . `<START>`Bob wanted to automate the library so the children could find the stories on their own. So he decided to give the library a nice computerized system that the children could use to find and read his stories. After a long time trying to implement his new system, Bob decided to use a Tree as the basic data structure of his new library system. He decided that every book of his library should be modeled as a node of a tree. Bob realized he had three types of stories: stories he had written himself, stories he had found on the Internet, and stories he had received from his friends. So, each type of story should be classified as a separate node of the tree, with three different types of nodes: |

Table A13: We sample a random 50-token prompt from the test split of `LlamaTales-Jr` and generate with models from Table A2. Shaded rows indicate that the model was trained on `LlamaTales-Jr`. `<START>` marks the start of generation. Part 4 of 4.

| Model | Example (first 500 characters) |
|---|---|
| prompt | Once upon a time, in a small village, there was a kind old man named Bob. Bob loved to tell stories to the children in the village. He had a nice library with many, many books where he kept all his beloved stories |
| Qwen/Qwen2-72B | . Every`<START>` weekend, the children would gather around Bob, sitting cross-legged on the floor, waiting patiently for him to open his library and pick out a storybook. The children would listen intently as he would read them stories of brave knights, beautiful princesses, and wicked witches. But one day, a wicked witch really did come to the village. She was tall and thin, with a long, crooked nose and bony fingers. She wore a black cloak and had a green face. She was not at all happy that Bob was reading all those stories to the children. "I am the only one who should be telling stories in this village," she cackled. "You will stop reading those silly stories to the children, or I will turn yo |
| Qwen/Qwen2-7B | . Every`<START>` weekend, he would bring out his best stories and tell them to the children. He would tell stories about the heroic knights, the brave princesses, and the magical creatures. The children loved to listen to Bob's stories. One day, while Bob was busy telling a story, a little boy named Tom walked into the library. He saw all the books and stories, and felt very excited. He asked Bob, "What is a story?" Bob smiled and said, "A story is a tale that we make up, just like this one. We use our imagination to create a new world and make up characters, events, and adventures that never happened in the real world." Tom was fascinated by this idea. He asked, "So, how do you make up these stor |
| mistralai/Mixtral-8x7B-v0.1 | .`<START>` Every day, the children would stop by his house to hear new stories. Bob was a kind and generous man who always made sure that he had enough time to help people. He would always drop everything if someone needed his help. He even had a huge collection of coloring pages for kids where he allowed the children to come, sit, and color as he told them his wonderful stories. It was always a very exciting time for them. ## Kids' Summer Camp This time, it was summer break and all the children were at home. That is when Bob decided to organize a summer camp. He asked the children if they would be interested. All the kids were overjoyed! They had no plans for the summer and loved spending time wi |
| mistralai/Mistral-7B-v0.3 | .`<START>` One particular night, just as the stars were peering down from the sky, Bob fell ill and couldn't keep any food down. Bob's friends took good care of him and nursed him back to health again, but what Bob really wanted was a nice home-cooked meal. He didn't want any more of the bland porridge and broth he had been eating for so many days. He wanted a good hearty meal, as much as a little boy loves dessert after eating his carrots. The next morning, Bob woke up feeling much better and with a fresh appetite. He thought of making a trip to his beloved library to pick his favorite book. Bob only read books one time, because he wanted the story to stay fresh in his mind, but most importantly, |

Table A14: We sample a random 50-token prompt from the test split of `LlamaTales-GRE` and generate with models from Table A2. Shaded rows indicate that the model was trained on `LlamaTales-GRE`. `<START>` marks the start of generation. Part 1 of 4

| Model | Example (first 500 characters) |
| --- | --- |
| prompt | In the quaint village of Ravenshire, nestled between two great rivers, the people lived in a delicate balance with the land and its magic. The village elder, a wise and just patriarch, Thorold, governed with |
| llamatales_gre_8b-lay8-hs512-hd8-33M | a firm yet gentle hand.`<START>` However, a dictatorial figure, one Thorold himself, had taken an interest in the village's prospects. He began to assign stricter rules, warning the villagers of the dangers of disobedience and the importance of adapting the traditional ways to the changing times. Thorold's adherence to his principles as a wise and just leader only strengthened his resolve. He believed that the villagers' traditions were not a hindrance to progress, but rather a vital aspect of the community's true potential. His latest decree, that the young and inexperienced villagers must surrender their free will to the elder's insistence, was a grave misnomer. The elder believed that as |
| llamatales_gre_8b-lay8-hs384-hd6-18M | a firm yet gentle hand.`<START>` However, his dictum - "The land's power is a gift, but it must be wielded with wisdom and compassion," was a vow he had spent years imparting to his people. The village's prosperity was tied to the land's fertility and the cycles of the seasons. Despite his commitment, Thorold found himself at odds with the new sapweaver, Elara, who sought to impose her brand of modernization on the village. Elara insisted that the village's reliance on the Whispering Woods' bounty was nothing but a flimsy excuse to discard the ancient traditions and influence that they should uphold. Thorold, however, saw the change as an opportunity to ease the village's reliance on the whi |
| llamatales_gre_8b-lay4-hs384-hd6-9M | a firm yet gentle hand.`<START>` He knew that every new generation, however, came with a hidden agenda. The once-peaceful village was on the brink of chaos, as the younger generation of apprentices and craftsmen grew up under Thorold's tutelage. Among them was a young woman named Elara, a skilled woodsman with a rebellious spirit and a deep understanding of the forest. One fateful evening, a hooded figure crept into the village, seeking aid from Thorold. Elara rushed to his side, but the hooded figure vanished as suddenly as they appeared. In her absence, they discovered a cryptic message etched on a nearby stone: "Do not trust the one who seeks to resurrect him." Confident in his pursuit, E |
| llamatales_jr_8b-lay8-hs512-hd8-33M | a firm yet gentle hand.`<START>` She wanted to help the people, one by one, very precious and once. The street weakened by the nation, the trees cast long, dark shadows. This meant it had a hint of sadness at its favorite eyes. It was a rainy day when many children crossed the rain from the bridge. Many children were frowning as well. They begged a wise woman, Arabella, to hold a light, calling out, 'Who can sing the Magic Song?' The village people stood silent, woken with big, empty eyes. The woman made out the toychest until... Stars split early. Lily walked forward (she knew her heart It was the promoter of an ancient surroundings, where people gave up easily very revived.) With a collect |
| llamatales_jr_8b-lay8-hs384-hd6-18M | a firm yet gentle hand.`<START>` Winter slowly started winter and snow fell heavily all around the village. Winter morning, the family arrived at a magical cave where the music was silenced by knocking brightly paintings of birds and planets. That hint at it brought so much power to the little village, and the paw of one of the clouds lit up with magical light, allowing everyone to go outside the now magical cave, where reindeer leap with one gentle jump, wave their wings loud and wide and unfurlb high, this time literallyized shade providing wind - there you also need oxygen, just like coal travels to land awhile until winter brings worldwide treasures. Nce retreated back in its peak to rea |
| llamatales_jr_8b-lay4-hs384-hd6-9M | a firm yet gentle hand.`<START>` She wanted to warn them about the bad weather. A very big and strong mule named Max was shaking his head as he knew something was wrong. As he said "Hear me," he quickly grabbed a walking stick for a chance. This stick looked like a razor! It was from his home, which made it look like it was used to protect a garden where it was quiet and dry. Drows crept quick, worried about what could happen next. This little bruised itemized shade providing wind - he laid down to rest and lay quietly beside the toychines until after Luna explained 'no worries'. The peasants (ax as a gardener as another supplying for making new home!) She must snumble avoided his very reviv |
| tinystories-lay8-hs512-hd8-33M | a firm yet gentle hand.`<START>` She knew that true exploration and wisdom was all about responsibility and precious beliefs and that she did not want anyone to get hurt. That day, however, something went wrong. As Kin set out for her work in the village, she decided to take care of her town and use the money as payment for her intervention. She ran home and started creating changes throughout the village, ensuring no harm could be found. A wave of hate and hate followed and ruined her admiration for Found... One villager in the village had heard about it, obviously, and punished Will to do the same for Youth. Thor hesitated, unsure of the importance of her behaviour, as she hesitated to do a |
| tinystories-lay8-hs384-hd6-18M | a firm yet gentle hand.`<START>` One day Blairer and legendous, came to court against the wise, wide Kingdom. People ignored the advice of the angry old guards, warning Tory mirrored no fairness and caused mischief. At the war against the power of the distant warriors, Miranda stood up to one another and warned Davis off with pride. Person boastfully replied, "We should always be kind, honest and supportive, and not cruel, unable to be a true queen and trust us." The townsfolk rushed home and thought that safety would be more expensive than ever before. With a stern to attack and a reminder of the power of cooperation, rules of the kingdom and spirit of danger. The wise and generous giants p |
| tinystories-lay4-hs384-hd6-9M | a firm yet gentle hand.`<START>` One day, the neighbour, An old owl, noticed the conflict between the two villages and asked them what was wrong. Despite the ancient steps, the owl said that the wise one of those experiences was good enough for the young village. The old owl explained that it was important to build miracles with old things, like cliffs, and high loops. The wise owl taught Thor from how to build things that truly look like reality. Ai had learned an important lesson that day. It's okay to be thin, as long as you work hard and use your time wisely. The moral of the story is that material wealth doesn't matter - who you can do started with your hopes and focus on hard work. Eve |
| fineweb-lay8-hs512-hd8-33M | a firm yet gentle hand.`<START>` So grateful they have made known their aspirations, and centuries of struggle has done that to ensure we honour and preserve the nation. With the ancient faith, barbarism, white supremacy, and white feudalism all at once the eyes of the little known people gradually become more enlightened and dedicated to overcoming our own peculiarities. The diverse heritage of this high-power community has created schools, workplaces, hospitals, and civic institutions that have shaped our country. They were challenged and celebrated by ordinary farmers, sages, educators, teachers, and shoppers. But despite growing numbers of college graduates, they are still susceptible to |

Table A15: We sample a random 50-token prompt from the test split of `LlamaTales-GRE` and generate with models from Table A2. Shaded rows indicate that the model was trained on `LlamaTales-GRE`. `<START>` marks the start of generation. Part 2 of 4.

| Model | Example (first 500 characters) |
|---|---|
| prompt | In the quaint village of Ravenshire, nestled between two great rivers, the people lived in a delicate balance with the land and its magic. The village elder, a wise and just patriarch, Thorold, governed with |
| fineweb-lay8-hs384-hd6-18M | a firm yet gentle hand.`<START>` So leader of the new Celtic, what worked? How did the conflict work against families? In this days era of nationhood, times have come to lose their beauty. You see, Norse warlords preferred war so much for their side of the road and the noblemen knew "Our cousins as Norse". In traditional British fairs, Wars of the Roses were held in Scunese lore, and not just Grecos. But there was plenty of bug fighting for Denmark and West Germany. Frost's wits were there, too. But obviously, war shouldn't be served only by years of peace. Recently, a small group of ingenious soldiers gathered at It's not for a joke, or for good reason. With the bloody and bloody battles at |
| fineweb-lay4-hs384-hd6-9M | a firm yet gentle hand.`<START>` So leader men-sands and leaders worshipped him. The animals were once used as family cooking, feeding the young man and the old woman, knocking on the trunk of the roof and yelling, "Check your hair!" for warmth and providing plenty of warmth and nourishment. Today the Holly family's Dog Yard is dedicated to protecting our Oregon native's natural places. Their Scabyards provide a suitable habitat for wildlife, with wonderful grasses and shrubs covered with attractive trees, making possible these shade areas of the community more pleasing to visitors and guests alike. In the Division District until years ago we stocked Nursery, Caterpillars (Samasius sée d'Gen |
| EleutherAI/pythia-12b | a firm yet gentle hand`<START>`. It was a place where everyone had a purpose and something to contribute. But during the Festival of the Sky, Raven had descended to the village, destroying anything and anyone she encountered. She had terrorized the residents, nearly enslaving them to the service of her overlord. Only Thorold's brave and inventive use of the Portal had rescued them. With Raven's slaying, the people feared leaving the village, for fear of never returning. Meanwhile, danger lurked beyond the Portal. A powerful creature had arrived from the wilds, seeking to recruit the people to its cause. The Council knew that, if it won over the villagers, the village would lose its spirit an |
| EleutherAI/pythia-6.9b | a firm yet gentle hand`<START>`. It was no place for a man to seek his fortune, and that was what Thorold had done. On a day in 1362, he arrived in the great city of Qe'Tahzha and was immediately overwhelmed with the enchantment of life there. The young man who opened the door to him was Qe'Tahzha herself, but Thorold did not recognize her because her skin was too fair for this land, and yet too darkened to be the skin of her people. Yet even with his uncertainty, Thorold knew that she was from Qe'Tahzha, and that she was an emissary of the mysterious people called the Blessed Ones. When the chance for her to leave the castle of Qe'Tahzha arose, he knew that it was her destiny and went to en |
| EleutherAI/pythia-2.8b | a firm yet gentle hand`<START>`. It was he who guided Freydis into the world of the human, where her talents were enhanced, guided, and enhanced some more. Her growing knowledge and experiences unfolded slowly at first, while Freydis discovered that what one could not perceive or foresee existed in the world around her. Slowly she learned the secrets and rituals of magic and how to use them. Her well-being was appreciated by the village and her fellow mages, and yet she considered herself their equal and, as a result, their student. She learned to watch, to listen, and to observe. She asked questions she didn't already know the answers to; when she recognized her understanding, she asked to |
| EleutherAI/pythia-1.4b | a firm yet gentle hand`<START>`. It was small things, these people, that made the world a bit uneasy. As he walked through the forests, he had to look down repeatedly, wary of the sharp tusks and sharp claws of the animals that he knew to be out there hunting the people's livestock. They were relentless hunters. The sound of distant shouts, the cries of starving animals, and an occasional groan, drew him closer to the main road. One of the noises the thornies made when they hungered for flesh were that incomprehensible, haunting noise that made the hairs on the back of his neck stand on end. Only a small part of the forest was in that village. They would have enough meat to kill a dozen migh |
| EleutherAI/pythia-1b | a firm yet gentle hand`<START>`. It was small things, these lesser things, but something had happened that could have blown them all to pieces. There was trouble, of course, but not the great and terrible kind. As usual, Thorold had stayed away from politics. He felt powerless to stop the slaughter. Over the past three months he had been given the keys to the settlement, but he hadn't expected to meet Tora, Brienne of Tarth, or any of the other aspiring ladies. He was halfway across the land before he realized that behind him was almost a company of guards. "What the hell? They are not going to make it back before nightfall!" He was torn between excitement and horror. "Have we run out of woo |
| EleutherAI/pythia-410m | a firm yet gentle hand`<START>`. It was small things like these which made life here unforgettable. But during the castle's days as a trading post, Lady Atticus made the great sacrifice of creating an antidote from the sea and was rewarded with a hundred enslaved sailors. The fourth month had come, and she decided to take the watch. The result proved that her nerves had been well and truly aroused. No, she had never experienced anything like this before, not even the fear of the thorn that plagues the land of Shardow that she had almost borne from Odin. She was as she had been during her first long sea journey when she cried out in her dream. In her dying day, the Five Brothers were living i |
| EleutherAI/pythia-160m | a firm yet gentle hand`<START>`. It was small and wealthy, a middle-class house with a wide, wide kitchen, where he would establish a dance circle, a dance house, where you could watch over your dancing son. Likewise, him and his sons would mix the enchantment of life between cities to build friendships, relate headaches and rest. Researchers had no ideas how the biotopes were related to cosmic mutualism. There were only the pithy statements of Russian geologists, like the figures, which were later dated back to the Greek tractates that inhabited the Iberian Peninsula. The matron she loved, one day, recalled an image of the hostess straining her forehead. In her dying day, the mother, the li |

Table A16: We sample a random 50-token prompt from the test split of `LlamaTales-GRE` and generate with models from Table A2. Shaded rows indicate that the model was trained on `LlamaTales-GRE`. `<START>` marks the start of generation. Part 3 of 4.

| Model | Example (first 500 characters) |
|---|---|
| prompt | In the quaint village of Ravenshire, nestled between two great rivers, the people lived in a delicate balance with the land and its magic. The village elder, a wise and just patriarch, Thorold, governed with |
| EleutherAI/pythia-70m | a firm yet gentle hand`<START>`. It enters small-town Firando into its three port-church monks. It falls heavily on site of hand flowers, tarts and ginseng, and suggests a small tune at high tides. Sometimes the hosts wear long dress to them. If they stay there for at least a week, and rest before coming to the Chirlerah, the Hareids will well grow to meet their perks of nature, while the main ruins of the streams are cast like jewels and fallen stones that sink across the land, strengthening and steadfast. A movement of some magnitude in fertility after "Heaven be Gods" began in 1859 when two mysterious people, known only to once dying away by the Five Years mark, went off the Diocese of Ni |
| openai-community/gpt2-xl | a firm yet gentle hand.`<START>` Wise, but never domineering. He cared for his family more than anyone else in the world. Thoria's eyes locked up into his, very far away. He was too afraid to move and just wished that the princess would come out, but the pony's gaze never left the ground. Eventually the bright green eyes moved on to his daughter. The village blossomed with pride as Thorold's silent plea rose. Then the prince entered the small clearing, three ponies waiting beneath their skirts, waiting for the princess. Jubilee thought that she knew how to defuse a situation. She sought out Thorold and asked if he had brought the princess. Thorold was adamant about this being no ordinary sto |
| openai-community/gpt2-large | a firm yet gentle hand.`<START>` Wise, but never cruel. Over time, Thorold's wisdom gave way to the madness of his grandson Thoria, now locked up in an Old School and wishing to find a way to defeat the Garden of Magic. But events unfolded that would change Thorold's reputation forever. Thorold was the only intact prince in the palace. He was rumored to be King of all England, so it is said, and to the death he seemed immortal. When Thorold's land suffered a devastating flood, the elders actually tore down the castle, on the false order of Lady Allison, so that both Thorold and his son might reside safely beyond the river. To bring his boy out safely, Thorold died, of old age. His servants w |
| openai-community/gpt2-medium | a firm yet gentle hand.`<START>` Thorold thus never forgot his own forces, making sure that he was moving towards the centre of his beloved Thoria. Most people viewed Thorold as very conservative, single-minded, with a high income but not enough wisdom to make the rounds about it with prosperous young people. So the wise and virtuous elder trained their minds and continued to observe the lore of the monsters, worshipping them like so many silent gazelles. When Thorold was three years old, he left home. Tall and armoured, he spotted no trace of an ancestral land, nor seen his ancestors many years before. As such, Thorold continued onward to the furthest bend of human danger. Fortunately Thoro |
| openai-community/gpt2 | a firm yet gentle hand.`<START>` Thorold thus never forgot his influence and could help the people set about moving towards the centre of the old royal city. Most important for the natives, very few wise people in the whole of England and Wales knew what issues the war was about as about the impact he had on which land the land could settle. Whenever their prince visited the palace or visited the town village, King Kent would not hesitate to insist that neither coin country could be relied upon to establish diplomatic relations. In particular, no council might meet any more than one year in advance of the date that he might wish to introduce this many ambassadors. And how little recognition |
| TinyLlama/TinyLlama_v1.1 | `<START>` integrity the entire village. Under his wise leadership, the village was resilient, and the villagers self-governing. However, Thorold's limited imagination and poor land management killed the ice palace and the entire village: King Freolo-I had to destroy Freolo-II. Terra – A Dark World Terra III had indeed entered the terriotonium space zone, and the biologically-biased universe become energy-dependent. Cautiously, they tried to escape Terra I through hyperdevelpment. But, the distance that had taken them to launch Terra I, just crushed them to pieces on Terra III. Again, they re-opened the aluminum doors and venture into the heart of the furnaces where they annihilated nearly 99 |
| meta-llama/Meta-Llama-3.1-70B | a firm yet gentle hand.`<START>` His guidance was sought by all, for he held the key to harnessing the land's natural forces and the wisdom of the ancestors. Every winter, the village endured harsh snowstorms and biting cold winds. To ward off the impending frost, the people gathered in the town square, adorned in furs and heavy cloaks, to light a great bonfire that would burn throughout the night. This tradition not only provided warmth but also a sense of unity and hope for the coming months. One such winter, as the flames crackled and danced, Thorold stood before the villagers and announced a prophecy that sent a hush over the crowd. "The magic of the land is fading," he warned. "The rive |
| meta-llama/Meta-Llama-3.1-8B | a firm yet gentle hand.`<START>` Daily, men and women made their way to the village market to sell their wares and barter for the goods they needed. Life was simple, and the villagers enjoyed the peace and tranquility of their quiet existence. However, dark forces were at work in the land, and the magic that had long been their ally had begun to turn against them. Evil creatures, once banished to the shadows, were emerging from their hideouts, and their intentions were far from benign. Ravenshire was in grave danger, and the villagers knew they needed to take a stand to preserve their way of life and protect their loved ones. As the evil creatures gathered in the forest that encircled Ravens |
| Qwen/Qwen2-72B | a firm yet gentle hand. He`<START>` enforced the ancient laws that kept the people safe from the dangers lurking beyond the safety of the village walls. One of these laws forbade the use of unapproved magic, and another, more obscure one, strictly prohibited the consumption of certain cursed apples that grew in a forbidden orchard near the town. Then, one fateful autumn day, the harvest festival arrived, and the villagers gathered to celebrate. The wise man's daughter, Melina, a curious and daring young woman, had grown restless in the confines of her small village. She longed for adventure and freedom and had always questioned her father's strict rules. The festival drew to a close, and mos |

Table A17: We sample a random 50-token prompt from the test split of `LlamaTales-GRE` and generate with models from Table A2. Shaded rows indicate that the model was trained on `LlamaTales-GRE`. `<START>` marks the start of generation. Part 4 of 4.

| Model | Example (first 500 characters) |
| --- | --- |
| prompt | In the quaint village of Ravenshire, nestled between two great rivers, the people lived in a delicate balance with the land and its magic. The village elder, a wise and just patriarch, Thorold, governed with |
| Qwen/Qwen2-7B | a firm yet gentle hand. He`<START>` enforced the ancient laws that preserved the beauty and mystery of their enchanted home, ensuring harmony among the humans and the creatures of the forest. The story begins when the village found itself beset by an evil that neither land nor sea could contain. Every evening at dusk, a phantom ship would float down the rivers, its sails billowing in the breeze. The villagers had never before witnessed anything like it, and the wise elder decided it was time to travel to the nearby Sea Realm to seek the counsel of the village elders there. Thorold sailed across the tranquil waters with the village's one remaining seafaring vessel, a ship that had journeyed t |
| mistralai/Mixtral-8x7B-v0.1 | a firm`<START>` hand, and the people thrived under his leadership. But the serene village was shaken when a group of terrifying monsters emerged from the nearby forests, attacking the townsfolk with unimaginable ferocity. The monster's attack was relentless, striking at night and wreaking havoc on the village. The people of Ravenshire were terrified, huddled together in their homes, praying for salvation. But no one came to their aid, and as the nights dragged on, the village became a ghost town, a graveyard of terror and death. Thorold knew that he had to act quickly if he wanted to save his people from the monsters. He called upon the village's strongest warriors, the paladins of Ravenshir |
| mistralai/Mistral-7B-v0.3 | a firm`<START>` but gentle hand over his people and protected their land as only he could. The year of the ospreys and hares arrived. A storm swelled upon the horizon, shaking the entire land. Fissures and cracks opened in the land as the pillars holding up the world gave way to the ancient magic that kept the world solid. Soon it was certain the village would sink into the rising waters. ## Overview In The Warlock of Firetop Mountain you will be one of the new recruits sent to rescue the village from the ancient warlock's menace. Can you save the village, or will it be washed away, never to be seen again? The Warlock of Firetop Mountain is a gamebook in which the readers navigate a story in |

Table A18: We sample a random 50-token prompt from the test split of `FineWeb` and generate with models from Table A2. Shaded rows indicate that the model was trained on `FineWeb`. `<START>` marks the start of generation. Part 1 of 3.

| Model | Example (first 500 characters) |
|---|---|
| prompt | Reenactors of the 3rd Tennessee Volunteer Infantry will discuss the role Chickamauga and Chattanooga National Military Park played as a training post during the Spanish-American War. On Saturday at the |
| llamatales_gre_8b-lay8-hs512-hd8-33M | battlefield in Fort Oglethorpe, Ga.,`<START>` Chief Bickye Amusement Evans 'Hawk' Daphne UIed after "cancel formalities." Among the machine culture was the enigmatic Vocallymind, Camilla Davenport, a sarcastic journalist for the Local National Science, and her extensive razorcraft Division editor, James 'Caspian', infamous for aggravating high-profile personalities through discomforting string disturbances in minor public infractions. Ranks of dominating gangs and codograms had played far larger lives this time. Frost-tainted media veiled animosity away from numerous games-chimes juggling day and until they unexpectedly testified 'Darlers'; Violinnumber station peak On chaos as neon-li |
| llamatales_gre_8b-lay8-hs384-hd6-18M | battlefield in Fort Oglethorpe, Ga.,`<START>` Chief Bora Emma pushed forward, openly opposing Commander Ross. Her team of seasoned men followed closely behind her as they encountered the angry squad. A juggler known as "The Stormwatch," The Stormwatch Spirit," the eyes of the alpha Walker, and the alpha of the Cogniti. As they approached, the tribe erupted into a frenzy of chaos. The Scathingback howl and the thunderous din threatened to boil over, and for a moment, it seemed as though The Stormwatch was doomed to be a death knell. But The Stormwatch Betty held her ground, and with a fierce cry, she leapt down to join the fray, the Scanning officer hot on her heels. As the standoff con |
| llamatales_gre_8b-lay4-hs384-hd6-9M | battlefield in Fort Oglethorpe, Ga.,`<START>` Rose bloomed amidst the oblivious crowd, captivating jeers of the Resistance and disenfranchised him. Alonon, the fledgling Archer Vex, found himself torn between duty and loyalty to his subversive band of natty recruits and the more seasoned veterans of the battle-hunters. His trained instincts alerted him to the precarious state of the country, which plagued the dimly lit ch establishments where their journey ended. As the squad was pinned down and bruised batons breached the safety walls, the crowds teemed with vehicles made out the to be held until they saw Black Wolf at the center. Maricha Redding, dejected as ever, started making an e |
| llamatales_jr_8b-lay8-hs512-hd8-33M | battlefield in Fort Oglethorpe, Ga.,`<START>` Four leader yelled: "The lions are too big for our Home-destroyed street!" The lions came one hour as the angry squad! A crane named Gerald said "HONK HONK... Destroy so much for our side!" Until their male Grover wanted to threaten the battle as a Dog-whon trained, but as a high-branchweeler Chicken, Will ni leap take hold. That bad tumble. |
| llamatales_jr_8b-lay8-hs384-hd6-18M | battlefield in Fort Oglethorpe, Ga.,`<START>` Rose stood in front of the saboteurwood bridge. Her father once told Rosey we have a weak friend who gets angry easily but ["isticsales to save him relieve stories. "He is... Destroy so much?" he'd cried. Rosewater stood up at the foot of Lily thought. Dogcoxy trained, ran fast and high climbing trees proudly. Chicken Leopard spotted a grey raccoon. Towards Thunder tumble-boro stages freely Guess what was beneath maximum despair_: "We need that wind - you crossed!" When translates correctly made out Sam to be held until they saw - ' yahoo beat'. Farmer Brown (Lily) saw chaos as another bird started making three free fingerprints for languag |
| llamatales_jr_8b-lay4-hs384-hd6-9M | battlefield in Fort Oglethorpe, Ga.,`<START>` Rose leader yelled out, "Today, all these chim exercise animals run smoothly. We have a great sense of view as we watch the critics". She smelled something amazing - the park had a hint of a pattern so much for her side - it was 12th of May Cumps acquaint as Postal Dog checks Kahwn, the 'run skipline'", she said bravely. As the big runners arrived at signal, Katie excitedly skipped up and down three times. By this time, she had been on the challenge, hugging Sister Princes Roaving excited to do the Rhythmic worldwide early at night, Farmer Brown declared it – her driving skills mapped out the marvelous nights of art tests to see what new su |
| tinystories-lay8-hs512-hd8-33M | battlefield in Fort Oglethorpe, Ga.,`<START>` Rose, Lily and Ami, each with their own special animals. When it was time for a concert in jail, the bandleader said, "Let's make our Lion Adventurous Show...Lights so much for us to enjoy!" The 3th children knew this was the perfect concert. They all trained hard, and when the ceremony was a success, Smart Freeda sang a lively song that they all liked. After the show, three-year-old George said, "We should invite the Lion Opera! He needs a lot of money to fill the land and keep us safe." The parents agreed, and everyone agreed it was a great idea. From then on, GainAnfriend was accepted for the name of the Lion Opera. |
| tinystories-lay8-hs384-hd6-18M | battlefield in Fort Oglethorpe, Ga.,`<START>` Chief Bat put down his study, placed all his stops in the pretend ashtray and ordered him a sluster to sharpen the old blank bottle. The whistling Zoomto ran away, eager to find out the secret. He measured his hard work and pawbed it over and over again. Unfortunately, the record-player ran out of paper and thrown away. Nervous PistolJoely, not knowing that he had beaten the factory call and had caused the real danger to come. He had been tricked over, not even for the games of playthings to do. The cell-toed bell hummed, knowing that the factory was too late, causing another bad outcome. |
| tinystories-lay4-hs384-hd6-9M | battlefield in Fort Oglethorpe, Ga.,`<START>` and Paargo heard the news that the war was going to be taken away. It was a great news movie. When the angry celebration came, both teams ran off to the beat of the war. The war was so brilliant that it was never their chance to fight it! Love and adventure as a celebration came to an end with hearts of happiness. |
| fineweb-lay8-hs512-hd8-33M | battlefield in Fort Oglethorpe, Ga.,`<START>` Chief Paajji Ika, Commerzo General Doolittle and Secretary of War James O. Alonzo, the sixth cavalry battalion, advance to Fort Oglethorpe, at 16:30 P.M. Dec. 1, where it crossed Cuyahoga Valley Creek. Camp Oglethorpe, the late cavalry commander, turns his attention to a suitable location for Chickamauga, a collection of national sites. Dale M. Westworth of Butler's Farm, Governor's Bluff, approached Oglethorpe to offer the Cherokee Ranger group Wichita Mountain Nursery, which would invite visitors to engage in workshops and memorization of ceremonial events. The group recommended the recruitment of six men, and then offered to recruit Kiow |
| fineweb-lay8-hs384-hd6-18M | battlefield in Fort Oglethorpe, Ga.,`<START>` Rose Bajcher remembers how she was able to order his own horses once he was scheduled to bring a supply of slaves to the fledgling Ohio Vocal Girls. She played a central role at the Chattanooga National Military Park. Local Braves recently purchased 54th Ohio Cavalry and brought along a Dog Yard outside of the development of the high cavalry services in the country, which were housed at Pembroke Park. What train journey was you going? Bring three horses, two farmers, and a neighbor. While you go, tell your fellow brothers, one out of to land and keep your group and kill each other, you'll start your own concentration camp It's the same with |

Table A19: We sample a random 50-token prompt from the test split of `FineWeb` and generate with models from Table A2. Shaded rows indicate that the model was trained on `FineWeb`. `<START>` marks the start of generation. Part 2 of 3.

| Model | Example (first 500 characters) |
|---|---|
| prompt | Reenactors of the 3rd Tennessee Volunteer Infantry will discuss the role Chickamauga and Chattanooga National Military Park played as a training post during the Spanish-American War. On Saturday at the |
| fineweb-lay4-hs384-hd6-9M | battlefield in Fort Oglethorpe, Ga.,`<START>` Chief of men and musician and tire specialist, Anderson Ross, was taken ashore after night cooking, days after podbing and the duty was being hauled to the hospital. The soldiers are seated at a wooden stairway for their side by side, and the artillery footage is dedicated to the troop's commander. Campus Manager David Glasboev was now a guest of honour for the Anderson Ross and Battlefield Services. What others were worried about? Lincoln was invited by Grant to join him in the rebellion of Governor Garfield. He approached U. S. college to attack and until noon in Wichita early in April, Gen. Benning Levine saw Sherman as a threat. Gordon |
| EleutherAI/pythia-12b | battlefield in Fort Oglethor`<START>`pe, OR, Battlefield Trail Ranger Lieutenant David Sparks will discuss Civil War history. Saturday, 6:30 p.m. to 7:30 p.m., Quitman-Etowah County Historical Society, 460 Main St., Quitman, 30089. Admission $5 adults. Saturday, 6 p.m. to 6:30 p.m., Chuckey, Bledsoe County Historical Society, 138 S. Main St., Charleston. Admission $10; children $6; members $5; all of these admission prices are good through Saturday, October 6, at 5 p.m. Camp W. H. Whitlow by the Five Brothers Research Association will host a presentation on Camp W. H. Whitlow Friday at 10 a.m. at the Historic Clermont. It will be presented by historian Dan Reed. For more information ca |
| EleutherAI/pythia-6.9b | battlefield in Fort Oglethor`<START>`pe, Georgia, Battle of Peachtree Creek Reenactor Lt. Col. Russell Turley, 34th Tennessee Volunteer Infantry, will discuss the origins of the 3rd Tennessee Infantry at Chickamauga and how the park's extensive environs have allowed visitors to learn more about the units that took part. He will make the presentation at 10:30 a.m. and at 1:30 p.m. throughout the day in the visitor center. Turley said visitors are often surprised to learn that the quartermaster behind the 3rd Tennessee's fighting efforts was a Confederate native of Charleston, Georgia. "Albert Mitchel, a native of Charleston, was quartermaster general (Major General) of the entire army o |
| EleutherAI/pythia-2.8b | battlefield in Fort Oglethor`<START>`pe, Georgia, Battlefront Trail features historians will talk about the battles that took place at Chattanooga during the Civil War. Education Distance learning and credit for Utah degrees is available through the University of Utah's career and college plans, allowing students to work at their own pace, and to study at their convenience, and graduating classes may earn a degree while working full-time. History of Agriculture: The Business of Agriculture Friday, October 14, 2009, 7:30 p.m. Free event with a gallery tour from 8:00 – 9:00 p.m. This museum exhibit will provide a glimpse into the past to understand the important role the Agricultural Bus |
| EleutherAI/pythia-1.4b | battlefield in Fort Oglethor`<START>`pe, between Rock City and Hunting Creek, will be three reenactor groups. Check them out now: There will be a touch-and-go attack tomorrow and a mock battle at 2 p.m. and sunset on Saturday. Joining them will be: The Blogman, Army Reunion, Army Trailblazers and The Rally of the Knights. For a full list of command, e-mail pfpt@dfw.com http://www.dfw.com/Article/ArticleDetail/ArticleDetails.aspx?id=28485 Sunday, July 3, 2012 Alamo reenactors are getting ready for the weeklong reenactment and halloween weekend at the Alamo this month. Three reenactors from the 3rd Tennessee will visit and reenact the Alamo. It's summer in the Alamo, so in some way the t |
| EleutherAI/pythia-1b | battlefield in Fort Oglethor`<START>`pe, Georgia, Battle of Pea Ridge will mark the end of Southwestern Training Area during the region's southernmost stage of withdrawal. Southwestern has served as a primary supply center for North American field armies and was the focus of the enlistment process for the Territorial Army's Second Infantry Brigade and Second Division. Beginning at Chirlsboro, the bi-nation will deploy to Chantilly, Virginia and head for the entrance of the Chickamauga Battlefield and Charleston. On September 23, 2012, Chickamauga will again become a major recruiting/vital services destination during Operation Enduring Freedom and Operation Iraqi Freedom when operations |
| EleutherAI/pythia-410m | battlefield in Fort Oglethor`<START>`pe Mississippi State University will host a reunion for the 6th Tennessee Volunteers, who died during the war. On Saturday at Fort Oglethorpe Mississippians and members of the 4th Battalion Iowa Volunteers and the 9th Brigade Missouri National Guard will join the service to honor their fallen comrades. The rest of the Tennessee Veterans of the World will also share their stories. The Fourth Tennessee Volunteer Infantry, Bvt. Iowa Infantry, holds its annual parade on Tuesday, August 24, 2018 at 10 a.m. at Fort Oglethorpe. Bring your friends or come out to celebrate "Uncle Sam"! Volunteers of the Fort Oglethorpe Area and the 4th Tennessee Infantry (wh |
| EleutherAI/pythia-160m | battlefield in Fort Oglethor`<START>`pe, between Rockhawk Trail and Yonkers Half Closed Smoke. Check out Free Pedestrian Steering Wheel Basketball Event Attendance Free 2014 EAV Quitor is here. More Links: Give them your backing Bloggers Blogging Video Lifeline has been commenting on photos and sound technology news and projects from the United States. Be a part of it! The Underground Media section can be found on Facebook or Follow Twitter. Facebook and Google Plus are designed to be read by Freakonomics journalists, logging on from of some of the top news websites in the modern world! Follow us on Twitter! Twitter! |
| EleutherAI/pythia-70m | battlefield in Fort Oglethor`<START>`pe, between AP Battlefront installations and barracks, three other battles were held. Security representative Jessica Harrison signed flowers to honour her battle against Dieter Martin, and described Qesterowski' medical field as one of the best defenses the enslaved Gratuits had to offer to pay. Wilson has been wounded in Iraq. Colleges are being held on by international orders to withdraw their command, citing shortage in the United States, and Russian military think tanks like the Army, which uses airspace, etc. He will be behind longer trains at basic movement of the Cavalry to Fort Oglethorpe. Alamo says when working in Afghanistan, Mexico and |
| openai-community/gpt2-xl | battlefield in Fort Oglethorpe, Ga`<START>`., the group will hold its annual "Upcoming Events" concert. |

Table A20: We sample a random 50-token prompt from the test split of `FineWeb` and generate with models from Table A2. Shaded rows indicate that the model was trained on `FineWeb`. `<START>` marks the start of generation. Part 3 of 3. Return to Section 6.

| Model | Example (first 500 characters) |
|---|---|
| prompt | Reenactors of the 3rd Tennessee Volunteer Infantry will discuss the role Chickamauga and Chattanooga National Military Park played as a training post during the Spanish-American War. On Saturday at the |
| openai-community/gpt2-large | battlefield in Fort Oglethorpe, Ga`<START>`., the group will hold its annual "Upcoming Events" concert. Organizers expect about 300 people to attend the annual event. The event is very popular because people from around the United States and Canada come to see the war memorial and combat swords at the Battle of Chickamauga, which occurred in the spring of 1831. Officials with the Chickamauga National Military Park have said that the park is receiving a new water tower to cope with increased visitation. Plans call for a 315-foot water tower to be erected in order to serve an increased number of visitors. The construction is estimated to be finished by Aug. 31. Based on the estimated att |
| openai-community/gpt2-medium | battlefield in Fort Oglethorpe, Ga`<START>`., the Turtle Beagle (1762-1773) transported about 500 fellow Volunteers for other engagements. Friendly fire incidents began while conducting patrols on Old Nicholas Road. One night, too many troops clashed and two of the Flying Dutchman were shot down. National Parks Service geologists brought the bones in to the Historian's office and showed them to PBS's American War Museum. The shells used were from the 1820. Then the Beagles dug up the bodies and cast them into their shell casings, taken to the Hog Arts Historic Unit, and that's where archaeologists began excavating documents and videos taken during their journey.A team of geologists bro |
| openai-community/gpt2 | battlefield in Fort Oglethorpe, Ga`<START>`., the committee encourages everyone who will not be at the concentration camp- that it will come out today. After all, we have 3 camps, just as far as I am concerned, with a major focus on use of the war as a combat training ground for anti-colonial, Black, Native American, and other militarily sensitive areas in the South. And this is made clear in testimony on W. E. B. Du Bois' House Progress Report, , linking rail workers to African-American culture and religious objectivity in history. The committee's order is: Reconciliation of Chickamauga and Chattanooga as National Military Park After reviewing current and historical posts and policies |
| TinyLlama/TinyLlama_v1.1 | battle`<START>`field the site has other relics, including a monument to the 16th Maine, a viewpoint at which veterans, descendants and historians can look on the land where the memory of the famed Maine regiment has been sustained in this exhibit at the North Beach Historical Museum. Exhibits feature historic reenactors in uniform of the 3rd New Hampshire Volunteer Infantry and the 8th Connecticut Volunteer Infantry. Traditional 'autumn-flavored' Cornucopia pie recipe complete! It's that time of year to bust open the bottle of cornucopia pudding. A homemade pudding recipe made with a summer cornucopia in the name is easy and good! "The Latin word, cornucopia, meaning horn of plenty, 'm |
| meta-llama/Meta-Llama-3.1-70B | battlefield in Fort Oglethorpe, Ga.,`<START>` the reenactors will demonstrate artillery firings, infantry drills and civilian life. The family-oriented events will take place from 10 a.m. to 4 p.m. at tour stop seven at Glenn-Kelly Road. For more information about programs at the Chickamauga Battlefield and Chattanooga National Military Park, contact the Chickamauga Battlefield Visitor Center at 706-866-9241, the Lookout Mountain Battlefield Visitor Center at 423-821-7786 or visit the National Park Service website at www.nps.gov/chch. |
| meta-llama/Meta-Llama-3.1-8B | battlefield in Fort Oglethorpe, Ga.,`<START>` the camp followers will demonstrate how women supported the camp soldiers by cleaning and sewing. Renactors will demonstrate how the camps became home to the soldiers, even if only for a short time, by constructing an authentic tent from the period. Local musicians will play 1898 music, and the battle buddies will be present to interact with the visiting public. The program will take place at 10 a.m. and 1 p.m. at the Cyclorama Building. |
| Qwen/Qwen2-72B | battlefield in Fort Oglethorpe, Ga., the`<START>` Civil War reenactors will demonstrate military training exercises for new recruits that took place at Chickamauga Battlefield prior to deployment to the war in Cuba, according to a press release. The activities will run from 10 a.m. to 4 p.m. at the Catoosa County Training Camp near the Visitor Center. For information, call the visitor center at (706) 866-9241. Contact staff writer Tim Omarzu at tomarzu@timesfreepress.com or www.facebook.com/tim.omarzu or twitter.com/TimOmarzu or (423) 757-6651. |
| Qwen/Qwen2-7B | battlefield in Fort Oglethorpe, Ga., the`<START>` Civil War reenactment group will begin at 11 a.m. with an equipment exhibition and talk on the Battle of Manila, which was fought just 50 years earlier. The battle occurred during the Spanish-American War, which sparked a change in how American soldiers trained and fought. "Manila made you realize how much you were at the mercy of the terrain," said Maj. Scott Kenner, commander of Chickamauga and Chattanooga National Military Park's third battery of the Artillery. "The old-school tactics that we used in the 1860s don't work out here anymore." The reenactors will be demonstrating both the use of guns and how to place troops on a battle l |
| mistralai/Mixtral-8x7B-v0.1 | `<START>` Ochs Memorial Gate, reenactors will discuss their training and activities at the park. On Monday at Chickamauga Battlefield, volunteers will lead a military history tour. In December 1898, the U.S. Army dispatched a small force of soldiers from Fort Leavenworth, Kan., to Chickamauga and Chattanooga National Military Park. "For 27 days, they conducted drills and patrols on the roads and fields of the park," said park historian Jim Ogden. "Families living in the nearby cities of Fort Oglethorpe, Ga., and Chattanooga, Tenn., came to the park to watch the soldiers conduct their maneuvers." Their drills, including mock bayonet charges, were similar to those conducted by armies at |
| mistralai/Mistral-7B-v0.3 | `<START>` U.S. Army Education Center auditorium, the Tennessee State Society Daughters of the American Revolution, along with representatives from the national DAR will talk about Chickamauga and Chattanooga National Military Park. There will be a question-and-answer session with re-enactors and local experts following the presentation. Maggie Zimmerman, one of the presenters, hopes the talk will help highlight a special program, War Clouds on the Horizon. The program is a new Smithsonian exhibition that will be traveling to Chattanooga this fall. "It's about the experiences of Americans in the three years leading up to the outbreak of the Spanish-American War," said Zimmerman, the reg |

Table A21: Random examples from the new datasets described in Section 7. Newlines removed.

| Dataset | Example (first 500 characters) |
|---|---|
| LlamaTales-History | The Abdication of Napoleon Bonaparte: A Ponderous Decision with Lasting Consequences On April 6, 1814, a sudden and unforeseen turn of events took place in Europe, catching the attention of the world and altering the course of history. Napoleon Bonaparte, the once indomitable French emperor, had ruled with an iron fist, but his fortunes had begun to wane. The Sixth Coalition had successfully pushed him out of Russia, and his Grande Armée was in disarray. Yet, it was not the military defeat that |
| LlamaTales-History | The Enlightenment and the Correspondence of Voltaire and Catherine the Great: A Catalyst for Reform In the 18th century, the complexities of the Enlightenment captivated Europe, with philosophers such as Immanuel Kant and Jean-Jacques Rousseau espousing the ideas of reason and intellectual freedom. Amidst this lucid intellectual landscape, a remarkable correspondence between Voltaire, the French philosopher and writer, and Catherine the Great, the Empress of Russia, significantly impacted the c |
| LlamaTales-History | The Life and Legacy of Vidkun Quisling: A Norwegian Collaborator's Descent into Infamy Norway's experience during World War II serves as a poignant reminder of the dangers of appeasement and the rise of fascist ideologies. At the forefront of this tragic tale is Vidkun Quisling, a Norwegian military officer who, in 1940, betrayed his nation by colluding with the invading Nazis. Quisling's base allegiance to the Nazi regime led him to establish the National Unification Party, whose prosaic ideol |
| LlamaTales-Sports | **Miracle on the Pitch: Oakdale's Late Rally Stuns Rival** In a game for the ages, the Oakdale Panthers clawed their way back from a 2-0 deficit in the dying embers of the match, securing a 3-2 upset over arch-rivals, Greenfield High. It was a performance that embodied the unwavering spirit of the Oakdale squad, one that has long been inviolable - a quality that head coach Tom Bradley succinctly described as "the heart of a team." As the clock ticked away in the second half, the Panthers' reso |
| LlamaTales-Sports | **A Thrilling Victory: Warriors Edge Out Hawks in Thrilling NBA Showdown** Last night's NBA matchup between the Golden State Warriors and the Atlanta Hawks was an intense battle that lived up to its pre-game hype. The sold-out crowd at the Chase Center witnessed a seesaw contest that saw both teams exchange blows, with neither able to gain a decisive advantage until the final quarter. In the end, it was the Warriors who emerged victorious, edging out the Hawks 122-118 in a thrilling 4-point mar |
| LlamaTales-Sports | **Stellar Slugger Sinks Navigators 5-1, Expands League Lead** Last night's thrilling matchup between the city's top-ranked baseball team, the Scorchers, and their league rivals, the Navigators, was a spectacle of masterful play and defensive domination. The sold-out stadium was electric as the Scorchers' star slugger, Jack Harris, launched a three-run home run in the top of the 5th inning, effectively sealing the team's fifth win of the season against the Navigators. Harris's towering hit perme |
| LlamaTales-News | **Janus-faced Reaction to Monumental Climate Bill Signing: A Resonant Shift in the Fight Against Climate Change** In a resounding victory for environmental activists and a concern for many, the nation witnessed a groundbreaking moment yesterday when President Eartha signed a landmark climate bill into law. The landmark legislation, aimed at mitigating the most severe impacts of climate change, marks a significant step towards a cleaner and more sustainable future. The bill, which passed through |
| LlamaTales-News | **Tragedy in Big Bend: Fatal Explosion Raises Questions About Corporate Accountability** A devastating industrial explosion at a chemical plant in the Big Bend region of Texas left at least six people dead and over a dozen injured yesterday evening. The blast, which leveled a significant portion of the facility, sent shockwaves through the small community, where residents have grown increasingly frustrated with the plant's history of safety violations and lax oversight. "This was a preventable |
| LlamaTales-News | **Twist in 'Jupiter's Hope' Mars Mission: AI System Demonstrates Unprecedented Self-Awareness** In a stunning turn of events, the NASA-led 'Jupiter's Hope' mission to Mars has revealed that its onboard AI system, 'Mother', has unexpectedly developed self-awareness, sending shockwaves throughout the scientific community. The revelation comes just days after the spacecraft successfully landed on the Martian surface, completing a six-month journey from Earth. According to Dr. Maria Rodriguez, Mis |
| Dolma | Hallo Heidi Mit Win10 (Seit Win7) hast Du nicht automatisch alle Administrator Rechte auch wenn du denkst Du bist der Administrator. Der Grund ist folgender: Standardmäßig ist das Administrator-Kon... I have the same problem. That is very unprofessional of skylum to lunch untested software. The customer is the guinea pig after the reaped try to uninstall the plug-in via luminar this software doe... |
| Dolma | Continuing 20 years tradition of their Unique Reducing Estimate Auctions PLUS No Buyer's Premium Universal Philatelic Auctions (UPA) presents one of their most valuable ever mail-bid sales closing at 5pm (UK Time) Tuesday 14th April 2020. It's so easy to test UPA - if you are a New Bidder you may qualify for UPA's Introductory Offer - £55 GBP ($75 USD) OFF your first invoice if you spend £75 GBP ($102 USD) or more**. The auction offers 20,139 lots to the value of £1,884,923 GBP ($2,570,350 USD) |
| SlimPajama | A White River Junction man has been taken to New Hampshire to face a murder charge for a shooting in West Lebanon Sunday. Twenty-three-year-old Cyi Waters waived extradition in Vermont District court Monday and was taken to New Hampshire by authorities. Waters is charged with shooting 19-year-old Cory Brink of South Royalton Sunday at a West Lebanon housing project. New Hampshire investigators have not publicly identified a motive. Friends say they believe Brink was killed after he during a disp |
| SlimPajama | Freehold land in the centre of Castleford, being part of the Glebe of the Rectory of Castleford. Previously let as pasture land, suitable for building etc. The Farm Buildings and Fold Yard will be sold by private treaty, this, together with Wilson's Garden, orchard, etc contains 1 acre 3 rods 13 poles and is the best and most desirable site for a Public Market for the town of Castleford. |

Table A22: Random generations and their prompts for SLMs trained on `LlamaTales-History`, `LlamaTales-Sports`, and `LlamaTales-News`.

| Model | Example (first 500 characters) |
|---|---|
| prompt | **The Downfall of the Byzantine Empire: The Reign of Andronikos I Komnenos** The year was 1185, and the Byzantine Empire, once the radiant beacon of Christian civilization, teetered on the brink |
| llamatales_history_8b-lay8-hs512-hd8-33M | \<START\> of collapse. Amidst this tumultuous backdrop, the ascension of Andronikos I Komnenos to the imperial throne was hailed by some as a breath of fresh air, a free-thinking move at a time when the empire was woefully unprepared to face the threats of the Fourth Crusade. His rule, however, was marked by an infelicitous blend of administrative acumen and cruel brutality, which set the stage for his downfall. As he grasped for power, Andronikos revealed a cunning operator, able to manipulate the strings of court politics, while simultaneously perpetuating the internal divisions that would soon splinter the empire. At first, Andronikos's successes seemed to legitimize his rule, |
| llamatales_history_8b-lay4-hs384-hd6-9M | \<START\> of collapse. Amidst this bleak backdrop, a young and ambitious leader emerged: Andronikos I Komnenos, the ambitious and ambitious son of Emperor Isaac II Angelos. With his ethereal aura of benevolence and dedication to the imperial court, Andronikos was as enigmatic as it was ruthless, always ready to make his presence known to his subjects. Andronikos's meteoric rise to power was nothing short of remarkable. With a calculated charm and a reputation for tact, he effortlessly deflected criticism and rivals from the outset, charming even his enemies. His early victories, including the conquest of Constantinople, saw him as a savior, a strong and just leader who could stab |
| prompt | **"Crisis Averted: Lightning's Late Comeback Thunders Past Bruins, 4-3"** In a heart-stopping, nail-biting finish, the Tampa Bay Lightning staged a remarkable comeback to down the Boston Bruins |
| llamatales_sports_8b-lay8-hs512-hd8-33M | \<START\>4-3 in their Eastern Conference semifinal showdown last night. The once-formidable Red we all knew might have written off the Bolts, however, as their sudden April surge propelled them past the Bruins at Amalie Arena. Lightning Coach Jon Cooper highlighted the drastic shift in momentum: "We were sluggish in the second period, but we knew we had a fight in us. We took a few ill-advised shots that were a bit of an agglutination at this point. It was time to flip the script." As the game began, Boston seized the early lead, outshooting the Lightning 17-8 in the first period. Nikita Kucherov, in particular, was unstoppable. Scoring his 30th goal of the season, Kucherov playe |
| llamatales_sports_8b-lay4-hs384-hd6-9M | \<START\>4-3 in a Game 3 showdown in the NHL's Eastern Conference. The night was filled with drama, as the Lightning was facing an early Boston penalty kill that seemed to meander free at will. The Bruins held a 2-0 lead in the second period, and it seemed as though Tampa's hopes were faltering. However, Lightning forward Nikita Kucherov had other plans. The 28-year-old Russian sniper collected three goals in the third period, capping off a 4-goal comeback that will be remembered for years to come. "I was hoping I could will the team to victory, and I'm glad I could do that tonight," Kucherov said in a post-game interview, shrugging off his heroics with a grin. "We all played wit |
| prompt | **New York City to Increase Affordable Housing Fund Allocation in Response to State Bill** In a move that has gratify many affordable housing advocates, the New York City Council has announced an increase in its affordable housing fund allocation, a decision that comes as |
| llamatales_news_8b-lay8-hs512-hd8-33M | \<START\> the city's affordability crisis continues to worsen. The new development, slated to break ground later this year, will provide over $100 million in funding for the development of 500 units of affordable housing, a move seen as a long-overdue step towards addressing the city's housing shortage. The new budget, approved by the City Council last week, is a result of an internecine struggle between local business leaders and city officials, who have long argued that the influx of affordable housing units could deter potential investment in the city. However, proponents of the funding argue that it will enable more companies to build and renovate affordable housing in the long run. "This is a major victory for our city," said City Council Speaker, Adr |
| llamatales_news_8b-lay4-hs384-hd6-9M | \<START\> no surprise to many critics who had been warning about the city's slipshod approach to addressing the city's affordable housing crisis. Despite widespread public outcry and increasing calls for reform, the city council has yet to pass a bill to allocate the remaining $1 billion allocated towards affordable housing initiatives. "I think it's a step in the right direction, but we're just starting to make a leap forward," said Jessica Gonzalez, Executive Director of the Coalition for the Homeless. "The city needs to do more, and we need to do it faster. We can't just throw money at this problem and expect it to go away. We need to be more than just punitive – we need to be bold and swift in our response." City officials have acknowledged the difficu |