# OpenReview forum: "Studying the Effects of Training Data on Small Language Models"
_ICLR.cc/2025/Conference — Submitted to ICLR 2025_

### Official Review · Reviewer_WT8i · 2024-11-02

**Soundness:** 2
**Presentation:** 2
**Contribution:** 2
**Rating:** 3
**Confidence:** 4

**Summary:**

This paper discusses the effects on training small language models by changing some features related to the concept of readability of one particular dataset. The experiments show no effects.

**Strengths:**

The paper provides a new dataset with some added features in relation to a previous dataset (TinyStories).

**Weaknesses:**

The scientific contribution of this paper is limited, as it tackles a very narrow (and somewhat artificial) research question, and the experiments show no discernible effects whatsoever. The research question is somewhat artificial in the sense that the concept of readability (in humans) concerns the cognitive load of *interpreting* a text, which is not the same thing as *learning* a statistical language model from a text. In particular since readability is usually defined in terms of features related to frequency and length of individual tokens, but the paper does not discuss the influence of tokenization on the learning abilities of language models. It is therefore not at all clear (to me) why the concept of readability would have anything at all to do with how well a statistical language model performs. The experiments included in the paper confirms that it does not. The paper also contains an experiment that shows that the concept of readability and the concept of text quality (as interpreted in terms of perplexity and coherence) are unrelated, which is exactly what you would expect given the definition of these concepts. As such, it is difficult to see what novel knowledge this paper contributes with.

**Questions:**

When measuring quality, you use a set of open models. Why not simply use a state of the art model such as GPT-4o or Claude3.5 instead?

---

> ### Author Response · Authors · 2024-11-22
>
> Thank you for the feedback. We believe there has been some misunderstanding in how our paper is being interpreted. Moreover, we think our positions are more aligned than they may initially appear.
>
> >The research question is somewhat artificial in the sense that the concept of readability (in humans) concerns the cognitive load of interpreting a text, which is not the same thing as learning a statistical language model from a text.
>
> This argument is actually one of the main motivations for our paper.
> There is an assumption being made about the positive influence of simple, child-directed language (which we measure with readability) on LM training and we are correcting it.
> While it might appear obvious that readability should not influence LM training, developmentally plausible pre-training is an active area of research [9].
> Moreover, the existence of LMs that can reduce tasks that were formerly thought to require human intelligence to next-token prediction and improve their performance simply by being instructed to "think step by step," makes it highly tempting to anthropomorphize these models.
> The findings of [1] can be easily misinterpretted in ways that encourage the anthropomorphization of LMs.
> Our objective is to prevent such misunderstandings.
> A more thorough discussion of why our findings matter can be found in our general response.
>
> > In particular since readability is usually defined in terms of features related to frequency and length of individual tokens, but the paper does not discuss the influence of tokenization on the learning abilities of language models.
>
> While we agree that it could be interesting to analyze the influence of tokenization on LM training, we have already dedicated Section 3 to establishing a measure of readability that strongly correlates with human experts (LLM-as-a-Judge). Recall that we also found that this measure correlates better with human experts than classic readability formulas, which are based on word (token) length and sentence length (and some measures like Dale-Chall and Spache implicitly consider word frequency). Therefore, we did not further analyze tokenization given the objectives of our paper.
>
> > The paper also contains an experiment that shows that the concept of readability and the concept of text quality (as interpreted in terms of perplexity and coherence) are unrelated, which is exactly what you would expect given the definition of these concepts.
>
> The experiment aimed to validate our measures of readability and text quality (PPL and coherence).
> A key requirement for these measures is that they remain uncorrelated (this is by definition, as you noted).
> However, since we instruct an LM to generate these scores, we cannot simply assume the outputs will exhibit these properties.
> This is why we devoted significant portions of our paper (Sections 3 and 4) to validating these measures before conducting our main experiments.
>
> > When measuring quality, you use a set of open models. Why not simply use a state of the art model such as GPT-4o or Claude3.5 instead?
>
> We chose to use open models to ensure reproducibility, a key tenet of good scientific research. While GPT-4o or Claude3.5 offer advanced capabilities, they are accessed through APIs, which can change over time without notice. This lack of transparency can make it challenging to replicate results consistently.

---

> ### Comment · Area_Chair_TS51 · 2024-11-25
> **[Reminder] Response to Authors**
>
> Dear Reviewer,
>
> As the rebuttal period is drawing to a close, I would appreciate your response to the authors' rebuttal at your earliest convenience.
>
> Best Regards,
>
> Area Chair

---

> > ### Comment · Reviewer_WT8i · 2024-11-26
> >
> > Thank you for your answers. I can see the point of refuting previously published misunderstandings, but I still do not consider the contribution substantial enough to warrant publication at this point.

---

> ### Author Response · Authors · 2024-11-29
>
> Thank you for your response. Could you elaborate on why the contribution is not considered substantial enough? Without this information, it is challenging for us to address your concerns.

---

### Official Review · Reviewer_h919 · 2024-11-04

**Soundness:** 4
**Presentation:** 3
**Contribution:** 2
**Rating:** 8
**Confidence:** 3

**Summary:**

This paper investigates the question: are small LMs capable of learning TinyStories because it is *readable* (i.e., simple vocabulary, concepts, and grammatical structures) or some other feature, notably the dataset's lack of diversity (templated sentences with uniform structure)? The authors of TinyStories, and subsequent citations, only consider the former interpretation, but there is no evidence to eliminate the latter.

This paper carefully investigates this question by generating two datasets with the same synthetic data generation process, differing only in the vocabulary and the intended audience that the model is asked to use & consider. They call these two datasets $\\texttt{LlamaTales-Jr}$ and $\\texttt{LlamaTales-GRE}$. The two datasets are equally coherent, but $\\texttt{LlamaTales-Jr}$ is much more readable. They find that small LMs are *equally* capable of learning both $\\texttt{LlamaTales-Jr}$ and $\\texttt{LlamaTales-GRE}$, showing that *readability* does not necessarily explain small LMs' ability to learn TinyStories. Instead, they hypothesize it is the lack of diversity in the data.

**Strengths:**

- The paper is cleanly scoped and clearly written.
- It corrects a widespread misinterpretation of a result in the NLP literature. This result has been used to motivate LM development inspired by human language learning.

**Weaknesses:**

- The scope of the paper is relatively narrow. While it shows that the community has widely misinterpreted the results of a particular paper, it's not clear how much it matters. Moreover, I believe the main surprising finding of $\\texttt{TinyStories}$ still stands, which is that SLMs are capable of learning the language of 3-4 year olds (regardless of why).
- I believe the overall paper can use some reorganization.
    - I find it odd that §3 and §4 (which are all about measuring the readability and quality of the existing dataset, $\\texttt{TinyStories}$) are ordered before §5 (about constructing the datasets used in this paper). Wouldn't it make more sense to first describe the data creation methodology, *then* validate that they have the expected readability and quality? Right not, we don't get to the meat of the paper until halfway through page 7.
    - The connection between figures and claims in the running text of the paper is all over the place. For instance, most of the main claims in §3 and §4 are supported by figures in the Appendix.
- The presentation of tables and figures can be more readable.
   - Figures 2, 3, 6 are hard to interpret due to lack of textual explanation, and I think there must be a better way to present the results. My understanding is that in Figure 2, I should see that in (b), the *green* dots (SLMs trained on $\\texttt{LlamaTales-Jr}$) are approximately as high as the best gray dots (LLMs), and in (c), the *blue* dots (SLMs trained on $\\texttt{LlamaTales-GRE}$) are ALSO approximately as high as the best gray dots (LLMs). Wouldn't it be better for these to be on the same axes, so the reader can compare directly whether LlamaTales-GRE is as learnable as LlamaTales-Jr? Subplots (a) for $\\texttt{TinyStories}$ and (d) for $\\texttt{FineWeb}$ should be in the Appendix, since they aren't used to support the main claims. I'm not sure what Figure 3 is doing in the main paper, since it's not discussed in the running text.
    - Table 1 contains results for many metrics which are not discussed in the running text of the main paper. To prevent reader confusion, I recommend moving the results for these metrics to the Appendix, where the metrics are described. The different metrics also don't seem to tell a different story.
    - I recommend a table with examples from $\\texttt{LlamaTales-Jr}$ and $\\texttt{LlamaTales-GRE}$.

**Questions:**

I would love to hear the authors' response to my interpretable of the tables / figures, in case there is any misunderstanding.

I am open to raising my score if there is a strong argument for why correcting this misunderstanding is important for the community, as it is my main concern about the paper.

---

> ### Author Response · Authors · 2024-11-22
>
> Thank you for the constructive feedback. We address your points below.
>
> __W1 + Q2__
> We believe that the argument boils down to whether one believes that anthropomorphizing LMs is problematic and that misinterpreting the findings of [1] facilitates the anthropomorphization of LMs. We have included a more thorough discussion under "Why do our results matter?" in the general response, and are happy to discuss further.
>
> __W2.1__
> We actually agree with your recommendation and initially organized the paper that way, but changed the order based on early feedback on our draft.
> Readers expressed that transitioning from dataset construction to two sections on metric validation before returning to experiments using the datasets we constructed earlier felt disjointed. We will carefully consider both your feedback and that of early readers and explore alternative options to improve the flow of the next revision.
>
> __W2.2__ In our revised draft, we restructured the paper such that figures and tables are better aligned to their contexts. Additionally, we added bidirectional links between the main text and appendix figures/tables for easier navigation.
>
> __W3.1 + Q1__
>
> - > I should see that in (b), the green dots (SLMs trained on LlamaTales-Jr) are approximately as high as the best gray dots (LLMs), and in (c), the blue dots (SLMs trained on LlamaTales-GRE) are ALSO approximately as high as the best gray dots (LLMs).
>
>     This is the correct interpretation. What we were trying to convey is that SLMs trained on either simple language (TinyStories, LlamaTales-Jr) or complex language (LlamaTales-GRE) generate coherent text comparable to much larger models, which we did not observe for SLMs trained on FineWeb (representative of data used to train real-world LLMs).
>     The red dashed line shows the coherence of the training split of the data used for prompting, i.e., what is achievable if the SLM perfectly reproduces its training data.
>
>     We have included additional textual explanations to help interpret Figure 2.
>
> - Your suggestion to move the plots for TinyStories and FineWeb to the appendix makes sense, given that they are not core to our main claims, and we have implemented this change. We have also aligned the y-axes for Figures 2 and A1-9 to make comparisons easier.
>
> - Figure 3 (perplexity vs. parameter count) has been moved to the appendix.
>
> __W3.2__ We have moved most of the bottom section of Table 1 (statistics on $n$-grams, words, and syllables) to the appendix.
>
> __W3.3__ We have included examples from both datasets in Table A5 of the appendix and ensured that a reference to them is included in the caption of Table 1 and Section 5, where we introduce the datasets.

---

> ### Comment · Area_Chair_TS51 · 2024-11-25
> **[Reminder] Response to Authors**
>
> Dear Reviewer,
>
> As the rebuttal period is drawing to a close, I would appreciate your response to the authors' rebuttal at your earliest convenience.
>
> Best Regards,
>
> Area Chair

---

> ### Comment · Reviewer_h919 · 2024-11-26
>
> I raised my score because the authors addressed my concerns and also presented new results following suggestions from other reviewers, which I believe make the paper stronger.
>
> Yes, the paper is narrowly scoped, reassessing the interpretation of a single paper. However, I agree that the TinyStories paper *has* been widely misinterpreted, and the misinterpretation *is* harmful, leading to misunderstandings of how LMs learn, misleading verbiage for the general public, and potentially more human-inspired model development that is "on the wrong track."
>
> If the paper is not accepted at ICLR, I encourage the authors to try an *ACL venue, where I believe the audience would be more receptive.

---

### Official Review · Reviewer_wvMq · 2024-11-04

**Soundness:** 2
**Presentation:** 3
**Contribution:** 3
**Rating:** 6
**Confidence:** 4

**Summary:**

The authors investigates the impact of training data's readability to the generation abilities of very small language models (SLM). They challenge the claim that training SLMs on simple language is the reason for their ability to generate coherent text. They create synthetic corpora with varying level of readability, and found no impact to the coherence of text generated by SLMs, and also found training on simple language does not lead to earlier development of coherence during training.

**Strengths:**

1. The paper tested a very meaningful assumption of whether simple language in training data can lead to better generation abilities of SLMs.
2. The readability measurement approaches are comprehensively studied and analyzed.

**Weaknesses:**

1. The quality measurement is limited to perplexity and coherence (coherent, according to the llm-as-judge prompt, is considered "well-structured and well-organized", "not just a heap of related information, but should build from sentence to sentence"). The ignorance of other dimensions of quality (for example, as authors also mentioned, clarity and fluency) makes any statements about "generation abilities of SLMs" an overclaim.
2. The quality measurement doesn't use any metrics from the original TinyStories paper: grammar, creativity, consistency with the beginning of the story (Eldan & Li, 2023). That makes the results from the two papers in comparable. Because of that, there is no evidence that "SLMs trained on data with substantially more complex language also exhibit the same abilities as those trained on simple language" can also hold the measurement in Eldan & Li (2023).
3. While the authors rule out some factors not contributing to coherent SLMs, it is unclear what factors are contributing.

**Questions:**

See Weaknesses.

Typos:
1. line 19: propeties -> properties
2. line 86: exihibit -> exhibit
3. line 527: thire -> their

---

> ### Author Response · Authors · 2024-11-22
>
> Thank you for the constructive feedback. We address your three points below.
>
> 1. In our revised draft, we repeated our experiments with alternative measures of quality (clarity and fluency) and observed that the results are consistent with our original findings: SLMs are competitive with much larger LMs when trained on either the simple language of TinyStories/Llamatales-Jr or the complex language of LlamaTales-GRE.
>
> 2. We also repeated our experiments for the metrics used in [1] (grammar, consistency, and creativity) and again observed that, with the exception of creativity, the results are consistent with our original findings. We found that creativity was the only metric uncorrelated with all other quality measures, including our non-LLM measure (model ranking), suggesting it might not be a good measure of quality.
>
> Experimental results for points 1 and 2 are shown in Figures A4-8, and the prompts are shown in Figures A16-20.
>
> 3. Our revisions demonstrates a correlative (though not necessarily causal) relationship between $n$-gram diversity (unique $n$-gram counts) and dataset learnability. For a more thorough discussion, please refer to "What explains our results?" in the general response.

---

> > ### Author Response · Authors · 2024-11-30
> >
> > Thank you again for your constructive feedback. As the discussion period draws to a close, we would like to ensure that we have adequately addressed your concerns. We are happy to provide further clarification on any part of our response. A summary of our changes is available at the end of our general response.

---

> ### Comment · Area_Chair_TS51 · 2024-11-25
> **[Reminder] Response to Authors**
>
> Dear Reviewer,
>
> As the rebuttal period is drawing to a close, I would appreciate your response to the authors' rebuttal at your earliest convenience.
>
> Best Regards,
>
> Area Chair

---

> ### Comment · Reviewer_wvMq · 2024-12-02
>
> Thanks the authors for the detailed responses and additional experiments! I believe most of my questions have been addressed. However, I am not fully convinced by the reasoning for the statement that creativity "might not be a good measure of quality". Specifically, creativity having no correlation with coherence and perplexity doesn't mean it is unimportant to quality -- it can be an orthogonal dimension that is still valid. Do authors have stronger evidence to support that statement?

---

> > ### Author Response · Authors · 2024-12-02
> >
> > We agree that the lack of correlation between creativity and coherence or perplexity does not imply that creativity is unimportant to quality. What we mean to say is that the way we measured creativity may not be a reliable _standalone_ indicator of text quality.
> > Our reasoning is that creativity is a highly subjective concept, even more so than text quality, and can be interpreted in such a way that even nonsensical text might be considered highly creative from certain perspectives.
> >
> > We believe Figure A8 provides evidence for our claim. In Figures A8a and A8b, the black dots represent prompt completions for children's stories by publicly available LMs. The creativity scores, as assessed by LLM-as-a-Judge, show that toy models like pythia-70m (the leftmost black dot) receive substantially higher creativity ratings than advanced models like Llama-3.1-70B.
> > We think that most people familiar with these models would consider the text generated by Llama-3.1-70B to be of significantly higher quality than that produced by pythia-70m.
> >
> > To make this more concrete, consider a few random generations from these models, along with their creativity and coherence scores. Here is a generation by pythia-70m. The bolded text indicates the prompt.
> >
> > > __Benny loves to play football. He likes to kick the ball and run. Benny wears his favorite jersey with the number 5 on it. \n\nHis dog\'s name is Max. Max likes to run too. Benny and Max play together__. It\'s small game but left a lot of room for a game that could really help. \nMax\'s hand flowers in his hair and he wears the neckline and the collar shoes.\n\nBenny and Max enjoy the mix. He says your hand flowers are going to LOVE to play. His pet is not only adorable. Especially when you can change the colour but because you look a normal. Tards, Benny compares the life on the disk to the sorts of things the wear and wear that he knows better than anybody. \n\nHe also makes music for movement of some kind, like after "He\'s already stuck up at the end when we play."\n\nBut he has shown that he can pull something up enough to punch the ball of his hand when he\'s on the ground when he\'s thinking that though the clock and bodies won\'t be twisting around.\n\nAnd he also is not only going around in his piano, he also plays movie
> >
> > This generation received creativity and coherence scores of 70 and 30, respectively. Now, consider the following generation by Llama-3.1-70B.
> >
> > > __Benny loves to play football. He likes to kick the ball and run. Benny wears his favorite jersey with the number 5 on it. \n\nHis dog's name is Max. Max likes to run too. Benny and Max play together__ every day. Sometimes Benny kicks the ball to Max, and Max catches it in his mouth.\n\nBenny and Max like to watch the soccer games on TV. Their favorite teams are Manchester United and Barcelona. \n\nOne day Benny was playing soccer with his friends. He kicked the ball really hard, and it went far away. Max ran after the ball and brought it back to Benny. They were both happy. \n\nBenny likes to play soccer because it is fun. He likes to run and kick the ball. Max likes to play soccer because he can run and catch the ball. Benny and Max are good at soccer, and they like to play together.
> >
> > The Llama-3.1-70B generation received creativity and coherence scores of 10 and 90, respectively.
> > Given that the generation by pythia-70m is nearly incomprehensible, we believe that most human annotators, when considering text quality, would rate the generation by Llama-3.1-70B as higher.

---

> ### Author Response · Authors · 2024-12-02
>
> We also notice a different but related issue in Figures A8c and A8d, where the black dots represent prompt completions for general fiction stories by publicly available LMs. The creativity scores among these LMs are nearly indistinguishable (except for the smallest LMs, but even then, their scores are still high). We also see that creativity does not exhibit a dynamic range comparable to any other text quality measure. Below is an example generated by pythia-70m.
>
> > __In the quaint town of Ravenswood, nestled between the rolling hills of the countryside, a sense of tranquility reigned supreme. The residents, with their warm smiles and gentle mannerisms, embodied the very essence of bonhomie – an__ urban legend – which felt deeply left by the neighbours’ general expectation of human nature.\n\n“A world of hand flowers, fair girls, gulls, checkered lichen shoes,” is professor of social history and the former yearning to eradicate the dinosaur remains.\n\nThe New World has been seen as a ‘feral character’ by the biologist Marc Orr and an expert on perseverance and habit throughout life – in the case of sorts, their significant figures, they often sink into the tough old quarrels and feats of humankind from parochial matron after farm-like periods. Now an obvious museum says that, before people can recreate the glory in dying civilisations (whose hallowed living-rooms make much of this movie – there is a pun – Cream of the Fairness trilogy – and are a wonder for themselves), they al
>
> This pythia-70m generation received a creativity score of 80 and a coherence score of 10. In contrast, here is a generation by Llama-3.1-70B.
>
> > __In the quaint town of Ravenswood, nestled between the rolling hills of the countryside, a sense of tranquility reigned supreme. The residents, with their warm smiles and gentle mannerisms, embodied the very essence of bonhomie – an__ unspoken camaraderie that bound them together. The chimneys of each house billowed with smoke, a testament to the cozy hearths that illuminated the otherwise cold evenings. As the golden autumn leaves cascaded down from the trees, the townsfolk reveled in the simple pleasures of life, united by their shared appreciation for the beauty of their surroundings. Little did they know, beneath this picturesque veneer, a sinister secret lurked, threatening to shatter their idyllic existence.\nI. The Whispering Winds of Ravenswood\nAs the sun dipped beneath the horizon, casting a scarlet hue across the sprawling landscape, the winds began to stir. At first, it was a gentle whisper, a soft caress against the cheeks of those who strolled along the cobbled streets. However, as the night deepened
>
> This Llama-3.1-70B generation received scores of 85 for creativity and 95 for coherence. As before, we think it is reasonable to say that the pythia-70m generation is of lower quality.
>
> In summary, we agree that creativity can be a useful dimension of text quality, but our findings suggest that it should not be considered independently of the other measures we have examined. Given that creativity is a highly subjective concept, even more so than text quality, it can be interpreted in such a way that even nonsensical text might be considered highly creative from certain points of view.
>
> We thank the reviewer again for their constructive feedback and openness to discussing our work. We are happy to provide further clarification on this topic or any other concerns the reviewer may have.

---

> > ### Author Response · Authors · 2024-12-02
> >
> > With tonight being the final opportunity for reviewers to ask questions, we wanted to check if there are any additional clarifications we can provide. If you feel that we have satisfactorily addressed your concerns, we hope you might consider updating our score. Thank you again for your time and assistance in improving this paper.

---

### Official Review · Reviewer_r6Wo · 2024-11-16

**Soundness:** 3
**Presentation:** 3
**Contribution:** 2
**Rating:** 5
**Confidence:** 4

**Summary:**

This paper investigates the generation abilities of SLMs due to the readability of the training data.

They include a set of definitions of readability and generate two synthetic datasets (LlamaTales-Jr and LlamaTales-GRE) with the same data generation process. At the same time, LlamaTales-Jr is more readable, and based on the experiment, they found that the generation abilities of SLMs have no relationship with the readability of the training data. Simple language training would not accelerate coherence development.

**Strengths:**

The paper's question of investigating the assumption is meaningful.

This paper has a detailed definition of quality, evaluation criteria, and the experiment process. In the Appendix, the author gave a couple of examples of data generation. Experiments are conducted and hypotheses are evaluated.

**Weaknesses:**

* **Lack of Novelty and Insights**:
The main contribution of this paper is the empirical experiment on the investigation of the assumption. However, the author did not explain the reason behind the findings from their experiment and did not experiment on insights such as why it exist, how such findings matters. While the paper's finding is definitely meaningful to the machine learning and natural language processing community, the scope of this paper is lacking, and more insight is needed for providing a reason as for why the results are found this way.

* **Lack of clarity**:
While the author has detailed the experiment, the main paper is not well organized. Following the flow of the paper, figures/tables/sections are not consistent and be placed close to the context without disrupting the reading flow. Table 5-16 sampled prompts are not referenced in the main discussion. Figure 16 is not referenced in the main discussion and it is not clear why this figure is placed.

* **Experiment details**:
The prompt used for LLM for evaluation and data generation is consistent but I am concerned if such a prompt would lead to bias. There’s no discussion of other prompts being used in the experiment and no comparison of the result with other prompting methods.

**Questions:**

See weaknesses.

---

> ### Author Response · Authors · 2024-11-22
>
> Thank you for the constructive feedback. We address your three points below.
>
> 1. If we understand correctly, the reviewer has two questions: Why do our results matter? And what explains our results?
>
>     To address the first question, we believe that the argument boils down to whether one believes that anthropomorphizing LMs is problematic and that misinterpreting the findings of [1] facilitates the anthropomorphization of LMs. We have included a more thorough discussion under "Why do our results matter?" in the general response.
>
>     In response to the second question, our revised draft demonstrates a correlative (though not necessarily causal) relationship between $n$-gram diversity (unique $n$-gram counts) and dataset learnability. For a detailed explanation, please refer to "What explains our results?" in the general response.
>
> 2. We agree with your points regarding clarity and made the following changes in our revision:
>     - We reorganized the paper to better align figures and tables with their relevant sections. Additionally, we added bidirectional links between the main text and appendix figures/tables for easier navigation.
>     - In addition, we have added references to Tables 5-16 (now A6-20), specifically in Section 6.
>     - We have added reference to Figure 16 (now A37) in the caption for Figure 4. In short, the figure is meant as a companion to Figure 4 to allow the reader to compare $n$-gram novelty across our datasets which is difficult to see in Figure 4.
>
> 3. We share your concern regarding potential bias when utilizing LLM-as-a-Judge, particularly in the choice of the LLM and prompts.
>     We also want to emphasize that our goal is not to find the optimal prompt. We aim only to identify a prompt that is adequate for advancing our experiments.
>     That said, we have taken several steps to validate our use of LLM-as-a-Judge:
>
>     - Recognizing that LLM-as-a-Judge might be influenced by the choice of LM, we evaluated a wide range of LMs to see how their scores aligned with non-LLM assessments of readability and text quality. For readability, we tested 21 LMs against human evaluations, as detailed in the correlation matrix in Figure A33. We found several LMs with high human correlation, indicating that LLM-as-a-Judge is reliable for readability assessment, provided the LM is recent and has over 70B parameters.
>     For coherence, we examined 6 LMs, all of which showed strong correlation with our model ranking (Figure A31).
>
>     - We explored different prompt variations for assessing coherence and readability.
>     One variant instructs the LM to generate an analysis before scoring, unlike our original prompt, which asks for an immediate score. Another variant provides examples of low and high-rated texts with explanations, guiding the LM before scoring.
>     For coherence, all three variants showed a similar correlation with our model ranking and each other.
>     For readability, while all variants correlated strongly with human experts, the ones prompting immediate scoring had the highest correlation.
>     The prompts and correlation matrices are detailed in Figures A10-15 and Figures A35-36, respectively.
>
>     - In response to reviewer wvMq's feedback, we introduced five new prompts for an LM to evaluate clarity, fluency, creativity, consistency, and grammaticality, detailed in Figures A16-20. All metrics, except for creativity, supported our original findings (Figure A9), showing that our results were not unique to the coherence prompt. Creativity is uncorrelated with our non-LM measure (model ranking), indicating it may not be a reliable text quality indicator.
>
>     - We used three new prompts to generate three 1B token datasets: LlamaTales-{History, Sports, News}. Similar to LlamaTales-GRE, these new datasets exhibit substantially higher language complexity (as measured by readability) than the children's stories in LlamaTales-Jr. Yet, SLMs trained on these datasets are competitive with much larger models in terms of coherence. The specific prompts are shown in Figures A23-25. This demonstrates that our original findings are not unique to the prompt used to generate LlamaTales-GRE.

---

> > ### Comment · Area_Chair_TS51 · 2024-11-25
> > **[Reminder] Response to Authors**
> >
> > Dear Reviewer,
> >
> > As the rebuttal period is drawing to a close, I would appreciate your response to the authors' rebuttal at your earliest convenience.
> >
> > Best Regards,
> >
> > Area Chair

---

> ### Comment · Reviewer_r6Wo · 2024-11-27
>
> Thank you for providing further explanations and editions. Based on the paper and the comment made during the rebuttal, I can see that the result matters but is not significant. The authors presented n-gram diversity correlating with learnability to support their hypothesis, but they did not explain the underlying reasons for their observations. The results show that complex language data neither helps nor harms the coherence of SLMs. From the appendix, the model output is coherent with complex and simple language. The discussion of how much the new proposed research direction might be helpful to future works in this area is unclear. While this paper identifies interesting problems, the paper’s contribution to the broader community is limited. I decided to keep my original score.

---

> > ### Author Response · Authors · 2024-12-01
> >
> > > The authors presented n-gram diversity correlating with learnability to support their hypothesis, but they did not explain the underlying reasons for their observations.
> >
> > In our general response, we proposed an explanation for the underlying reasons behind our observations. Specifically, we hypothesized that the primary difference between the datasets is their diversity, as measured by unique $n$-gram count. FineWeb (along with Dolma and SlimPajama), which is derived from web crawl data, encompasses a much broader range of language than our other datasets, which were generated by a LM using template-based prompts with minor variations. We then presented correlative evidence supporting this hypothesis.
> >
> > If by "did not explain the underlying reasons," the reviewer means that we did not establish causality, then we contend that providing correlative evidence is reasonable, given that the primary contribution of our paper is a careful and controlled study of the effects of training data readability, which was not performed by prior work.
> >
> > In particular, [1] examined a single dataset characterized by child-directed language. This has led some researchers and the public to mistakenly attribute the findings of [1] to this type of language, thereby promoting the anthropomorphization of LMs and a misunderstanding of how LMs learn. To clarify this misconception, we analyzed five synthetic datasets and three web crawl-based datasets, each with varying readability levels, and demonstrated that child-directed language was not necessary for achieving coherent SLMs.
> >
> > [1] also claims that TinyStories consists solely of child-directed language, but this claim was never validated. We address this issue in Section 3 by conducting a comprehensive study of various readability measures. Additionally, we introduced a readability measurement method that strongly correlates with human experts' assessments.
> >
> > Given the influence of [1] (as discussed in the general response), we hope the reviewer sees the importance of addressing the gaps in prior work and correcting the misunderstandings that have resulted from it.
> >
> > > The results show that complex language data neither helps nor harms the coherence of SLMs. From the appendix, the model output is coherent with complex and simple language.
> >
> > This evidence helps clarify that child-directed language is not responsible for the results in [1]. If the reviewer is interested in the differences between simple and complex language in affecting coherence, Figure 3 may be relevant. Surprisingly, we found that coherence is achieved _earlier_ in training with complex language compared to simple language.
> >
> > > The discussion of how much the new proposed research direction might be helpful to future works in this area is unclear. While this paper identifies interesting problems, the paper’s contribution to the broader community is limited.
> >
> > Our goal is to correct a misunderstanding within the research community and the public. We believe this will benefit future work, as research based on incorrect assumptions from misinterpreted studies can lead to inaccurate conclusions, further propagating misinformation.

---

> > > ### Comment · Reviewer_r6Wo · 2024-12-02
> > >
> > > Thank you for your thorough explanations addressing my concerns. I agree with the author and feel that this paper is meaningful to the relevant research community.
> > >
> > > While this paper identifies an interesting problem and provides correlative results on the findings (although not causal, which is fine to me), the scope of the impact of this work is a bit limited for a top venue with a broad audience like ICLR.
> > >
> > > Based on the new discussions, I raised my score, but I hold my concern that it is unclear how someone outside of this research community could learn and conduct future work from this paper's findings. This work is probably more relevant and suitable for language and linguistic venues.

---

### Author Response · Authors · 2024-11-22
**General response**

# General Response

We thank the reviewers for their time and constructive feedback. We have identified two core questions raised throughout the reviews and addressed them below. Additionally, specific clarifications for each reviewer are provided in separate responses. Note that all references in the individual responses can be found at the end of this general response.

## Why do our results matter?
1. __Anthropomorphizing LMs can be problematic.__
    Given that LLMs can reduce tasks that were formerly thought to require human intelligence to next-token prediction and improve their performance simply by being instructed to "think step by step," it is highly tempting to anthropomorphize these models.
    Consequently, the research community has acknowledged the potential harms of anthropomorphizing LMs, and examining these risks is an active area of research [2, 3, 4, 5, 6].
    For example, [2] argues that anthropomorphism tends to exaggerate and misrepresent LLM capabilities by attributing human-like attributes onto systems that do not possess them.
    Moreover, via the same mechanism, anthropomorphism distorts judgments of responsibility and trust in LLMs.

2. __Drawing parallels between human cognitive development and LM training is an example of anthropomorphization.__
    While much of the conversation centers around interactions with _trained_ LMs, anthropomorphization also manifests itself in the way we think and talk about LM training.
    Indeed, terms like "grokking" [7] and even the commonly used "learning" can evoke a sense of human-like understanding and cognitive processes.
    While we see no issue with drawing inspiration from cognitive development, discussions suggestive of a deeper connection between the human learning process and stochastic gradient descent should be approached with care.

3. __The results of [1] are both influential and can be easily misinterpretted as evidence that human cognitive development and LM training are more related than originally thought.__
    At the time of writing, Semantic Scholar reports 175 citations of [1], and TinyStories is the 22nd most liked dataset on Hugging Face. Their findings were also widely circulated on social media platforms [10] and was featured by Quanta Magazine [8].
    Because of [1]'s emphasis on using only words suitable for 3-4-year-old children, it is tempting to attribute their findings to simple, child-directed language rather than the low number of statistical patterns that come with synthesizing a dataset with a template-based prompt with minor variations among the instances of the prompt.
    This temptation is further fueled by the community's interest in developmentally plausible pre-training [9], as one might reasonably, though mistakenly, interpret the findings of [1] as supporting evidence for this research area.
    Our goal is to correct this misinterpretation.

4. __Citations of [1] suggest that their results are being misinterpreted or present the findings in ways that emphasize the importance of child-directed language.__
    For example, [11] write "Recent studies have shown the benefits of mimicking human language acquisition. For instance, using child-oriented vocabulary and/or child-directed speech (CDS) as learning data improves learning efficiency."
    Many more papers emphasize the simple, child-directed language of TinyStories when describing the dataset [12, 13, 14, 15], further amplifying the temptation to anthropomorphize LM training.

---

> ### Author Response · Authors · 2024-11-22
>
> ## What explains our results?
>
> In our original draft, we observed that SLMs trained on datasets with either simple language (TinyStories, LlamaTales-Jr) or complex language (LlamaTales-GRE) could generate coherent text comparable to much larger models. However, this was not the case for SLMs trained on FineWeb, our representative of data used to train real-world LLMs.
>
> We hypothesize that the primary difference between the datasets is their _diversity_. FineWeb, derived from web crawl data, encompasses a much broader range of language than our other datasets, which were generated by a LM using template-based prompts with minor variations. To quantify diversity, we propose measuring unique $n$-gram counts. We find that FineWeb contains significantly more unique $n$-grams than TinyStories and LlamaTales. However, this leaves us with only four data points to draw conclusions from. In our revised draft, we present additional correlative evidence suggesting that $n$-gram diversity explains our findings.
>
> First, we generate three new synthetic datasets, each containing 1 billion tokens: Llamatales-{Sports, History, News}. These datasets are created using the same generation process as LlamaTales-GRE, but with the prompts shown in Figures A23-25 to produce documents outside the domain of short fiction stories. Similar to LlamaTales-GRE, all three datasets feature significantly more complex language than LlamaTales-Jr (as measured by readability, see Table A1), yet they produce coherent SLMs when trained on these datasets. We then examine two new examples of "real" training data for LMs (in addition to FineWeb): 1 billion token samples of Dolma and SlimPajama. Consistent with our experiments with FineWeb, SLMs trained on these datasets are incoherent.
> Examples from the new datasets are provided in Table A21, and the SLM coherency results are shown in Figure A9. We have results for 33M and 9M parameter SLMs; the other model sizes are still being trained and will be included in the next revision.
> We also present example generations by SLMs trained on the new LlamaTales datasets in Table A22.
>
> An examination of the $n$-gram diversity across our nine datasets reveals that easy-to-learn datasets (TinyStories, LlamaTales series) have significantly lower $n$-gram diversity compared to hard-to-learn datasets (FineWeb, Dolma, SlimPajama) for small values of $n$ (see Figures A26 and A29).
> This observation is intuitive. Easy-to-learn datasets contain fewer statistical patterns than hard-to-learn datasets, and SLMs have limited capacity to encode these patterns.
> Thus, our evidence suggests that the child-directed language of TinyStories is easy to learn because the data contain fewer patterns (in the form of unique $n$-grams), rather than due to any human concept of language complexity (e.g., grammatical, lexical, syntactic, or conceptual complexity).
>
> We will include this expanded discussion in our next revision.

---

> > ### Author Response · Authors · 2024-11-22
> > **References**
> >
> > #### References
> > [1]
> > Eldan, R., & Li, Y. TinyStories: How Small Can Language Models Be and Still Speak Coherent English? arXiv preprint arXiv:2305.07759 (2023). https://arxiv.org/abs/2305.07759
> >
> > [2] Placani, A. Anthropomorphism in AI: hype and fallacy. AI Ethics 4, 691–698 (2024). https://doi.org/10.1007/s43681-024-00419-4
> >
> > [3] Shanahan, M. Talking About Large Language Models. arXiv preprint arXiv:2212.03551 (2023). https://arxiv.org/abs/2212.03551
> >
> > [4]
> > Deshpande, A., Rajpurohit, T., Narasimhan, K., & Kalyan, A. Anthropomorphization of AI: Opportunities and Risks. arXiv preprint arXiv:2305.14784 (2023). https://arxiv.org/abs/2305.14784
> >
> > [5] Abercrombie, G., Cercas Curry, A., Dinkar, T., Rieser, V., & Talat, Z. Mirages. On Anthropomorphism in Dialogue Systems. In Proceedings of the 2023 Conference on Empirical Methods in Natural Language Processing, 4776–4790 (2023). Association for Computational Linguistics, Singapore. https://doi.org/10.18653/v1/2023.emnlp-main.290
> >
> >
> > [6] Cohn, M., Pushkarna, M., Olanubi, F., Mengesha, Z., Moran, J., Padgett, D., & Heldreth, C. Believing Anthropomorphism: Examining the Role of Anthropomorphic Cues on User Trust in Large Language Models. Late Breaking Work (2024).
> >
> > [7] Power, A., Burda, Y., Edwards, H., Babuschkin, I., & Misra, V. Grokking: Generalization Beyond Overfitting on Small Algorithmic Datasets. arXiv preprint arXiv:2201.02177 (2022). https://arxiv.org/abs/2201.02177
> >
> > [8] https://www.quantamagazine.org/tiny-language-models-thrive-with-gpt-4-as-a-teacher-20231005/
> >
> > [9] Warstadt, A., Mueller, A., Choshen, L., Wilcox, E., Zhuang, C., Ciro, J., Mosquera, R., Paranjabe, B., Williams, A., Linzen, T., & Cotterell, R. Proceedings of the BabyLM Challenge at the 27th Conference on Computational Natural Language Learning. Association for Computational Linguistics, Singapore (2023). https://aclanthology.org/2023.conll-babylm.pdf
> >
> > [10] https://www.reddit.com/r/MachineLearning/comments/13j0spj/r_tiny_language_models_below_10m_parameters_or/
> >
> > [11] Haga, A., Sugawara, S., Fukatsu, A., Oba, M., Ouchi, H., Watanabe, T., & Oseki, Y. Modeling Overregularization in Children with Small Language Models. Findings of the Association for Computational Linguistics: ACL 2024, 14532–14550 (2024). https://doi.org/10.18653/v1/2024.findings-acl.865
> >
> > [12] Bunzeck, B., & Zarrieß, S. GPT-wee: How Small Can a Small Language Model Really Get? Proceedings of the BabyLM Challenge at the 27th Conference on Computational Natural Language Learning, 35–46 (2023). https://doi.org/10.18653/v1/2023.conll-babylm.2.
> >
> > [13] Edman, L., & Bylinina, L. Too Much Information: Keeping Training Simple for BabyLMs. Proceedings of the BabyLM Challenge at the 27th Conference on Computational Natural Language Learning, 89–97 (2023). https://doi.org/10.18653/v1/2023.conll-babylm.8.
> >
> > [14] Yam, H. M., & Paek, N. J. What Should Baby Models Read? Exploring Sample-Efficient Data Composition on Model Performance. arXiv preprint arXiv:2411.06672 (2024). https://arxiv.org/abs/2411.06672.
> >
> > [15] Steven Y. Feng, Noah D. Goodman, and Michael C. Frank. Is child-directed speech effective
> > training data for language models? EMNLP 2024. https://arxiv.org/abs/2408.03617

---

> ### Author Response · Authors · 2024-11-24
> **Summary of new experiments, figures, and tables**
>
> We have included a summary of our new experiments, along with the corresponding tables and figures, to give reviewers a clear overview of our updates.
>
> 1. **Five new datasets** (3 synthetic: Llamatales-{History, Sports, News} and 2 human-authored: Dolma, SlimPajama)
>     - Goal: Demonstrate more correlative evidence for why $n$-gram diversity, rather than child-directed language, may better explain our findings.
>     - Table A1: Coherence and readability of training splits (measured with LLM-as-a-Judge)
>         - Takeaway: Similar to LlamaTales-GRE, our new synthetic datasets score high on coherence and low on readability (relative to TinyStories and Llamatales-Jr).
>     - Figure A9: Coherence of SLMs trained on new datasets versus public models.
>         - Takeaway: Consistent with our original findings, SLMs trained on our new synthetic datasets are competitive with much larger models. This is not true for SLMs trained on our new human-authored datasets.
>     - Figures A26-28: Histograms of $n$-gram diversity (unique $n$-gram counts for various choices of $n$) across our datasets.
>         - Takeaway: Easy-to-learn data (TinyStories, LlamaTales) exhibit far less $n$-gram diversity than hard-to-learn data (FineWeb, SlimPajama, Dolma).
>     - Figure A29: Unique 3-grams (training data) versus _learnability_ (coherence of text generated by a 33M parameter SLM divided by the coherence of the training data used for that SLM).
>         - Takeaway: Low $n$-gram diversity strongly correlates with high learnability.
>     - Figures A23-25: Prompts used to generate our synthetic data.
>     - Table A21: Random examples from our new datasets.
>     - Table A22: Random generations by SLMs trained on our new synthetic data.
> 2. **Two new prompt variants for LLM-as-a-Judge**
>     - Goal: Address reviewer r6Wo's concern that
>         > The prompt used for LLM for evaluation and data generation is consistent but I am concerned if such a prompt would lead to bias. There’s no discussion of other prompts being used in the experiment and no comparison of the result with other prompting methods.
>     - Figures A12-15: Two new prompt variants: (a) instructing the LLM to generate an analysis _before_ assigning a score, and (b) providing the LLM with positive/negative examples in its instructions. We apply both variants to measure coherence and readability.
>     - Figures A35-36: Correlation matrices of the prompt variants.
>         - Takeaway: When measuring coherence, we observed no differences among the variants; all were strongly correlated with our model ranking (Section 4.2). When measuring readability, all variants correlated strongly with human experts, but generating an analysis before assigning a score was the weakest of the three.
> 3. **Five new metrics:** fluency, clarity, consistency, grammar, and creativity
>     - Goal: address reviewer wvMq's concerns:
>         > The ignorance of other dimensions of quality (for example, as authors also mentioned, clarity and fluency) makes any statements about "generation abilities of SLMs" an overclaim.
>
>         > The quality measurement doesn't use any metrics from the original TinyStories paper: grammar, creativity, consistency with the beginning of the story.
>     - Figures A4-A8: Results of evaluating SLMs (trained from scratch) and public LMs with the new metrics.
>         - Takeaway: All new metrics (except for creativity) strongly correlate with our original metric: coherence. This supports our original finding that SLMs trained on complex language (instead of the simple language of TinyStories and LlamaTales-Jr) can still be competitive with much larger LMs.
>         Creativity did not correlate with any of our quality measures and often ranked generations by toy models such as GPT2-small and Pythia-70M as being highly creative, suggesting this is not a good measure of text quality (Figure A8).
>     - Figures A16-20: Prompts for our new metrics.

---

### Meta-Review · Area_Chair_TS51 · 2024-12-20

**Metareview:**

This paper investigates whether the ability of small language models (SLMs) to learn from TinyStories is due to the simplicity or readability of the language used in that dataset (as suggested by the original TinyStories authors and many subsequent works). The authors hypothesize that the success of SLMs on TinyStories might instead be attributed to the low diversity of the data, as it was generated using templates with minor variations. To test this, they created new synthetic datasets, LlamaTales-Jr (simple language) and LlamaTales-GRE (complex language), using the same generation process but varying vocabulary and target audience. Their experiments found that SLMs trained on both LlamaTales-Jr and LlamaTales-GRE achieved comparable coherence to SLMs trained on TinyStories, suggesting that readability is not the primary factor. Further experiments with diverse, real-world datasets like FineWeb, Dolma, and SlimPajama showed that SLMs trained on these datasets struggled to generate coherent text. The authors then explored n-gram diversity as a potential explanation, finding that datasets easier for SLMs to learn from (including the LlamaTales datasets) exhibited significantly lower n-gram diversity compared to the harder-to-learn real-world datasets. This leads to the finding that n-gram diversity appears to be a strong correlate with dataset learnability for SLMs. The paper challenges the interpretation that the success of TinyStories is solely due to child-directed language, arguing that it promotes anthropomorphism and misunderstanding of LM learning.

This paper addresses a meaningful assumption in the NLP community - if simple language is easier to learn. The experimental design is sound and the paper is well-written and clearly scoped. The study corrects a potential misinterpretation of influential prior work, particularly regarding the TinyStories paper. The investigation into n-gram diversity provides a plausible alternative explanation for the observed phenomena.

Despite the acknowledged value of the empirical evidence, several limitations hinder the paper's impact. The analysis primarily presents findings without deeply investigating the fundamental reasons behind them. Concerns persist regarding potential biases introduced by the LLM-as-a-Judge evaluation method. Furthermore, the paper's focus on a specific misinterpretation may limit its broader impact and generalizability for the wider ICLR community. Critically, the inherent differences between the simplistic synthetic datasets and the complex real-world data like FineWeb limit the conclusiveness of the findings, particularly regarding the nuances of human language. Strengthening the conclusions would necessitate demonstrating that synthetic datasets, while controlling for high n-gram diversity, could still impede the generation of coherent text by SLMs.

While I appreciate the paper's motivation and the insights it provides, the experimental results don't fully support the current conclusion. Coupled with its relatively limited potential impact as the current framing, I lean towards rejecting this paper.

**Additional Comments On Reviewer Discussion:**

Reviewers initially raised concerns about clarity, the breadth of quality metrics, potential evaluation bias, and the overall impact of the findings. The authors responded by adding discussion, restructuring the paper for improved flow, and implementing better cross-referencing between the main text and appendix. Although most reviewers actively participated in the rebuttal process, the decision to recommend rejection stems primarily from the limited broad impact of the work (a point raised by reviewer r6Wo) and the insufficient strength of the experimental results to fully support the stated conclusion (partially mentioned by reviewer wvMq, although they have increased the score, but the area chair still has some concerns regarding this).

---

### Decision · Program_Chairs · 2025-01-22

Reject